# Brain-actuated functional electrical stimulation elicits lasting arm motor recovery after stroke

A. Biasiucci [1], R. Leeb[1,2], I. Iturrate[1], S. Perdikis[1,3], A. Al-Khodairy[4], T. Corbet[1], A. Schnider[5], T. Schmidlin[2], H. Zhang[1], M. Bassolino[2], D. Viceic[2], P. Vuadens[4], A.G. Guggisberg[5] & J.d.R. Millán[1]

Brain-computer interfaces (BCI) are used in stroke rehabilitation to translate brain signals into intended movements of the paralyzed limb. However, the efficacy and mechanisms of BCI-based therapies remain unclear. Here we show that BCI coupled to functional electrical stimulation (FES) elicits significant, clinically relevant, and lasting motor recovery in chronic stroke survivors more effectively than sham FES. Such recovery is associated to quantitative signatures of functional neuroplasticity. BCI patients exhibit a significant functional recovery after the intervention, which remains 6–12 months after the end of therapy. Electro-encephalography analysis pinpoints significant differences in favor of the BCI group, mainly consisting in an increase in functional connectivity between motor areas in the affected hemisphere. This increase is significantly correlated with functional improvement. Results illustrate how a BCI–FES therapy can drive significant functional recovery and purposeful plasticity thanks to contingent activation of body natural efferent and afferent pathways.

[1] Defitech Foundation Chair in Brain-Machine Interface, Center for Neuroprosthetics and Institute of Bioengineering, École Polytechnique Fédérale de Lausanne, Geneva 1202, Switzerland. [2] Center for Neuroprosthetics, École Polytechnique Fédérale de Lausanne, Sion 1951, Switzerland. [3] Wyss Center for Bio and Neuroengineering, Geneva 1202, Switzerland. [4] SUVACare - Clinique Romande de Réadaptation, Sion 1951, Switzerland. [5] Division of Neurorehabilitation, Department of Clinical Neurosciences, University Hospital of Geneva, Geneva 1211, Switzerland. These authors contributed equally: A. Biasiucci, R. Leeb. Correspondence and requests for materials should be addressed to J.d.R.Mán. (email: jose.millan@epfl.ch)

Despite considerable efforts over the last decades, the quest for novel treatments for arm functional recovery after stroke remains a priority[1]. Synergistic efforts in neural engineering and restoration medicine are demonstrating how neuroprosthetic approaches can control devices and ultimately restore body function[2–7]. In particular, non-invasive brain-computer interfaces (BCI) are reaching their technological maturity[8,9] and translate neural activity into meaningful outputs that might drive activity-dependent neuroplasticity and functional motor recovery[10–12]. BCI implies learning to modify the neuronal activity through progressive practice with contingent feedback and reward —sharing its neurobiological basis with rehabilitation[13].

Most attempts to use non-invasive BCI systems for upper limb rehabilitation after stroke have coupled them with other interventions, although not all trials reported clinical benefits. The majority of these studies are case reports of patients who operated a BCI to control either rehabilitation robots[14–19] or functional electrical stimulation (FES)[20–23]. A few works have described changes in functional magnetic resonance imaging (fMRI) that correlate with motor improvements[17,18,22].

Recent controlled trials have shown the potential benefit of BCI-based therapies[24–27]. Pichiorri et al.[26] recruited 28 subacute patients and studied the efficacy of motor imagery with or without BCI support via visual feedback, reporting a significant and clinically relevant functional recovery for the BCI group. As a step forward in the design of multimodal interventions, BCI-aided robotic therapies yielded significantly greater motor gains than robotic therapies alone[24,25,27]. In the first study, involving 30 chronic patients[24], only the BCI group exhibited a functional improvement. In the second study, involving 14 subacute and chronic patients, both groups improved, probably reflecting the larger variance in subacute patients' recovery and a milder disability[25]. The last study[27] showed that in a mixed population of 74 subacute and chronic patients, the percentage of patients who achieved minimally clinical important difference in upper limb functionality was higher in the BCI group. The effect in favor of the BCI group was only evident in the subpopulation of chronic patients. Moreover, the conclusions of this study are limited due to differences between experimental and control groups prior to the intervention, such as number of patients and FMA-UE scores, which were always in favor of the BCI group.

In spite of promising results achieved so far, BCI-based stroke rehabilitation is still a young field where different works report variable clinical outcomes. Furthermore, the efficacy and mechanisms of BCI-based therapies remain largely unclear. We hypothesize that, for BCI to boost beneficial functional activity-dependent plasticity able to attain clinically important outcomes, the basic premise is contingency between suitable motor-related cortical activity and rich afferent feedback. Our approach is designed to deliver associated contingent feedback that is not only functionally meaningful (e.g., via virtual reality or passive movement of the paretic limb by a robot), but also tailored to

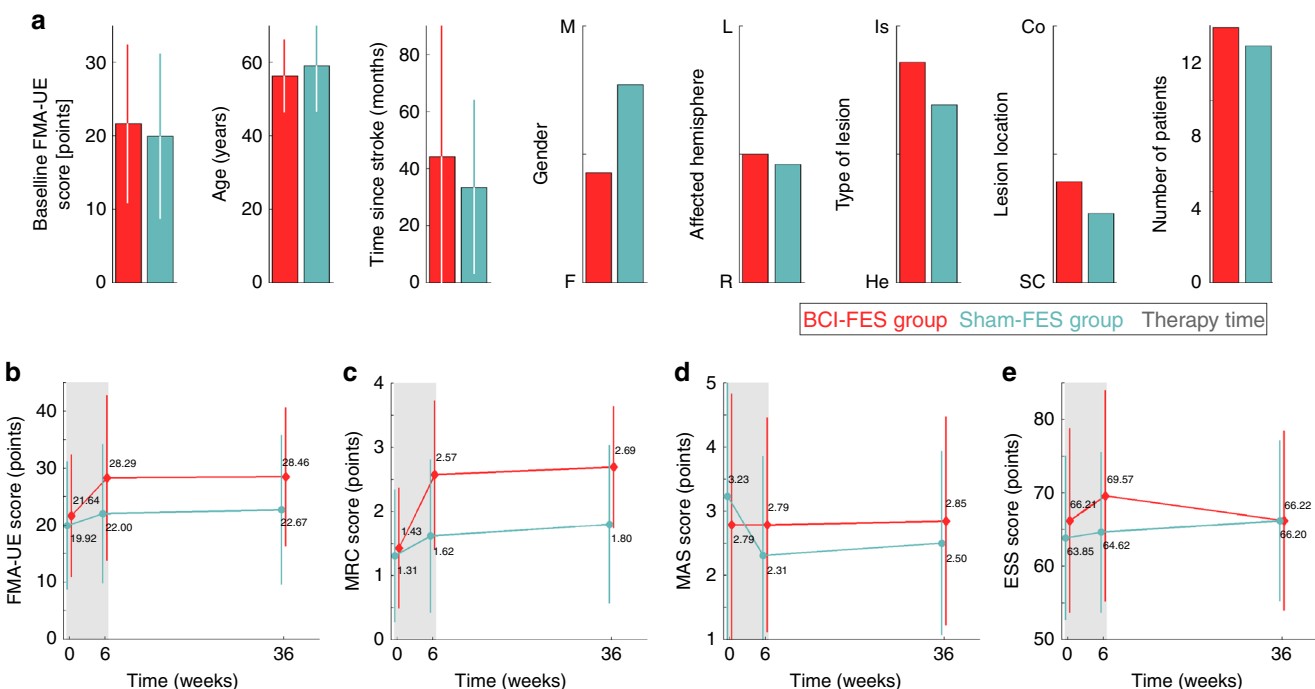

**Fig. 1** Patient demographics and clinical scores. All plots report mean values ± standard deviation for BCI–FES group ($N = 14$, red) and sham-FES group ($N = 13$, light blue). **a** Patients' main characteristics, including baseline Fugl-Meyer score (upper extremity), age, time since stroke, gender, affected hemisphere, type of lesion, lesion location, and number of patients per group. No statistical significant difference between groups was found for any of these factors before the intervention ($p > 0.05$ for all tests). **b** Primary outcome is the Fugl-Meyer assessment for the upper extremity (FMA-UE), measuring motor function. FMA-UE scores are reported immediately before patients received the intervention, immediately after it ended (6 weeks) and at a follow-up session done 6–12 months after the end of the intervention (average 36 weeks). The BCI group exhibited a significant (TIME x GROUP interaction, $p = 0.04$) and clinically relevant functional recovery after the intervention ($6.6 \pm 5.6$ FMA-UE points, above the threshold of 5 points) that was retained 6–12 months after the end of the therapy (Bonferroni-corrected two-tailed paired $t$-test, $p = 0.56$). **c–e** Secondary outcome scores: Medical Research Council Scale (MRC), measuring muscle strength, Modified Ashworth Scale (MAS), measuring spasticity, and European Stroke Scale (ESS), measuring the overall motor and cognitive state. As for the primary clinical outcome, target muscle strength recovery ($1.1 \pm 1.1$ MRC points) was significant for the BCI-FES group (Bonferroni-corrected non-parametric signed-rank test, $p = 0.02$), but not for the sham-FES group ($p = 0.11$). BCI group also retained improvement in MRC scores at the follow-up clinical evaluation ($p = 0.69$). No significant differences were found for the ESS scores or the Ashworth wrist extension score (mixed ANOVA, $p > 0.05$ for all tests)

**Table 1 Clinical scores for all patients**

| ID code | Fugl-Meyer UE | | | Ashworth wrist flexor | | | Ashworth wrist extensor | | | MRC wrist extensor | | | ESS | | |
|---|---|---|---|---|---|---|---|---|---|---|---|---|---|---|---|
| | Pre | Post | Follow-up | Pre | Post | Follow-up | Pre | Post | Follow-up | Pre | Post | Follow-up | Pre | Post | Follow-up |
| BCI01 | 11 | 21 | 21 | 2 | 2 | 2 | 1 | 0 | 1 | 1 | 1 | 2 | 61 | 62 | N/A |
| BCI02 | 35 | 44 | 45 | 1 | 1 | 1 | 1 | 1 | 0 | 2 | 4 | 4 | 79 | 100 | N/A |
| BCI03 | 37 | 55 | 44 | 2 | 2 | 3 | 0 | 0 | 0 | 2 | 4 | 3 | 84 | 84 | N/A |
| BCI04 | 35 | 44 | 35 | 2 | 2 | 2 | 1 | 1 | 1 | 3 | 3 | 4 | 79 | 79 | N/A |
| BCI05 | 7 | 14 | N/A | 4 | 3 | N/A | 4 | 3 | N/A | 0 | 3 | N/A | 49 | 55 | N/A |
| BCI06 | 23 | 31 | 28 | 2 | 2 | 3 | 0 | 0 | 0 | 2 | 3 | 3 | 70 | 72 | 70 |
| BCI07 | 24 | 28 | 31 | 0 | 1 | 2 | 3 | 3 | 3 | 1 | 2 | 3 | 76 | 78 | 78 |
| BCI08 | 35 | 47 | 48 | 0 | 1 | 0 | 0 | 0 | 0 | 1 | 4 | 4 | 77 | 81 | 85 |
| BCI09 | 11 | 11 | 14 | 2 | 3 | 2 | 1 | 1 | 2 | 0 | 2 | 2 | 39 | 41 | 41 |
| BCI10 | 14 | 26 | 26 | 1 | 1 | 3 | 0 | 0 | 0 | 2 | 3 | 2 | 57 | 62 | 62 |
| BCI11 | 22 | 22 | 23 | 3 | 3 | 3 | 2 | 1 | 0 | 2 | 3 | 3 | 65 | 65 | 65 |
| BCI12 | 8 | 8 | 8 | 2 | 3 | 3 | 2 | 2 | 3 | 2 | 2 | 2 | 63 | 63 | 63 |
| BCI13 | 16 | 16 | 18 | 0 | 0 | 1 | 0 | 0 | 0 | 0 | 0 | 1 | 61 | 62 | 62 |
| BCI14 | 25 | 29 | 29 | 2 | 2 | 2 | 1 | 1 | 0 | 2 | 2 | 2 | 67 | 70 | 70 |
| mean | 21.6 | 28.3 | 28.5 | 1.6 | 1.9 | 2.1 | 1.1 | 0.9 | 0.8 | 1.4 | 2.6 | 2.7 | 66.2 | 69.6 | 66.2 |
| std | 10.8 | 14.5 | 12.2 | 1.2 | 0.9 | 1.0 | 1.2 | 1.1 | 1.2 | 0.9 | 1.2 | 0.9 | 12.6 | 14.4 | 12.3 |
| sham01 | 23 | 31 | 26 | 3 | 2 | N/A | 3 | 1 | N/A | 1 | 1 | N/A | 68 | 73 | N/A |
| sham02 | 4 | 8 | 10 | 3 | 2 | 2 | 3 | 1 | 1 | 0 | 1 | 1 | 38 | 41 | 41 |
| sham03 | 5 | 4 | 5 | 4 | 3 | N/A | 4 | 2 | N/A | 0 | 0 | N/A | 50 | 50 | N/A |
| sham04 | 25 | 24 | 30 | 3 | 1 | 3 | 2 | 1 | 1 | 1 | 1 | 1 | 70 | 70 | 70 |
| sham05 | 11 | 11 | 11 | 0 | 0 | 0 | 0 | 0 | 0 | 2 | 3 | 3 | 64 | 64 | 64 |
| sham06 | 8 | 8 | 8 | 3 | 3 | 1 | 0 | 0 | 0 | 0 | 0 | 0 | 61 | 61 | 61 |
| sham07 | 19 | 21 | 22 | 3 | 2 | 3 | 1 | 0 | 1 | 2 | 2 | 2 | 70 | 72 | 72 |
| sham08 | 25 | 32 | 37 | 1 | 2 | 3 | 1 | 1 | 0 | 2 | 3 | 3 | 77 | 77 | 77 |
| sham09 | 22 | 26 | 31 | 3 | 3 | 3 | 1 | 1 | 1 | 2 | 2 | 3 | 68 | 68 | 68 |
| sham10 | 40 | 43 | 45 | 1 | 1 | 2 | 0 | 1 | 0 | 3 | 3 | 3 | 79 | 79 | 77 |
| sham11 | 31 | 32 | N/A | 0 | 0 | N/A | 0 | 0 | N/A | 2 | 3 | N/A | 57 | 57 | N/A |
| sham12 | 13 | 13 | 13 | 3 | 3 | 3 | 0 | 0 | 0 | 0 | 0 | 0 | 58 | 58 | 58 |
| sham13 | 33 | 33 | 34 | 0 | 0 | 0 | 0 | 0 | 1 | 2 | 2 | 2 | 70 | 70 | 74 |
| mean | 19.9 | 22.0 | 22.7 | 2.1 | 1.7 | 2.0 | 1.2 | 0.6 | 0.5 | 1.3 | 1.6 | 1.8 | 63.8 | 64.6 | 66.2 |
| std | 11.2 | 12.2 | 13.1 | 1.4 | 1.2 | 1.2 | 1.4 | 0.7 | 0.5 | 1.0 | 1.2 | 1.2 | 11.2 | 11.0 | 11.0 |

For every clinical score, values are reported for pre, post, and follow-up evaluations

reorganize the targeted neural circuits by providing rich sensory inputs via the natural afferent pathways[28], so as to activate all spare components of the central nervous system involved in motor control. FES fulfills these two properties of feedback contingent on appropriate patterns of neural activity; it elicits functional movements and conveys proprioceptive and somatosensory information, in particular via massive recruitment of Golgi tendon organs and muscle spindle feedback circuits. Moreover, several studies suggest that FES has an impact on cortical excitability[29,30].

To test our hypothesis, this study assessed whether BCI-actuated FES therapy targeting the extension of the affected hand (BCI–FES) could yield stronger and clinically relevant functional recovery than sham-FES therapy for chronic stroke patients with a moderate-to-severe disability, and whether signatures of functional neuroplasticity would be associated with motor improvement. Whenever the BCI decoded a hand-extension attempt, it activated FES of the extensor digitorum communis muscle that elicited a full extension of the wrist and fingers. Patients in the sham-FES group wore identical hardware and received identical instructions as BCI–FES patients, but FES was delivered randomly and not driven by neural activity.

As hypothesized, our results confirm that only the BCI group exhibit a significant functional recovery after the intervention, which is retained 6–12 months after the end of therapy. Besides the main clinical findings, we have also attempted to shed light on possible mechanisms underlying the proposed intervention. Specifically, electroencephalography (EEG) imaging pinpoint

significant differences in favor of the BCI group, mainly an increase in functional connectivity between motor areas in the affected hemisphere. This increase is significantly correlated with functional improvement. Furthermore, analysis of the therapeutic sessions substantiates that contingency between motor-related brain activity and FES occurs only in the BCI group and contingency-based metrics correlate with the functional improvement and increase in functional connectivity, suggesting that our BCI intervention might have promoted activity-dependent plasticity.

## Results

**Clinical outcome metrics**. Fig. 1b–e and Table 1 report the primary and secondary clinical scores for all patients (pre, post, and follow-up). Two patients, one per group, could not do the follow-up clinical evaluation. Previous to the intervention, the two groups were statistically homogeneous. No significant differences were found in clinical scores FMA-UE (two-tailed unpaired $t$-test, $p = 0.69$) and MRC (Wilcoxon rank-sum test, $p = 0.79$), time since stroke (two-tailed unpaired $t$-test, $p = 0.49$), lesion type (Fisher's exact test, $p = 0.38$), or lesion location (Wilcoxon rank-sum test, $p = 0.39$). Similarly, no significant demographic differences were found in age (two-tailed unpaired $t$-test, $p = 0.54$) or gender (Fisher's exact test, $p = 0.12$). Also, patients in the two groups had similar levels of cognitive impairments that did not prevent them from following instructions, as measured with the Raven test (two-sample unpaired $t$-

**Table 2 FMA-UE subscores for the Wrist and Hand sections**

| | Wrist (max 10 pt) | | | Hand (max 14 pt) | | |
| --- | --- | --- | --- | --- | --- | --- |
| | Pre | Post | Follow-up | Pre | Post | Follow-up |
| BCI | 1.3 ± 1.3 [0, 4] | 3.2 ± 3.1 [0, 8] | 2.5 ± 2.6 [0, 8] | 2.9 ± 3.1 [0, 11] | 4.7 ± 4.1 [0, 12] | 4.3 ± 4.3 [0, 13] |
| Sham | 2.0 ± 2.7 [0, 7] | 2.6 ± 3.3 [0, 9] | 2.4 ± 3.0 [0, 9] | 3.4 ± 3.0 [0, 8] | 3.7 ± 3.0 [0, 8] | 4.8 ± 4.0 [0, 10] |

Mean ± standard deviation and minimum/maximum range. Wrist section: max 10 points. Hand section: max 14 points

test, $p = 0.15$; corrected scores obtained in a sub-group of patients: 11 and 9 patients in the BCI and sham groups, respectively: BCI $= 33.0 ± 11.5$; sham $= 25.8 ± 9.5$).

Both groups received a similar amount of therapy in terms of the number of runs per session (BCI $= 6.0 ± 0.7$; sham $= 5.9 ± 0.7$, $p = 0.73$, Wilcoxon rank-sum test). For the FMA-UE score (Fig. 1b), a significant effect was found for the TIME × GROUP interaction (mixed-design ANOVA, $F_{1,46} = 3.5$, $p = 0.04$). Post-hoc tests revealed a significant increase for the BCI group (two-tailed paired $t$-test, $p = 0.005$), but not for the sham group (two-tailed paired $t$-test, $p = 0.21$). BCI patients improved by $6.6 ± 5.6$ points [0, 18], above the threshold of 5 points considered being clinically relevant[31], whereas sham patients did only by $2.1 ± 3.0$ points [−1, 8]. This post–pre difference was statistically significant (two-tailed paired $t$-test, $p = 0.0143$). Eight BCI patients recovered five or more FMA-UE points, while only two sham patients did (odds ratio $= 7.33$, 95% confidence interval $= 1.16$-$46.23$, $p = 0.03$). The effect size of the BCI–FES intervention was large (Cohen's $d = 1.03$). Furthermore, BCI subjects retained improvements 6–12 months after the end of therapy (pre $= 21.6 ± 10.8[7,37]$, post $= 28.3 ± 14.5[8,55]$, follow-up $= 28.5 ± 12.2$ [8,48]; two-tailed paired $t$-test, $p = 0.56$). For the sham group, follow-up scores did not differ either (two-tailed paired $t$-test, $p = 0.11$).

The increase in MRC score (strength of the target muscle extensor digitorum communis) was also significantly larger for the BCI group compared to the sham group (post–pre intervention between groups, Wilcoxon rank-sum test, $p = 0.03$) (Fig. 1c). Further analysis showed a significant pre vs post difference only for the BCI group (Bonferroni-corrected Wilcoxon signed-rank test, $p = 0.02$), but not for the sham group ($p = 0.5$). Similar to the FMA-UE, BCI group retained improvement in MRC scores at the follow-up clinical evaluation (BCI: pre $= 1.4 ± 0.9$ [0, 3], post $= 2.6 ± 1.2$ [0, 4], and follow-up $= 2.7 ± 0.9[1,4]$, $p = 0.69$). No significant differences were found neither for the ESS score (TIME × GROUP interaction: mixed-design ANOVA, $F_{1,25} = 2.73$, $p = 0.11$) (Fig. 1d) nor the Ashworth wrist extension score (post–pre intervention between groups, Wilcoxon rank-sum, $p = 0.42$) (Fig. 1e).

Since FES was applied to the hand muscles in order to generate a hand (wrist and fingers) extension, analysis of the FMA-UE scores of the wrist and hand revealed significant increases after intervention for the BCI group in these sections (Wilcoxon signed-rank test, $p = 0.008$ and $p = 0.004$ for wrist and hand, respectively), but not for the sham group (Wilcoxon signed-rank test, $p = 0.25$ and $p = 0.5$ for wrist and hand, respectively), see Table 2. For the wrist section (max 10 FMA points), the BCI and sham groups improved from 1.3 to 3.2 points and from 2.0 to 2.6 points, respectively. For the hand section (max 14 FMA points), the BCI and sham groups increased from 2.9 to 4.7 points and from 3.4 to 3.7 points, respectively. At the follow-up evaluation, FMA-UE scores of the wrist and hand sections for the BCI group decreased, but remained above the pre-intervention values (FMA-wrist: pre $= 1.3 ± 1.3$ [0, 4], post $= 3.2 ± 3.1$ [0, 8], and follow-up $= 2.5 ± 2.6$ [0, 8]; FMA-hand: pre $= 2.9 ± 3.1$ [0, 11],

post $= 4.7 ± 4.1$ [0, 12], and follow-up $= 4.3 ± 4.3$ [0, 13]). This decrease was not significant for either of the sections (Wilcoxon signed-rank test, post vs follow-up, $p = 0.063$ and $p = 0.63$ for wrist and hand, respectively). Thus, for the BCI group, the accumulated average increase in FMA-UE scores in the wrist and hand was 3.7 points, out of a total of 6.6. The intervention did not only lead to improvements in hand and wrist FMA-UE subscores, but also in the 'synergies' (pre vs post, BCI group $p = 0.02$; sham group $p = 0.19$; Wilcoxon signed-rank test, Bonferroni corrected) and 'combined movement synergies' subscores (pre vs post, BCI group $p = 0.03$; sham group $p = 0.50$).

**Increase of EEG connectivity within affected hemisphere.** Mixed-design ANOVA statistical tests were applied to measure changes in functional connectivity during the resting task (BCI trial where the patient is asked to rest). A significant effect was found for the TIME × GROUP interaction for the effective connectivity within the affected sensorimotor cortex (channels C5*, C3*, and C1*) for both μ and β frequency bands (μ: $F_{1,22} = 7.36$, $p = 0.013$); (β: $F_{1,22} = 5.47$, $p = 0.029$) (Fig. 2a, b). This improvement was significant for the BCI group comparing the pre vs post intervention in the μ band (two-tailed paired $t$-test, $p = 0.011$). Additionally, the differences post intervention between the two groups became significant both in the μ (two-tailed unpaired $t$-test, $p = 0.003$) and β bands (two-tailed unpaired $t$-test, $p = 0.035$). The effective connectivity from the affected sensorimotor to the supplementary motor area (SMA) and premotor regions (channels FC5*, FC3*, and FC1*) also showed a significant TIME × GROUP interaction in μ and β bands (μ: $F_{1,22} = 9.51$, $p = 0.005$; β: $F_{1,22} = 4.28$, $p = 0.049$) (Fig. 2c, d). Post-hoc tests showed a significant difference for both bands between the groups post intervention (two-tailed unpaired $t$-test, μ: $p = 0.007$ and β: $p = 0.005$, respectively), and for the pair pre/post for the BCI–FES group in μ band (two-tailed paired $t$-test, $p = 0.0003$). No significant interaction was found within the SMA and premotor region, nor from there to the sensorimotor cortex. Similarly, no significant changes were found in the effective connectivity neither between the two hemispheres nor within the healthy hemisphere.

In order to further investigate the possible role of functional connectivity in motor improvement, we analyzed the correlation between the change in effective connectivity within the affected sensorimotor cortex and the improvement in the FMA-UE score, irrespectively of the GROUP (BCI or sham) (Fig. 2e, f). Both changes in the μ and β bands showed a significant correlation (Pearson's correlation, μ: $r = 0.41$, $p = 0.045$; β: $r = 0.48$, $p = 0.02$).

**Brain-computer interface.** Selected discriminant EEG features for patients in the BCI group were localized in the ipsi- and contralesional sensorimotor areas, more prominently over the former, in frequency bands normally associated with voluntary movements—i.e., in the μ and β bands (Fig. 3a; Supplementary Fig. 1). Individual BCIs were built with the data of the offline

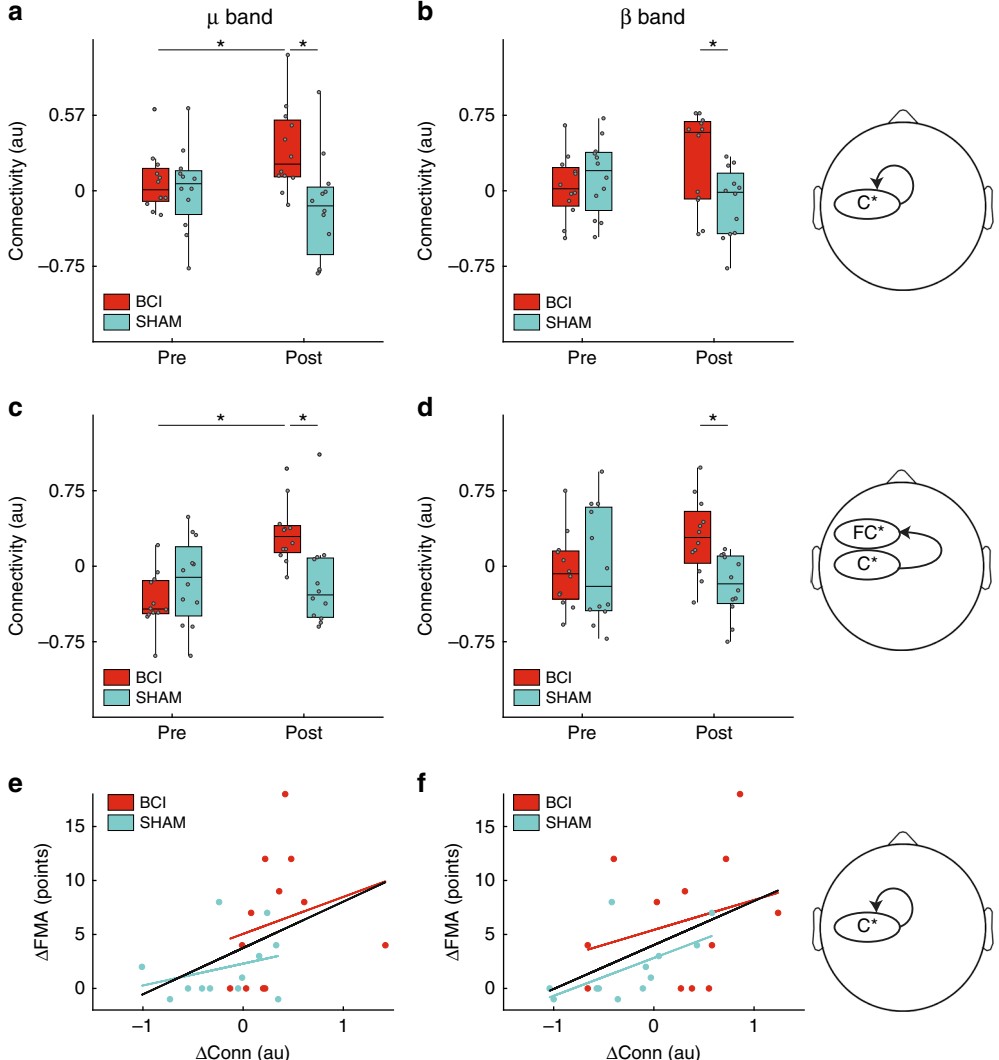

**Fig. 2** EEG effective connectivity within affected hemisphere during resting task. A mixed-design ANOVA revealed a significant increase of EEG effective connectivity after intervention in μ (10–12 Hz) (left column) and β (18–24 Hz) (right column) frequency bands for the BCI group ($N = 14$, red) as compared to the sham group ($N = 13$, light blue). Statistical differences are indicated (⁎ $p < 0.05$, ⁎⁎ $p < 0.001$, post-hoc Bonferroni-corrected two-tailed paired/ unpaired $t$-tests, see text for results on mixed ANOVA). **a**, **b** EEG effective connectivity changes within the affected sensorimotor cortex (channels C5*, C3*, and C1*), represented by boxplots (box: 25–75% percentiles, whiskers: 5–95% percentiles). Single values are also shown, jittered along the $x$-axis for a better visualization. **c**, **d** EEG effective connectivity changes from the affected C* to the FC* line. **e**, **f** Change of connectivity (post–pre intervention) within the affected sensorimotor cortex vs FMA post–pre intervention, together with the least-squares fit line for both groups pooled ($N = 24$, black line). Significant correlations were found in both μ and β frequency bands (Pearson's correlation, μ: $r = 0.41$, $p = 0.045$; β: $r = 0.48$, $p = 0.02$). Least-square fits for each group separately are also shown for representation purposes (colored lines; $N = 12$ for BCI and sham groups)

calibration session, aiming at having an expected false positive rate close to zero—i.e., the BCI should not elicit FES when the patient was not actually attempting a movement (average offline performance across BCI patients: true positive rate, $TPR = 94.6 \pm 3.4\%$; false positive rate, $FPR = 3.0 \pm 3.9\%$; no-decision, $ND = 24.2 \pm 20.7\%$; Fig. 3b, left; Supplementary Fig. 2).

We monitored the trial-based online performance (hit rate, average number of FES commands over the total number of 15 trials in a run) (Fig. 3b, center) and the time required by patients to deliver a BCI command (Fig. 3b, right). The slight decline of hit rate over time is explained by a tendency of the therapists to increase confidence threshold values so as to shape task difficulty and maintain high patient motivation. Supplementary Fig. 3 confirms it, as both the patients' simulated BCI performance with a fixed, conservative threshold, and the actual confidence threshold exhibited a slightly increasing trend.

We further explored the contingency between neural correlates of motor attempt (as detected by the BCI) and FES delivery, as well as the relationship of this contingency with motor improvement and effective connectivity changes. In the case of the sham patients, we built their individual BCI with data from the calibration session as for the BCI patients. Given the instruction to patients for a sustained motor attempt, its neural correlate is not confined to the attempted movement onset. Therefore, we only view as critical the decoding of the cortical pattern immediately preceding (as well as during) the FES, where the efferent and afferent volleys should coincide. We can thus compute the contingency table between motor decoding and FES (Supplementary Table 1), where the diagonal matrix elements represent the different types of contingency, and the non-diagonal elements quantify its absence. Table 3 reports standard contingency metrics derived from the contingency table

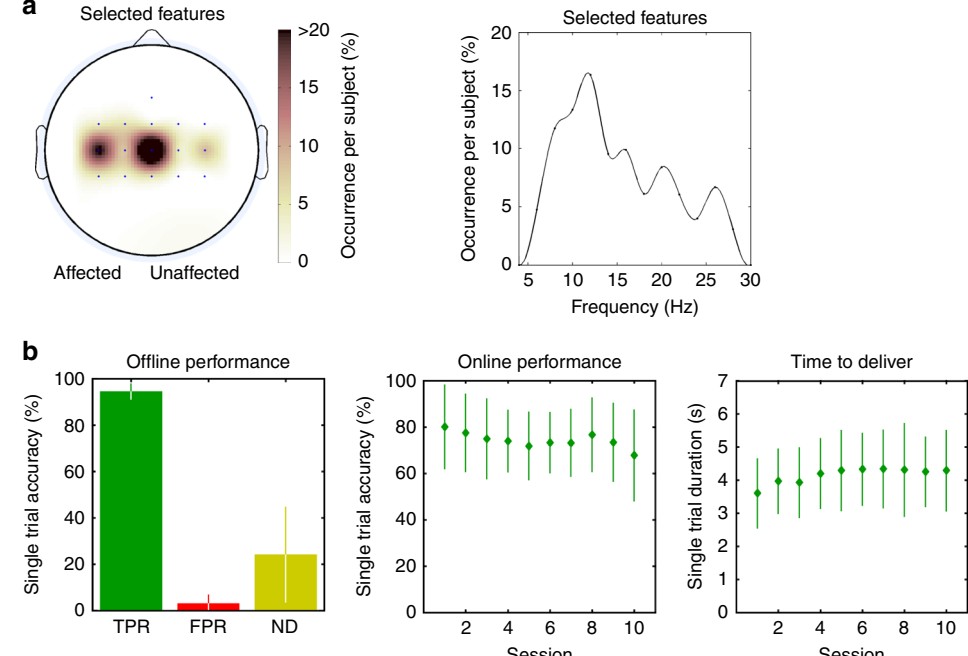

**Fig. 3** BCI features and performance. For all subjects of the BCI–FES group ($N = 14$): **a** Selected discriminant EEG features used for closed-loop control by their electrode location (the affected hemisphere is on the left side) and frequency distribution. **b** Left. Average offline single-trial performance estimated in the calibration session (±standard deviation): true positive rate (TPR), false positive rate (FPR), and no-decision (ND). Center. Average online single-trial classification performance for each session (±standard deviation). Right. Average time required by the BCI to detect a movement attempt in each session (±standard deviation)

corresponding to SMR detection at FES onset, which we hypothesize might promote functional activity-dependent plasticity. In particular, Accuracy results (see also Fig. 4), which combine all elements of the contingency table, show that: (i) it was significantly better for BCI patients than for sham patients (two-tailed unpaired $t$-test, $p < 10^{-10}$) (Fig. 4a left), (ii) it correlates with FMA improvement when all patients are taken together (Pearson's correlation, $r = 0.48$, $p = 0.012$) (Fig. 4a right), and (iii) it correlates with increases in effective connectivity within the affected sensorimotor cortex in the μ and β bands (Pearson's correlation, μ: $r = 0.49$, $p = 0.02$; β: $r = 0.55$, $p = 0.005$) (Fig. 4b). Interestingly, similar results also hold for other metrics in Table 3 (correlation with effective connectivity not shown for legibility). Additionally and critically, our experimental design successfully decoupled BCI output from FES in the sham group.

It is also worth noticing that the first-order metrics of contingency (elements of the contingency table that were directly and purposefully manipulated through our experimental and group designs) already exhibited the hypothesized correlation with FMA improvement (positive for TP, negative for FP and FN). In addition, analysis of all these aspects of contingency in 2 s long intervals around the FES onset shows that our experimental design successfully coupled SMR activity to FES in the BCI group, also well into the FES period (Supplementary Fig. 4). Sustained contingency remains a good predictor of recovery in this interval (Supplementary Fig. 5). These results suggest that contingency (i.e., co-occurrence) of SMR and FES is critical, although precise timing between FES and SMR onset is probably not a requirement.

Furthermore, we have built a multiple linear regressor to assess how these metrics can predict recovery ($\Delta$FMA = $w_1$TP + $w_2$TN + $w_3$FP + $w_4$FN). Evaluating this model with leave-one-out cross-validation showed that its explained variance amounted to $r^2 = 32\%$, 10% higher than that explained by classical

performance metrics such as accuracy described above ($r^2 = 22\%$). Linear regressors for predicting increases in effective connectivity within the affected sensorimotor cortex in the μ and β bands also confirmed that explanatory power of the first-order contingency metrics (μ: $r^2 = 38\%$; β: $r^2 = 44\%$).

## Discussion
BCI–FES therapy resulted in a statistically significant, clinically important, and lasting reduction of impairment in chronic moderate-to-severe stroke patients. In particular, the preservation of clinically relevant improvements at least 6 months after end of therapy is remarkable. Former studies either contained no follow-up evaluation[24,26,27] or revealed no long-lasting effects[25]. The parallel group experimental design demonstrates that direct BCI control is the key therapeutic factor underlying motor recovery, as only the BCI–FES group exhibited significant increases in FMA-UE scores after therapy. Importantly, benefits of the BCI–FES intervention extended beyond the hand and wrist, the target of FES, to other sections of the FMA-UE—i.e., sections 'synergies' and 'movement combining synergies'.

Regarding the secondary clinical outcomes, the BCI–FES group also exhibits a significant and lasting increase of the strength of the target muscle extensor digitorum communis, as measured by the MRC score. It is known that somatosensory input, in the form of peripheral nerve stimulation, increases strength of the innervated muscles by that nerve[32]. Nevertheless, this and other kinds of FES seem neither to have long-lasting effects[33] nor to be superior to standard care[34], which might explain why sham-FES patients did not exhibit such an increase even at the end of the therapy. In this respect, our work shows how a BCI turns a weak intervention, such as FES, into an effective one by enforcing contingency between BCI and FES. This finding opens the door to new combinatorial and personalized interventions where closed-loop decoding of brain activity plays a key role. On the other

**Table 3 Statistics of metrics of contingency between motor attempt decoding and FES and their correlation to motor recovery across all patients**

| Metric | Definition | Group statistics | | | Correlation with ΔFMA | |
|---|---|---|---|---|---|---|
| | | BCI | Sham | p-Val | r | p-Val |
| TP | | 72.36% ± 10.53% | 42.30% ± 11.52% | <10⁻⁶ | 0.38 | 0.051 |
| TN | | 12.52% ± 7.31% | 12.16% ± 6.93% | 0.89 | 0.11 | 0.60 |
| FP | | 0.00% ± 0.00% | 21.50% ± 11.33% | <10⁻⁶ | −0.29 | 0.15 |
| FN | | 14.00% ± 8.43% | 24.05% ± 6.70% | 0.002 | −0.50 | 0.009 |
| True positive rate (TPR)/recall/sensitivity | $\frac{TP}{(TP+FN)}$ | 83.76% ± 9.78% | 63.79% ± 1.01% | <10⁻⁶ | 0.51 | 0.007 |
| True negative rate (TNR)/specificity | $\frac{TN}{(TN+FP)}$ | 100.00% ± 0.00% | 34.83% ± 5.50% | <10⁻²⁴ | 0.47 | 0.014 |
| Positive predictive value (PPV)/precision | $\frac{TP}{TP+FP}$ | 100.00% ± 0.00% | 66.26% ± 17.88% | <10⁻⁶ | 0.29 | 0.148 |
| Negative predictive value (NPV) | $\frac{TN}{(TN+FN)}$ | 49.85% ± 23.75% | 33.46% ± 18.81% | 0.059 | 0.47 | 0.013 |
| Accuracy | $\frac{(TP+TN)}{(TP+TN+FP+FN)}$ | 85.95% ± 8.40% | 54.46% ± 4.85% | <10⁻¹⁰ | 0.48 | 0.012 |

Statistical significance of group differences extracted with two-tailed unpaired *t*-tests. Statistical significance of correlations to motor recovery extracted with Student's t distribution
ΔFMA: post–pre FMA scores

hand, no significant improvements were found in other stroke scales evaluating activities of daily living, like ESS and Barthel Index, or spasticity (Ashworth).

Concomitant functional imaging with EEG might pinpoint possible mechanisms underlying these changes, consisting in an increase of functional connectivity between motor areas in the affected hemisphere. Significantly, this phenomenon was correlated with improvements in FMA-UE scores (see further analyses in Supplementary Methods: Functional connectivity/FMA correlations and Supplementary Fig. 6). Reactivation of ipsilesional M1 seems to be crucial for motor recovery[35]. Also, the change in EEG effective connectivity between ipsilesional M1 and premotor areas observed in BCI patients is in line with previous studies that highlighted the strong relationship of this functional connectivity pattern with motor deficits and their improvement with therapy after stroke[36]. Furthermore, increases in μ and β interactions among ipsilesional brain areas after stroke have been demonstrated to be associated with better motor performance and recovery[37,38]. There is also evidence that recruitment of contralesional circuitries can influence recovery[39,40], although not all studies have found it[41,42]. Also, a recent study[19] showed that chronic stroke patients achieved a significant behavioral improvement (Action Research Arm Test) after controlling a rehabilitation robot with a BCI based on contralesional channels. Nevertheless, this study neither had a control group nor included follow-up evaluation. It is also worth noticing that we found electrophysiological changes not only present at the network level as measured by functional connectivity, but also associated with larger desynchronization within motor areas (see Supplementary Methods: Neural desynchronization and source localization and Supplementary Fig. 7). Even though the identified post-intervention changes in electrophysiology are not a proof of an underlying cortical reorganization, they are highly indicative of functional plasticity mechanisms accompanying the proposed intervention and potentially promoting motor recovery. In turn, functional plasticity has been shown to be associated with cortical reorganization[43]. Hence, while we can offer no direct evidence of structural plasticity effects undergoing the proposed therapy, we speculate that this would be a reasonable expectation in the light of our findings.

Our results suggest that decoding of spared voluntarily-modulated EEG sensorimotor rhythms, reinforced by activation of body natural efferent and afferent pathways through FES, may promote purposeful cortical reorganization in relevant cortical areas. This might have given rise to the strong and lasting recovery observed in the BCI–FES group. A plausible explanation for the clinical efficacy of our approach is its specific hypothesized impact on corticospinal tract (CST) projections. Stroke patients exhibit a pattern of motor improvement that follows two stereotypical paths[44,45]; they will either recover about 70% of their initial motor impairment or show little to no improvement. This seems to be largely independent of the lesion type, patient age, and the amount of therapy that is provided to the patients[46]. It turns that the group with poor proportional improvement is characterized by a greater damage to the CST[46,47]. Hence, we speculate that the BCI–FES intervention might have strengthened CST projections in chronic patients that, initially, did not follow this proportional recovery rule[44–46]. In particular, the synchronous activation of cerebral motor areas and of peripheral effectors might have induced Hebb-like plasticity[48] and strengthened CST projections[49]. Indeed, rich sensory inputs have been shown to drive brain plasticity[28]—in particular, in sensorimotor areas in the case of somatosensory stimulation. This effect might have been amplified by the closed-loop coupling of intention decoding with action execution. Unfortunately, we do not have imaging data to probe this hypothesis.

Our results put forward a speculative, but plausible and in line with literature, mechanistic interpretation of our BCI intervention as they show how the necessary time contingency between FES and motor decoding is satisfied only in the BCI group. Furthermore, the metrics characterizing this contingency aligned with the clinical improvement and positive changes in functional connectivity in the affected sensorimotor cortex. This suggests that the efficacy of the proposed therapy could indeed be explained in the framework of Hebbian plasticity. We anticipate that, encouraged by our findings, future work will substantiate this hypothesis with more suitable imaging techniques.

As rich afferent feedback is central in our hypothesis, the recruitment of muscle spindles and Golgi tendon organs via FES might have played a key role. Recent animal studies demonstrated the fundamental role of muscle spindles in locomotor recovery after spinal cord injury[50]. Muscle spindles might be equally essential for recovery after stroke. Other effective BCI-based interventions used a robot to move patient's affected hand[24,25,27] providing proprioceptive feedback via muscle spindles. However, FES depolarizes more motor and sensory axons, sending larger sensory volleys from muscle spindles and Golgi tendon organs into the CNS[51]. Maybe critical for the present work, the

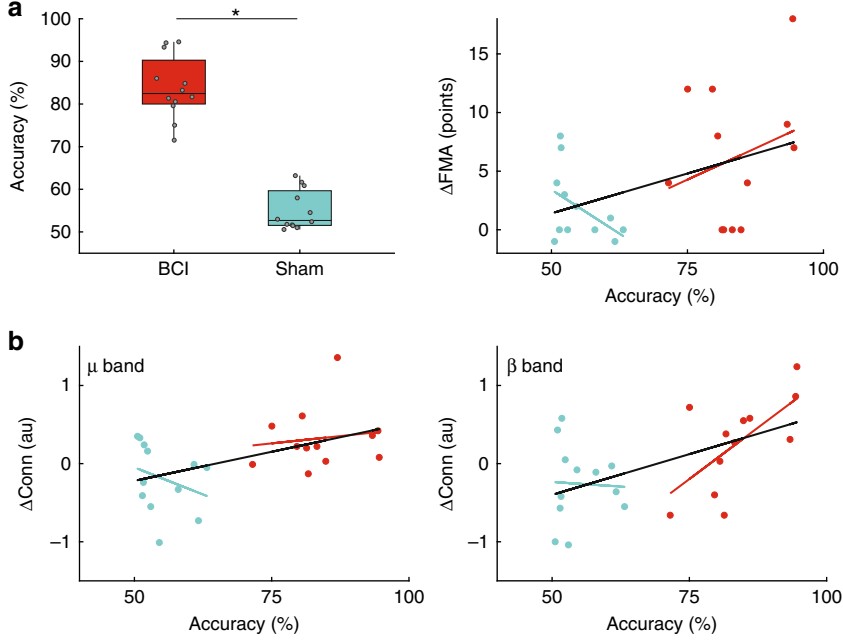

**Fig. 4** Accuracy of contingency between last PSD sample classification and FES. **a** Left. Values for the BCI ($N = 14$, red) and the sham ($N = 13$, light blue) groups (two-tailed unpaired $t$-test, $p < 10^{-10}$). Right. Accuracy vs $\Delta$FMA score (post–pre intervention), together with the least-square fit lines for both groups pooled (black line) and for each group separately (color lines). **b** Left. Accuracy vs $\Delta$Connectivity in μ band (post–pre intervention), together with the least-square fit lines for both groups pooled (black line, $N = 24$) and for each group separately (color lines). Right. Accuracy vs $\Delta$Connectivity in β band (post–pre intervention), together with the least-square fit lines for both groups pooled (black line). Correlations between these metrics were significant (Pearson's correlation, μ: $r = 0.49$, $p = 0.02$; β: $r = 0.55$, $p = 0.005$). Fits for each group separately are shown for representation purposes (colored lines; $N = 12$ for BCI and sham groups)

monosynaptic excitatory projections from spindles onto moto-neurons may activate them concurrently with the presumed descending cortical command, thereby causing Hebbian association. Furthermore, and perhaps key for our hypothesis that a BCI–FES intervention has specific effects on CST projections, FES also causes antidromic activation of motoneurons and Renshaw neurons in the spinal cord[52–54] thus reinforcing motor structures and its contingency with ascending sensory fibers. While we believe all these mechanisms are likely to co-occur and contribute to recovery, our data are not sufficient to disentangle the exact role of each individual mechanism or their combination.

Another component that distinguishes our BCI approach from previous attempts is that BCI confidence threshold was adjusted at each therapy session for each patient in order to make the acquisition of contingent feedback (i.e., delivery of FES) hard but feasible[55]. This property, in combination with the low false positives of the BCI, helped patients to be effectively engaged and attentive to the motor task. Active participation and close attention play an important role in promoting plasticity[56,57].

The combination of these three properties may explain why our BCI approach yielded larger motor improvements than pre-vious controlled trials with chronic stroke patients[24,25] and similar to those reported in another recent trial[27], where the chronic sub-population exhibited higher FMA-UE scores and much shorter time since stroke at baseline than those of our own study, and the two groups remain unbalanced in a number of possible confounds. The partial absence of the aforementioned properties may also be the reason why other BCI–FES studies did not find clinical gains[22].

Apart from single-patient case reports, only another study has coupled successfully BCI with FES for upper limb rehabilitation[23]. However, it had a small sample size (4 patients), did not have a control group, and did not include follow-up evaluation.

Furthermore, only one out of four patients exhibited a clinically significant improvement in the motor function department of the FMA-UE. This might be related to the fact that authors calibrated the whole BCI decoder (including feature selection) at each ses-sion in order to optimize BCI performance. Parsimonious adaptation of the BCI decoder might be beneficial by capturing and exploiting the evolution of functional plasticity during recovery. Nevertheless, very frequent recalibration might overall hinder plasticity since continuously changing BCI features sub-stantially modify the recruited efferent pathways participating in eventual activity-dependent plasticity mechanisms. The effects of timing and intensity of adaptation in BCI-based rehabilitation paradigms is a little-studied topic that warrants further investigation.

Other kinds of therapies for motor rehabilitation after stroke have yielded significant improvements in controlled trials with chronic patients, namely robotics[58] and constraint-induced movement therapy (CIMT)[59]. BCI–FES can be naturally com-bined with the former (see below) and be complementary to the latter. Indeed, although CIMT provides high functional recovery, only patients with a high degree of residual mobility in arm/hand function can benefit from it. This is the potential advantage of a BCI-based intervention, which only requires patients to attempt a movement. This was the case for two patients in the BCI group (BCI05 and BCI06) with complete hand paralysis and, in one case (BCI06), participating in the BCI–FES therapy 15 years after stroke, for whom a recovery of motor activity is exceptional. They regained voluntary muscular contraction resulting in wrist extension and signs of fingers extension. As a result, patients BCI05 and BCI06 recovered 7 and 8 FMA-UE points, respec-tively. On the contrary, none of the plegic sham-FES patients showed signs of recovery.

**Table 4 Patients' characteristics, including lesion etiology**

| Patient | Gender | Age (years) | Diagnosis | Lesion site | Lesion side | Time since stroke (month) |
|---|---|---|---|---|---|---|
| *BCI–FES group* | | | | | | |
| BCI01 | M | 64 | Ischemic | Subcortical | Right | 10 |
| BCI02 | M | 71 | Ischemic | Cortical | Right | 14 |
| BCI03 | M | 49 | Ischemic | Subcortical | Right | 10 |
| BCI04 | F | 50 | Ischemic | Cortical | Right | 19 |
| BCI05 | F | 49 | Ischemic | Cortical & subcortical | Left | 13 |
| BCI06 | F | 67 | Ischemic | Subcortical | Left | 176 |
| BCI07 | F | 41 | Ischemic | Subcortical | Left | 39 |
| BCI08 | M | 48 | Ischemic | Cortical & subcortical | Right | 14 |
| BCI09 | F | 52 | Ischemic | Subcortical | Right | 25 |
| BCI10 | F | 54 | Hemorrhagic | Cortical & subcortical | Left | 67 |
| BCI11 | M | 59 | Hemorrhagic | Cortical & subcortical | Left | 65 |
| BCI12 | F | 57 | Ischemic | Cortical & subcortical | Left | 79 |
| BCI13 | M | 76 | Ischemic | Cortical & subcortical | Left | 16 |
| BCI14 | F | 52 | Ischemic | Cortical & subcortical | Right | 10 |
| | | | | | | |
| *Sham-FES group* | | | | | | |
| sham01 | M | 40 | Hemorrhagic | Cortical & subcortical | Right | 18 |
| sham02 | M | 58 | Ischemic | Cortical & subcortical | Right | 23 |
| sham03 | M | 75 | Hemorrhagic | Subcortical | Left | 15 |
| sham04 | M | 53 | Ischemic | Subcortical | Left | 21 |
| sham05 | M | 65 | Hemorrhagic | Subcortical | Left | 38 |
| sham06 | M | 57 | Ischemic | Cortical & subcortical | Left | 62 |
| sham07 | F | 62 | Ischemic | Cortical & subcortical | Left | 121 |
| sham08 | M | 63 | Ischemic | Subcortical | Right | 12 |
| sham09 | M | 36 | Ischemic | Cortical & subcortical | Right | 16 |
| sham10 | F | 49 | Ischemic | Subcortical | Left | 48 |
| sham11 | F | 76 | Ischemic | Cortical & subcortical | Right | 15 |
| sham12 | M | 73 | Ischemic | Cortical & subcortical | Right | 35 |
| sham13 | M | 60 | Hemorrhagic | Subcortical | Right | 11 |

Despite the promising results, our BCI–FES study suffers from a number of limitations that will need to be overcome in future clinical trials. The most critical one is that, despite the trial has been designed as double-blind, there is a possibility that certain therapists with previous experience with our BCI in other studies[60,61] could have inferred the group to which some patients were allocated by making educated guesses given the observed BCI feedback. This fact may have inflated the effect size. Nevertheless, this possibility is unlikely as, in this case, FMA-UE improvements would hardly correlate with other indexes (functional connectivity and contingency accuracy).

A related issue is that we did not check for any placebo effect that could have influenced patients to engage differently depending on the allocated group. However, all patients used the same setup and received the same instructions. Moreover, since patients were asked to perform as many runs of 15 movement attempts (trials) as they could (3–8), the fact that both groups did the same number of runs per session (BCI = $6.0 \pm 0.7$; sham = $5.9 \pm 0.7$) indicates that all subjects exhibited a similar high level of engagement and motivation. Also, single-sample detection rate was almost identical for the BCI and sham groups (BCI: $63.8\% \pm 17.1\%$; sham: $63.7\% \pm 18.1\%$; $p = 0.99$, Wilcoxon rank-sum test). Besides, this detection rate did not correlate with FMA improvement (Pearson's correlation, $r = -0.33$, $p = 0.1$). Thus, BCI aptitude, as well as patients' motivation/engagement that should be reflected in BCI aptitude, cannot explain either the clinical outcomes.

A third limitation, although not critical, concerns the limited accuracy and repeatability of hand movements generated by FES. A hand orthosis provides more controlled kinematics[62] and, thus, it would be interesting to explore the combination of hand orthosis and FES so as to evaluate whether a BCI that operates both together yields larger motor improvements.

There are also some potential confounding factors that may have affected our analysis. The first one is that enrolled patients, being chronic, may have had different latent motor control. Patients did not undergo any kind of pre-study rehabilitation to determine their latent capacity and account for it in the allocation to the two groups. However, we screened their historical data to ensure stability in motor recovery. Also, the size of the population should have balanced this effect across groups. In fact, both groups exhibited similar standard deviations in FMA-UE and MRC values at all clinical evaluations (pre, post, and follow-up). A second confounding factor is that we did not control or ask whether patients practiced with their affected arm/hand outside therapy. Still, it is reasonable to expect that, if patients practiced, it is because the therapy had some beneficial effect that they did not experience before. The third confounding factor relates to the distribution of FES among subjects. That is, although both groups were supposed to receive a similar number of FES, the BCI group happened to get a slightly larger amount of FES per run than the sham group (BCI = $11.0 \pm 1.5$; sham = $9.9 \pm 0.4$). Nevertheless, this difference was not statistically significant (Fisher's exact test, $p = 0.12$). Furthermore, taking all patients together, there was no statistical correlation between the number of FES and the increase in FMA-UE (Pearson's correlation, $r = 0.26$, $p = 0.20$). FES parameters were also excluded as confounding factors (see Supplementary Methods: Functional electrical stimulation). A fourth confounding factor might arise from the individual lesion characteristics. Yet, neither the volume, nor the center of gravity, nor the distribution (individual voxels) of the lesions differed significantly between the two groups (Supplementary Fig. 8).

In this study, we have not been able to explain why four BCI patients did not improve and two sham patients did improve above 5 FMA-UE points. None of the previously reported analyses explained these exceptions. No demographic or clinical factor (including lesions, see Supplementary Fig. 8) could shed light on this matter. This issue could have been addressed by means of additional, higher-resolution neuroimaging techniques like fMRI, diffusion tensor imaging, or motor evoked potentials, which unfortunately we did not have at our disposal.

Finally, we delivered FES on the extensor digitorum communis muscle only. Our choice was motivated by the fact that hand mobility is the most critical for daily living. Nevertheless, simultaneous stimulation of different muscle sets to accomplish more complex hand gestures, and arm postures, is worth investigating in future clinical trials, especially in combination with rehabilitation robots.

## Methods

**Subjects**. Between 18 September 2012, and 4 August 2015, 377 individuals were screened for eligibility, of whom 27 were included (BCI–FES = 14; sham-FES = 13) (Supplementary Fig. 9), following previous studies with a similar sample size[24,26]. Twenty-one participants were recruited at SUVACare - Clinique Romande de Réadaptation, Sion and six at the University Hospital of Geneva. The cantonal ethical committees (Commission Cantonale Valaisanne d'Éthique Médicale and Commission d'Éthique de la Eecherche sur l'Être Humain du Canton de Genève) approved the study protocol and each participant gave written informed consent prior to their eligibility assessment. No adverse events related to the study occurred. The clinical trial registration number is EudraCT Number: 2017-000755-97. At the time of this study, prospective clinical trial registration was not mandatory for nonpharmacological studies; it was therefore registered retrospectively.

The main inclusion criteria were first ever cerebrovascular accident resulting in chronic impairment (minimum 10 months from stroke) and moderate-to-severe disability (severe hand paralysis with a FMA-UE score ≤ 40 points). Further inclusion criteria were minimum age of 18 years, good or corrected eyesight, and unilateral cortical lesion (left or right hemisphere) and/or subcortical lesion. All participants were able to comprehend simple instructions and did not have cognitive deficits that could prevent them to undertake the rehabilitation task, as tested during a neuropsychological screening using the Raven test that assesses non-verbal reasoning and problem-solving[63] (see Deltour[64] for a French standardization).

Other exclusion criteria were factors hindering EEG acquisition (e.g., skin infection, wound in the scalp, dermatitis, uncontrolled muscle activity, metal implants under electrodes), heavy medication affecting the central nervous system (including vigilance), concomitant serious illness (e.g., fever, infection, and metabolic disorders), unilateral spatial neglect, severe dystonia/involuntary movements, other neurological disorders (e.g., Parkinson's disease), severe or recent heart disease, and active implants (e.g., pumps, pacemaker, phrenic pacemakers, and pain stimulators). In contrast to other studies, low BCI performance was not an exclusion criterion[25]. Figure 1a, Supplementary Fig. 8, and Table 4 provide information about the patients and their lesions.

All patients were in their plateau phase of recovery, as determined by comparing their clinical scores (primary and secondary outcomes) measured during the eligibility assessment to equivalent data from previous routine controls. They received conventional arm physical therapy—sessions of 45 min comprising mobilization and activities of daily living—in addition to BCI–FES or sham-FES in order to filter out potential effects due to non-use and atrophy. After concluding the intervention, the participants resumed their normal life where they did not receive any particular therapy.

The participants were enrolled sequentially and assigned to either BCI–FES or sham-FES therapy. Four patients were initially enrolled to the BCI group. Relevant metrics concerning brain control were extracted from these first four subjects and used to simulate brain control in the sham group (i.e., distribution of detected commands and time to deliver a brain command). This procedure has been used in animal studies on BCI-based rehabilitation[65], and was essential to build a comparable sham intervention where the only therapeutic factor missing was direct brain control. Subsequent allocation of patients to the two groups was done to balance scores of the most important covariates (i.e., baseline FMA-UE, age, gender, time since stroke, affected hemisphere, type of lesion, lesion location, and number of patients per group). Lesion size and location were determined either through magnetic resonance imaging or computerized tomography scans. The trial was double-blind, as neither the patients nor the evaluators or the therapists supervising the therapeutic sessions were aware of the group allocation. The hardware equipment used during the therapy by both groups was identical, and software tools were developed so as to hide whether they would implement actual or sham brain control.

**Primary and secondary outcome metrics**. The primary clinical outcome metric of the study was the change in FMA-UE (from 0 to 66, plegic to normal). Secondary outcomes included the Medical Research Council score for strength of the FES target muscle extensor digitorum communis (MRC: from 0 to 5, no force to normal), the modified Ashworth scale for spasticity of the wrist extensor and flexor (MAS: from 0 to 5, normal to rigidity), and the European stroke scale score for general disability (ESS: from 0 to 100, worst to normal). Clinical evaluations were performed immediately before and after the intervention, as well as 6–12 months after the end of the intervention (average 36 weeks).

**Statistical analyses of outcome metrics**. For the statistical comparison of categorical variables, we used Fisher's exact test, which is robust in the case of small sample sizes. In the case of continuous variables, Lilliefors corrected Kolmogorov–Smirnov tests were first applied to check the normality of the variables. For Gaussian variables, a mixed two-way ANOVA with one within-subjects factor (TIME: pre, post, or, for the FMA-UE only, TIME: pre, post, and follow-up) and one between-subjects factor (GROUP: BCI, sham) was performed. Sphericity of the data was assumed as assessed by a Mauchly test. Post-hoc tests were performed with two-tailed paired or unpaired $t$-tests, depending on the factor, and Bonferroni corrected. For non-Gaussian variables, a non-parametric test (Wilcoxon rank-sum test) was applied on the difference post–pre intervention between the two groups. Additionally, a Wilcoxon rank-sum test was applied to ensure no significant differences between the pre-intervention values of the two groups (BCI and sham). If significance was found in the difference post–pre intervention, separate analyses were done with Wilcoxon rank-sum or Wilcoxon signed-rank tests, depending on the factor, and Bonferroni corrected.

Correlations between variables were measured using Pearson's correlation coefficient. Associated p-values from the correlations were estimated using $t$-distributions with $N$-2 degrees of freedom.

**Brain-computer interface**. All patients, BCI and sham, performed an initial session where their individual BCI classifier was calibrated to differentiate brain activity corresponding to either hand-extension attempt or resting. The same classifier was used throughout the therapy. EEG was recorded at a sampling frequency of 512 Hz with 16 active surface electrodes placed over the sensorimotor cortex—i.e., on positions Fz, FC3, FC1, FCz, FC2, FC4, C3, C1, Cz, C2, C4, CP3, CP1, CPz, CP2, and CP4 of the 10/20 system (reference: right mastoid; ground: AFz; g·tec gUSBamp, Guger Technologies OG, Graz, Austria). This calibration session was not included in the therapy time.

During the calibration session, the patients were asked to attempt to extend the affected hand (fingers and wrist) or to rest, in random order. A trial (movement attempt or rest) started with the "Preparation" cue (a cross in the middle of the screen) during 3 s, then a "Start" cue appeared for 1 s indicating the type of trial, followed by 4 s of movement attempt or resting, and finished with the appearance of the "Stop" cue during 2 s. Inter-trial intervals lasted from 3 to 4.5 s. The calibration sessions consisted of 3 runs of 15 trials each.

Each of the EEG channels was spatially filtered with a Laplacian derivation, whereby the weighted sum of the voltages of orthogonal neighboring channels is subtracted from that channel[66]. Then, the PSD of each spatially filtered EEG channel was estimated during the active period (i.e., movement attempt or resting following the "Start" cue) for the frequency bands [4 40] Hz with 2 Hz resolution over the last second. PSDs were computed every 62.5 ms, using the Welch method with five internal Hanning windows of 500 ms (75% overlap).

EEG data were analyzed offline and the most discriminant EEG spatio-spectral features between resting and movement attempts were selected to build the BCI classifier through machine learning techniques[67,68]. Specifically, a canonical variate analysis-based method[67] was employed to rank all candidate PSD features in terms of discriminant power. A maximum 10 of the candidate features were then manually selected taking into account both discriminancy and prior neurophysiological knowledge (task-relevant frequency bands and channel locations). Selected discriminant EEG features were fed to a Gaussian classifier that yielded the probability distribution of a PSD sample to belong to either class "movement attempt" or "rest". The BCI integrated these probabilities over time to accumulate evidence in favor of each class (see Supplementary Methods: BCI and Supplementary Fig. 10). When one of the probabilities reached a predefined confidence threshold, the BCI sent the corresponding command and the trial ended. Otherwise, if not enough evidence was accumulated in favor of one of the classes at the end of the active period, which we refer to as a trial time-out (fixed to 7 s), then the trial was considered "no-decision".

Importantly, our analysis shows that BCI performance was not affected by artifacts (Supplementary Methods: Investigation on influence of artifacts) so that motor decoding inferences were actually corresponding to physiological motor EEG correlates.

**BCI- and sham-FES therapies**. Both groups received therapy two times per week for a period of 5 weeks, directly in the centers (10 sessions in total). We rescheduled missed sessions, and therapy did not exceed 6 weeks for any patient. Each session lasted ~60 min, including preparation and device setup. We used the same 16-channel EEG system as for the calibration session.

During each therapy session, the patients were asked to perform three to eight runs of 15 hand-extension movement attempts (trials). They were encouraged to do as many runs as possible. Therapists started each trial (i.e., movement attempt) through a key press whenever they considered the patient to be ready. The protocol of a trial (Supplementary Fig. 10) was very similar to that of the calibration session, except for the duration of the active period that lasted at most 7 s. The BCI integrated the class probabilities, which were visualized as a cursor moving up (movement attempt) or down (resting) in a screen. Whenever the cursor reached a predefined confidence threshold for the class movement attempt, the BCI delivered a command that activated FES and the trial finished. If no "movement attempt" was detected in the 7 s after "Start" cue, the trial was terminated without FES (trial time-out/miss). Two patients in the BCI group required longer time-outs to be able to deliver BCI commands (15 s).

Cursor movement (output of the BCI) was visible only to the therapist, so that patients could solely focus their attention on the paretic hand. The therapist monitored the cursor and gave verbal instructions to patients on how to perform the task avoiding both compensatory strategies and generation of artifacts that could contaminate the EEG (eye and facial movements). The instructions dictated a single, sustained and slow attempt of a full (palm and fingers) paretic hand extension that should take place within the 7 s of the corresponding trial. In case of residual abilities, overt hand movements were allowed (but not compensatory elbow, shoulder or trunk movements). During the trial duration, the participants were explicitly instructed to avoid blinking and shifting their attention off their hand, as well as required to abstain from any head and eye movements, including speaking. Moreover, the participants were briefed to sustain the motor attempt during the FES stimulation until the full hand extension was achieved.

The confidence threshold that initiated FES was adjusted after each run for each patient by the therapist, so as to shape the task difficulty in a way that it was hard but feasible[55], targeting an average online performance of 10–12 FES, out of 15, per run.

For the sham-FES therapy, patients received identical instructions and wore identical equipment to the BCI group. Data from the first four patients in the BCI group were used to estimate the behavior of the FES dynamics for the sham group, which had a command delivery of 9–11 FES per run and a FES delivery time in the range 3.5–5.5 s.

FES was performed through a commercial system (Krauth & Timmermann MotionStim8, Hamburg, Germany), with a single bipolar channel applied on the affected limb in order to inject current (having a pulsed, square waveform) into the extensor digitorum communis muscle. Electrical stimulation parameters such as current amplitude (ranging between 10 and 25 mA) and stimulation frequency (ranging from 16 to 30 Hz), as well as electrode placements were configured at each session by the therapist in order to elicit the desired hand-extension movement in a comfortable way for the patient. The stimulation train consisted of a 1 s-long ramp where the pulse-width was increased from 10 μs to the maximum 500 μs, followed by another 1 s of constant stimulation at maximum pulse-width.

**EEG markers of neuroplasticity via effective connectivity**. To investigate neuroplastic effects, all patients recorded a pre- and post-intervention high-density EEG session, where they performed 45 attempts of affected hand extension or resting, in random order, using the same protocol as for the calibration session. We mainly analyzed network properties, as measured by functional connectivity changes (directed EEG effective connectivity[69], see below) at rest.

Sixty-four EEG channels covering the whole scalp were recorded with a Biosemi ActiveTwo system (BioSemi B.V., Amsterdam, Netherlands) at a sampling frequency of 2048 Hz. EEG were first bandpass filtered using a 4th order non-causal Butterworth filter between 1 and 50 Hz, downsampled to 512 Hz, and common-average referenced. In order to analyze and illustrate data in a uniform manner across patients, EEG channels were flipped for patients with a lesion in the right hemisphere so as to localize the lesion over the left hemisphere for all subjects —i.e., electrode C3* covers the lesional hemisphere and electrode C4* the unaffected hemisphere. Thus, the two hemispheric motor areas correspond to the information from electrodes [FC5*, FC3*, FC1*, C1*, C3*, C5*, CP5*, CP3*, and CP1*] (affected hemisphere) and [FC6*, FC4*, FC2*, C2*, C4*, C6*, CP6*, CP4*, and CP2*] (unaffected hemisphere).

Due to technical problems or EEG artifactual contamination, we could only use data from 24 patients out of 27 (BCI = 12; sham = 12). For all electrophysiological analyses, following literature on EEG motor correlates, we concentrated on the frequency bands μ (10–12 Hz) and β (18–24 Hz).

Noisy electrodes were manually detected and removed from the following computations. Data were used to compute effective brain connectivity among scalp locations through the short-time direct directed transfer function (SdDTF), a type of functional connectivity preserving the directionality of information[70]. This method is a modification of the directed transfer function (DTF) using multiple trials to increase the temporal resolution and adopting partial coherence to avoid indirect cascade influences, where some region may influence another only through a third brain area[69].

For each recording session (pre- or post-intervention) and subject, all resting trials were used to compute the SdDTF between all pairs of electrodes in the two motor areas defined above for each patient group. SdDTF was calculated with a sliding window of 1 s overlapping 800 ms in order to obtain smooth modulations.

SdDTF results were then referenced to the baseline level, estimated from 3.5 to 1 s before the "Start" cue, and averaged in the time window [0 2.5] s after "Start" for different frequency bands.

For the analysis, we identified three regions of interest for each hemisphere: fronto-central (FC line of electrodes), central (C line), and centro-parietal (CP line). We then computed the effective connectivity between and within regions for each hemisphere, and between equivalent regions of both hemispheres, by averaging across the pairs of electrodes involved.

**Code availability**. Data analyses were conducted in Matlab using scripts available from the corresponding author upon reasonable request.

**Data availability**. The data that support the findings of this study are available from the corresponding author upon reasonable request.

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

## Acknowledgements

We are grateful to the patients, and their families, who participated in this study. We warmly thank Aurélie Bouzerda-Wahlen for neuropsychological evaluation of the patients, and the therapists for their dedication: Vanessa Buhlmann, Hédi Dimassi, Nathalie Pattaroni (SUVACare), and Cécile Magnin (University Hospital of Geneva). R. L. and S.P. also thank their current employer, MindMaze SA, Switzerland, for granting them time to work on the revision of the manuscript. This work was partially funded by the EU-ICT FP7-224631 project TOBI (A.A.K. and J.d.R.M.), Swiss NCCR Robotics (J.d. R.M.), the Swiss canton of Valais (R.L., T.S., M.B. and D.V.), and the Wyss Center for Bio and Neuroengineering in Geneva (S.P. and J.d.R.M.).

## Author contributions

A.B., R.L., and J.d.R.M. designed and coordinated the study. A.A.K., A.S., T.S., P.V., and A.G. enrolled patients. A.B., R.L., A.A.K., T.C., T.S., M.B., D.V., and A.G. collected data. A.B., R.L., I.I., S.P., T.C., T.S., H.Z., M.B., D.V., and J.d.R.M. analyzed the data. A.B., R.L., A.G., and J.d.R.M. wrote the first draft of the manuscript. A.B., R.L., I.I., S.P. and J.d.R.M. revised the manuscript. All authors approved the manuscript.

## Additional information

**Competing interests:** A.B. has formed a company that is developing a neuromuscular stimulation device and therapy for stroke patients. This device and therapy are not related to the work described in this paper. The remaining authors declare no competing interests.

