## [Peer Review File · Nature Communications]

Reviewers' comments:

Reviewer #1 (Remarks to the Author):

Generally, this was an interesting study that offered both a clinical and mechanistic analysis of BMI based stroke rehabilitation. As cited by the authors, there have been several studies that have examined combining BMI with robotic and other passive movement interventions for stroke rehab, but few studies have aimed to examine the benefits of BMI combined with peripheral electrical stimulation. The work is very similar to other studies that have looked at robotic controlled therapy instead of NMES, but NMES is the novel component of the present study. Unfortunately the present study did not directly compare their results to state-of-the-art BCI+robot therapy, which also has shown to have functional therapeutic benefit. The statistical analysis of the present study is solid, and the work is robust and could likely be reproduced in an independent lab.

I have some concerns that the study design does not match the stated hypothesis. The hypothesis states that it is the added value of the rich sensory inputs as feedback that gives rise to the boosting of activity-dependent plasticity. If that is the fundamental hypothesis, then it seems that the study should have examined one group that was receiving sensory feedback via NMES and one group that is NOT receiving sensory feedback. The sham group in the study still received the same sensory feedback, even if it was not directly tied to the BCI command. As designed, the study really is testing whether or not plasticity and functional recovery arises from sensory feedback via NMES alone, or as a result of tying NMES to the volitional command derived through BMI. These are two different questions. The authors either need to reformulate their hypothesis, or redesign their experiment to test their stated hypothesis and include a group performing BCI alone (with no passive movements) and another group with a BCI and exoskeleton (where the sensory input received would be different from that of NMES).

Please see attached document for additional specific comments.

Title

Brain-Actuated Neuromuscular Electrical Stimulation Elicits Lasting Arm Motor Recovery After Stroke

Summary

The study investigates the combined use of non-invasive (EEG) brain machine interface with neuromuscular electrical stimulation (NMES) to restore motor function to persons with chronic stroke. The study hypothesizes that BMI-NMES will enhance functional recovery because it is “tailored to reorganize the targeted neural circuits by providing rich sensory inputs via the natural afferent pathways so as to activate all spare components of the central nervous system involved in motor control...[and]...elicits functional movement and conveys proprioceptive and somatosensory information, in particular via massive and timely recruitment of muscle spindle feedback circuits”. The study compares the functional recovery and cortical plasticity of two groups: group 1 received BMI-triggered NMES, while another received sham NMES. The investigation shows statistically significant improvement of the BMI group and not the sham group based upon clinical functional metrics (Fugl-Meyer Upper Extremity, Muscle Strength measures, etc...). The BMI group also showed enhanced activity in mu and beta bands that were not present in the sham group. The investigators conclude that this study offers a novel use of BMI and NMES for stroke recovery in addition to elucidating the underlying mechanisms of this recovery.

General Comments (Major)

Generally, this was an interesting study that offered both a clinical and mechanistic analysis of BMI based stroke rehabilitation. As cited by the authors, there have been several studies that have examined combining BMI with robotic and other passive movement interventions for stroke rehab, but few studies have aimed to examine the benefits of BMI combined with peripheral electrical stimulation. The work is very similar to other studies that have looked at robotic controlled therapy instead of NMES, but NMES is the novel component of the present study. Unfortunately the present study did not directly compare their results to state-of-the-art BCI+robot therapy, which also has shown to have functional therapeutic benefit. The statistical analysis of the present study is solid, and the work is robust and could likely be reproduced in an independent lab.

I have some concerns that the study design does not match the stated hypothesis. The hypothesis states that it is the added value of the rich sensory inputs as feedback that gives rise to the boosting of activity-dependent plasticity. If that is the fundamental hypothesis, then it seems that the study should have examined one group that was receiving sensory feedback via NMES and one group that is NOT receiving sensory feedback. The sham group in the study still received the same sensory feedback, even if it was not directly tied to the BCI command. As designed, the study really is testing whether or not plasticity and functional recovery arises from sensory feedback via NMES alone, or as a result of tying NMES to the volitional command derived through BMI. These are two different questions. The authors either need to reformulate their hypothesis, or redesign their experiment to test their stated hypothesis and include a group performing BCI alone (with no passive movements) and another group with a BCI and exoskeleton (where the sensory input received would be different from that of NMES).

Specific Comments

Abstract

- “We hypothesize...efferent and afferent pathways” is an incomplete sentence. Possibly missing a word?

Introduction

- “*Synergistic efforts...body function*”: Missing a number or prominent references, including Hochberg 2012, Collinger 2013, Bouton 2016, Ajiboye 2017
- “*Nevertheless, the therapeutic role of BCI-based therapies is yet unclear*”: Not sure that this sentence fits given that you just spent a paragraph stating all the studies that have shown clinical benefits of BCI based training

- “*We hypothesize that...involved in motor control*”: If the hypothesis is essentially that NMES is better because it provides sensory feedback during BCI training, than shouldn't the study focus directly on comparing BCI driven NMES vs BCI driven exoskeleton control, or some other passive movement control that does not provide sensory feedback? The current setup of the study answers the question of whether or not BCI driven movements provide additional therapy than NMES driven rehab alone. Besides, doesn't passive movement of the affected limb also provide sensory feedback via stretch reflexes, Golgi tendon organs, etc... I don't think your hypothesis is really about the providing of sensory feedback, as much as stating that neuroplasticity is greater during active BCI control vs passive movement.

Materials and Methods

- “*All patients...data from previous routine controls*”: It is unclear what constitutes this data from previous controls.
- “*They received conventional arm physical therapy...*”: How do you determine the role that the conventional therapy played in affecting your results? Is it that BCI+NMES alone was the beneficial combination, or did BCI+NMES simply enhance plasticity that was already occurring as a result of this other confounding training?
- “*Each of the EEG channels was spatially filtered...*”: It is doubtful that Laplacian spatial filtering alone is enough to remove all the typical artifacts from EEG (specifically ocular, scalp EMG). Scalp EMG is known to dwarf EEG signals, and many advanced algorithms have been used to ensure that EEG is not contaminated by scalp EEG. And yet as far as I can tell, you do not employ any of those algorithms in this study. How are you thus sure that your movement EEG data does not just incorporate scalp EMG activity (for example) during a movement trial, and no scalp EMG during a non-movement trial? This is a classic EEG difficulty that I am surprised the investigators did not naturally discuss.
- “*...machine learning techniques...*”: need more info on machine learning approach. Cross-validation approach? How were the optimal features chosen?
- “*verbal instruction*”: Please be more specific. The verbal instructions can have an effect on what strategy the participants used for the BMI.
- “*...could contaminate the EEG (eye and facial movements)*”: Just giving the participants Instruction to minimize these artifacts is not enough. How in your data did you ensure that these artifacts were not present?

Figure 1

- How was the confidence threshold determined?
- I would like to see a correlation analysis between the decided BMI movement and the NMES in the sham group. We do not know anything about what BMI signals the sham group provided and whether or not they did/did not fall into the confidence threshold. What would it mean for your results if the sham group was actually not providing BMI commands that exceeded the threshold?

Figure 2

- The lack of decline between 6-36 weeks is of note and needs to be further discussed. What were participants receiving during this time? Their standard rehab (passive movements)?

Figure 3

- It is surprising to claim activity-dependent plasticity given that there is on average 4s between initiation of the cortical activity and actuation of the NMES. Presumably you would expect that activity dependent plasticity would require a much smaller timescale between cortical input and associated movement. Can you provide evidence of other literature where you see plasticity occurring with such a large latency?

Reviewer #2 (Remarks to the Author):

See attached PDF.

Review Biasiucci et al “Brain-Actuated Neuromuscular Electrical Stimulation Elicits Lasting Arm Motor Recovery After Stroke”

Main Review Questions:

WHAT ARE THE MAJOR CLAIMS OF THE PAPER?

The authors show that muscle stimulation, triggered based on movement intent detected from the EEG (EEG-FES), induces greater functional recovery than muscle stimulation alone. The authors also study the neural correlates of this clinical finding by doing functional connectivity and “rhythm” analysis on EEG data.

ARE THE CLAIMS NOVEL? IF NOT, PLEASE IDENTIFY THE MAJOR PAPERS THAT COMPROMISE NOVELTY

To my best knowledge, no previous paper has tested the clinical effects of EEG-FES in a controlled experiment with as many stroke patients as in this study. The investigation of neural correlates is also novel, although I am concerned by several aspects of it (details in coming sections). However, one study that came out earlier this year presented a very similar intervention (EEG-FES to improve arm and hand function), and showed comparable clinical improvements after 1 month of treatment:

Ibáñez, J., Monge-Pereira, E., Molina-Rueda, et al. Low Latency Estimation of Motor Intentions to Assist Reaching Movements along Multiple Sessions in Chronic Stroke Patients: A Feasibility Study. *Front Neurosci* 11, 126, 2017

This paper had other similarities, such as the authors also retrained the classifier every day. In contrast, they did not have a control group and the intervention group only had 4 patients, which the authors did not follow up. However, the methods were very similar.

There is at least another conceptually similar study, although it focuses on rehabilitation of lower limb function. Mrachaz-Kersting and colleagues used EEG to trigger a common peroneal nerve stimulus during a specific phase of the movement-related cortical potential. Using this approach, they were able to improve walking function in stroke patients. This study had a control group and was similarly powered to the current paper, although they only performed 3 sessions and did not do a follow-up assessment:

Mrachacz-Kersting, N., Jiang, N., Stevenson, et al. Efficient neuroplasticity induction in chronic stroke patients by an associative brain-computer interface. *J. Neurophysiol.* 2015

WILL THE PAPER BE OF INTEREST TO OTHERS IN THE FIELD?

Neurorehabilitation has been undergoing a technological revolution for several years now. This study tests a (relatively) novel intervention that is based on emergent ideas in the field (e.g., associating the motor intent and assisted movement, or inducing Hebbian plasticity), and does so in a relatively large group and with controls. As such it is likely to be of interest to the field.

WILL THE PAPER INFLUENCE THINKING IN THE FIELD?

As mentioned above, this study provides clinical evidence that supports quite convincingly emerging ideas in the field. I am more concerned about the impact of the authors' investigation of underlying mechanisms, as the authors did not report a thorough single patient analysis (only a correlation of one of the features with the gross clinical improvement).

ARE THE CLAIMS CONVINCING? IF NOT, WHAT FURTHER EVIDENCE IS NEEDED?

There is a clear group effect in terms of clinical improvement. However, a single-patient analysis of most aspects of the study is missing. Most critically, the authors only report group neural correlates, namely changes in functional connectivity and synchronization/desynchronization of cortical rhythms. The findings of the later also seem rather vague, and the analysis should be made in a more detailed fashion (see some possible ideas below).

I am also concerned about the lack of comments about 4 (of 14) patients in the intervention group that did not improve clinically. The same about 4 patients in the control group that improved, two of them in a extent similar to the intervention group.

ARE THERE OTHER EXPERIMENTS THAT WOULD STRENGTHEN THE PAPER FURTHER? HOW MUCH WOULD THEY IMPROVE IT, AND HOW DIFFICULT ARE THEY LIKELY TO BE?

There are a few experiments the authors could do, but that seem rather unrealistic at this point. One would be measuring corticospinal excitability using TMS to generate motor evoked potentials and test more directly the extent of corticospinal changes. The other would be recording functional MRI, rather than EEG and do single patient analysis of hand/digit representations (as in Ejaz et al Nature Neurosci 2015) as well as more precise connectivity measurements.

ARE THE CLAIMS APPROPRIATELY DISCUSSED IN THE CONTEXT OF PREVIOUS LITERATURE?

The most important paper that the authors do not discuss and that I am aware of is Ibáñez et al. 2017.

There are also at least a couple of reviews that address the use of neurostimulation triggered or controlled by detected movement intent that the authors could benefit from discussing:

Ethier, C., Gallego, J., Miller, L., 2015. Brain-controlled neuromuscular stimulation to drive neural plasticity and functional recovery. *Curr. Opin. Neurobiol.* 33, 95–102
Jackson, A., Zimmermann, J.B., 2012. Neural interfaces for the brain and spinal cord--restoring motor function. *Nat. Rev. Neurol.* 8, 690–9

Secondary review questions:

IS THE MANUSCRIPT CLEARLY WRITTEN? IF NOT, HOW COULD IT BE MADE MORE ACCESSIBLE?

Yes, writing is quite good. Figure captions should have larger font size.

COULD THE MANUSCRIPT BE SHORTENED TO AID COMMUNICATION OF THE MOST IMPORTANT FINDINGS?

I do not think so. In fact the paper would benefit from following each main result by a short interpretation. The methods could be moved to the end.

HAVE THE AUTHORS DONE THEMSELVES JUSTICE WITHOUT OVERSELLING THEIR CLAIMS?

Yes, for the most part. I am only concerned about:

- 1) some claims that are being made based on two correlations that are barely significant ($p = 0.048$ and $p = 0.046$), and for which the data do not seem strikingly convincing (see “Is the statistical analysis of the data sound?”)
- 2) the authors also claim that they were the first group to re-adjust the BCI every session to facilitate ease of use. However, at least Ibáñez et al. have done it before
- 3) I also miss that the authors do not discuss that the intervention patients improved in the gold standard clinical test for stroke (FMA-UE), but not according another scale of upper limb function (ESS, European stroke scale). Neither did they comment on the lack of significant improvements in spasticity (Ashworth scale). They also did not comment on the four BCI patients that did not improve, or the 4 control patients that did improve.

HAVE THEY BEEN FAIR IN THEIR TREATMENT OF PREVIOUS LITERATURE?

See comment above, overall regarding Ibáñez et al., 2017

HAVE THEY PROVIDED SUFFICIENT METHODOLOGICAL DETAIL THAT THE EXPERIMENTS COULD BE REPRODUCED?

I would welcome additional details about how they trained the classifier, and on some of the functional connectivity and synchronization/desynchronization analysis methods. The latter could go as Supplementary information.

IS THE STATISTICAL ANALYSIS OF THE DATA SOUND?

For the most part, yes. I am only concerned about:

1) the analysis in Fig. 4 (bottom): The authors compute the linear correlation (using Pearson) between connectivity changes and clinical improvements. They find that these variables are significantly correlated, which they are, but the P-values are $p = 0.048$, and $p = 0.046$ respectively. Looking at their data and given their small sample size, I think they should be more cautious when interpreting this result. However, they use this result several times to back up other claims.

2) in the sLoreta analysis, the authors pooled all the intervention and control patients together and then computed their post-intervention differences. They then report results for the group differences using a quite generous significance threshold ($P < 0.05$). They should report the exact value. Most importantly, also show single-patient and group pre vs. post differences.

3) Given the large standard deviation in Fig 3b and 4, the authors should report single-patient results in addition to error bars. I presume that examination of within-patient data may also illuminate within group differences

SHOULD THE AUTHORS BE ASKED TO PROVIDE FURTHER DATA OR METHODOLOGICAL INFORMATION TO HELP OTHERS REPLICATE THEIR WORK? (SUCH DATA MIGHT INCLUDE SOURCE CODE FOR MODELLING STUDIES, DETAILED PROTOCOLS OR MATHEMATICAL DERIVATIONS).

Nothing is critical, although Supplementary Material providing details on the Classifier and the neurophysiological analyses could be helpful. The authors do provide references, though.

ARE THERE ANY SPECIAL ETHICAL CONCERNS ARISING FROM THE USE OF ANIMALS OR HUMAN SUBJECTS?

No.

Other comments:

- Use of FES vs. NMES. I think the authors should rather use FES as they do use muscle stimulation to generate a functional movement. This distinction is important, for example, for the Discussion, where some studies they mention used stimulation mostly to activate sensory pathways (e.g., Conforto et al.), which is different to what the authors did.

- I do not think the authors should call the BCI "brain control" as they only used detected movement intent to trigger the stimulation; brain-triggered?

- There are several acronyms that are not described: K-S, sdDTR, MNI standard brain, SMA, CIMT, etc.

ABSTRACT:

- Explain more clearly that association of neural activity and generated movement is key.
- Last line: “correlated with the primary outcome” -> “functional improvement”

INTRODUCTION:

- A forward step: a step forward
- “in particular via massive and timely recruitment of muscle spindle feedback circuits” -> Clarify
- “several studies suggest that NMES increases cortical excitability” -> the authors only cite one

METHODS:

- Was the lesion size identified using imaging or clinically? Describe
- Provide brief details about the methods for selecting the classifier features. I don't think canonical variate analysis (the method used by Leeb et al.) is a “state-of-the-art” machine learning technique; it's rather established at this point.
- Did the authors leave the classifier constant across sessions and only change the threshold?
- What would be the maximum number of NMES trials per run in the case of perfect performance?
- EEG markers of neuroplasticity: the authors may be measuring changes in functional connectivity that are not mediated by changes in underlying circuitry. Since these are very difficult to differentiate, I'd suggest using another word such as functional connectivity.
- “Indirect cascade influences” -> explain
- How was the threshold adjusted? based on a few calibration trials at the beginning of the session?

RESULTS:

- Why do the authors think that online performance seems to overall decrease over time? Functional connectivity changes or changes in cortical rhythms that make the classifier become obsolete? A within-subject cross-session analysis would be an informative Suppl. Fig. to understand if this is a group effect or an spurious result of inter-subject differences. The authors could also compare BCI performance offline with the online threshold as well as with an optimally-selected threshold. This would be another interesting Suppl. Fig.
- An interesting Supplementary analysis would be to test BCI performance offline leaving out specific channels, to try and understand the contribution of that specific area. For example, the authors comment on the Discussion that contra-lesional channels may also support recovery. Would their classifier performance decrease drastically without them?
- How long was the rest period the authors used for inferring cortical connectivity? Did they just append inter-trial periods? (“BCI trial where the patient is asked to rest”). Similarly, did the authors use the movement-related part of the trial for the sLoreta analysis? I could not find this information in the paper.
- Clarify the following statement: “and also between the two groups after the intervention in favor of the BCI group in the μ (two-tailed unpaired t-test, $p = 0.003$) and β bands (two-tailed unpaired t-test, $p = 0.035$)”

- “Both changes in connectivity in the μ and β bands showed a significant correlation (Pearson’s correlation, μ : $r = 0.41$, $p = 0.048$; β : $r = 0.41$, $p = 0.046$).” As mentioned above, I realize the changes are significant (only very slightly), but by looking at the raw data, it seems to me that the distribution of FMA-UE is quite scattered and the fits quite unreliable (e.g., the positive slope in μ for the BCI group seems to be largely driven by the patient with a connectivity change of 1.5). Please comment on this.

- I am surprised about the sLoreta results. First, the main change happens in Cz, not around C3* (leg area vs. hand/wrist area). Second, the change in beta seems greater in the unaffected as opposed to the affected hemisphere, and again close to the midline. I also have difficulties reconciling these results with the functional connectivity results. The groups were not different in terms of ERD/ERS before the intervention but ERD/ERS power increased post-intervention, for the BCI group only, in both hemispheres. At the same time, functional connectivity only changed for the affected hemisphere and for the BCI group. How do the authors combine these observations?

DISCUSSION:

- “Clinically important” -> “significant”

- “to other sections of the FMA-UE — i.e., sections ‘synergies’ and ‘movement combining synergies’” -> The authors should describe this more clearly in the Results: they do talk about the hand/wrist vs. upper limb items in general, but do not emphasize this important observation.

- The authors state that “BCI turns a weak intervention, such as NMES, into an effective one” -> I think the key is the association between motor intent and evoked movement. In my view this is a new intervention, rather than a combination of two.

- “consisting in an increase of neural desynchronization over the affected hemisphere “ -> In the results, the authors say that this increase is bilateral.

- While I agree with the following statement: “In particular, the change in EEG connectivity between ipsilesional M1 and premotor areas observed in BCI patients is in line with previous studies that highlighted the strong relationship of this connectivity pattern with motor deficits and their improvement with therapy after stroke (Wu et al., 2015).” it is interesting that Guggenmos and colleagues targeted premotor and sensory cortex and were able to induce clear recovery after a motor cortical stroke. Please discuss.

- “brain nodes “ -> brain areas?

- “purposeful cortical reorganization in relevant cortical areas” -> I agree that there seem to be group changes in ERD/ERS and connectivity but these do not prove cortical reorganization, in the sense of synaptogenesis, neurogenesis or LTP/LTD of existing synapses. It could just be changes in neural activity patterns generated with exactly the same structure.

- The authors should elaborate better on the role of CST projections. Their involvement seems likely, but the reasoning is not entirely clear to me.

- “NMES recruits more muscle spindles and activates them faster, conveying also somatosensory information. “ -> Please Clarify. NMES also activates Golgi tendon organs when the muscle is contracted. Moreover, spindle stimulation experiments normally use higher stimulation frequencies. Critically for the present work, the monosynaptic excitatory projections from spindles onto motoneurons may activate them concurrently with the presumed descending cortical command, thereby causing Hebbian association.

- “Remarkably, the distribution of significantly different voxels where neural desynchronization emerged post-intervention and that were correlated with increase of FMA-UE scores (Fig. 5) matched the distribution of selected features for the BCI decoders, both spectrally and spatially (Fig. 3a).” -> These are the affected areas, which are motor, so it’s not surprising that they are included in the decoders. What would happen if they associated activity from non-motor areas? would the intervention still work? This would be interesting to discuss
- “Combination of these three properties may explain why our BCI approach yields relatively larger motor improvements than previous controlled trials with chronic stroke patients (Ramos-Murguialday et al., 2013; Ang et al., 2014). “- > Ibáñez et al got comparable clinical improvements although in a convenience sample of only 4 subjects. Comment
- “Nevertheless, the correlation over all patients of 16 FMA-UE improvements with changes in both types of EEG indexes (cortical connectivity and neural desynchronization) makes this possibility unlikely” -> The correlation coefficient for at least one of those was barely significant. Rephrase

Reviewer #3 (Remarks to the Author):

Dr. Millan and colleagues present an intriguing paper on the use of EEG recording and neuromuscular electrical stimulation to aid in the motor recovery of subjects with arm paresis from stroke. Towards this end, they employ a BCI-NMES therapy that uses changes in EEG signal to drive stimulation of a hand extensor. They perform a number of controls to rule out inter-subject variability and EEG artifact, and suggest that individuals that had undergone BCI-NMES therapy demonstrated hand function improvement compared to control. If their findings are true, this approach may provide an important advancement in the treatment of individuals with moderate-to-severe paralysis from stroke and plausibly other disorders.

While I am fairly enthusiastic about the prospective use of such an approach, there are a number of fundamental limitations in the study design and interpretation. Overall, I do not believe that this paper would be suitable for publication in Nature Communications, at least in its present form.

MAJOR CONCERNS

Foremost, the study is highly limited in its focus on hand extension. Monotonic improvements such as this have been previously shown with BCIs in both humans and primates. However, the major limitation of focusing on only a single muscle group is that they do not clearly allow for improvement in more than one functional movement direction. For example, it would not be easily possible to train individuals to perform both flexion AND extension movements under the current paradigm which would be essential for meaningful use.

Second, while the authors provided a control group, they did not perform a switch-over control to confirm that improvement in performance was not simply due to a difference between rather than within patients. Even subtle differences in stroke size and impingement on the motor cortex can have dramatic differences in the likelihood of recovery over 6-12 months. While the authors do a commendable job in attempting to match the two groups, this is not sufficient to demonstrate that improvement is causally attributable to the BCI.

Third, and in line with the comments above, it is not clear that there was any blinding performed. There is also an unusual procedure where the therapist gave verbal instructions to the patients to avoid eye or other movements that could 'contaminate' the EEG signal. Yet, it is not apparent whether the EEG signals were not affected and in whom.

Fourth, the use of an ANOVA is a non-standard way of looking for change in connectivity. While the authors provide a few references to its use, it would be helpful to confirm that similar findings could be obtained by more standard coherence and/or time-series analyses such as Granger causality. On that same note, did the authors examine coherence with the opposite hemisphere? There has been broad evidence that recruitment of contralateral circuitries such as the SMA can significantly influence recovery.

Lastly, there has been extensive literature on the use of biofeedback and physical therapy for improving motor performance in individuals with stroke. It would be important to demonstrate that the proposed improvement provided through the BCI-NMES system was not simply due to simple visual/sensory feedback.

MINOR COMMENTS

The references and manuscript structure do not appear to conform to the Nature Communication

format.

For patient comparison, it would be helpful to provide figures that display the precise stroke locations, sizes and distributions across the two patient groups.

Given the overlap, it would be important to include references to work such as that of Eberhard Fetz and/or Ali Razai.

Reviewer #4 (Remarks to the Author):

Biasiucci and colleagues performed an important and exciting study in the field of therapeutic neuroprosthetics. They hypothesized that BCI-controlled neuromuscular stimulation could drive a positive reorganization of the CNS and promote recovery after stroke. They tested an NMES system to activate the finger extensors of the affected arm, either in synchrony with the decoded intent to open the hand, or randomly as a sham-NMES comparison. The subjects were in chronic phase of a stroke, at least 10 months post-infarct. They assessed changes in hand function as measured by the Fugl-Meyer Assessment as a primary outcome, as well as the EDC muscle strength and European stroke scale for general disability as secondary outcome, immediately, six months, and 12 months after the intervention. In addition, the authors evaluated the EEG connectivity and neuronal desynchronization in both groups before and after treatment to obtain indication of neuroplasticity.

All the patients received the same instructions, which were to attempt hand extension continuously during the active period of the trial, which lasted up to 7s (or 15 seconds for 2 BCI patients). For the BCI group, NMES was delivered upon detection of movement attempt, and for the sham group NMES was delivered 3.5 to 5.5 seconds after the "start" cue. This delay corresponded to the average time it took for the BCI classifier to reach the confidence threshold.

This design allowed the team to evaluate whether NMES could induce beneficial plastic and functional changes when applied by itself, or guided by the BCI detection of motor intent. Both groups wore the same hardware during interventions, and importantly, neither patients, therapists nor evaluators were informed of group allocation.

The authors find that patients in the BCI group improved their wrist and hand extension function to a clinically relevant level compared to the sham group. Also, they find indications of greater neuroplastic changes for the BCI group.

This study is definitely novel and important the positive results are encouraging for clinician and patients, and intriguing for the research community. However I do have a few concerns, that I hope will help improve the potential impact of this study.

I have three major concerns related to the efficacy of the BCI system, the design of the experiment, and the details of the NMES :

1) In the field of Brain-Computer Interfaces, classifying between "active" and "rest" states is probably the easiest possible task. In figure 3, the authors report what seems like an average ~70%-80% single trial accuracy. I'm interpreting this to mean that the BCI failed to detect the active state once every 4 or 5 trials, even though patients were actively trying to extend their fingers for 7 seconds. When the BCI successfully detected a movement attempt, it actually needed a delay of 3 to 5 seconds

from the start of the neural activity related to motor intent.

In my opinion this poor temporal resolution is an important confounding factor, which make the distinction between the BCI-NMES and sham-NMES intervention unclear. The authors' claim that, in the BCI case, the stimulation is "driven by neural activity". However, for the sham NMES intervention, the delivery of NMES also occurred in conjunction with the patients' attempting hand extension. In both cases, peripheral stimulation is delivered concomitantly with neural activity related to voluntary effort. In the BCI case, the exact timing of the NMES onset was related to the algorithm confidence and an arbitrary threshold.

Therefore, the authors' interpretation that "The synchronous activation of cerebral motor areas and of peripheral effectors might have induced Hebb-like plasticity and strengthened CST projection" in the BCI group but not in the sham group is difficult to accept, as the conjunction of cerebral activity and peripheral stimulation is similar in both groups.

However, the difference between the groups may lie in the intensity of the voluntary effort. In the discussion, the authors clarify that the BCI task was made difficult by adjusting parameters every session. "Active participation and close attention play an important role in promoting plasticity". Thus it's possible that the BCI patients were more actively engaged than the sham group. If the modulation of voluntary effort was the determining factor for the therapeutic effects, it would be important to emphasize that.

-> Is this what Figure 5 suggests? It's not clear to me. My understanding is that the data for this figure were obtained not during BCI control, but in a separate recording, similar to the calibration task, while patients responded to instructions to rest or attempt movement, with no NMES delivered. While this figure is compelling in that it shows improved neuronal activation for the BCI group, it does not show a difference in neuronal modulation during the task. Can the authors produce analyses of EEG activity recorded during the therapy session, in order to understand whether the BCI group produced a greater voluntary effort?

2) The authors point out that : "There is a possibility that evaluators could have known the group to which some patients were allocated. This fact may have inflated the effect size." This is a very odd observation. If the authors know, because of a comment from evaluators or patients for example, that there are cases when evaluators did know the group to which patients belong, then it should be clearly and plainly admitted. If the authors are just speculating and never had any indication that this could be the case, then this comment should be removed from the manuscript. Now I'm assuming that the reason why this comment is there, is that the authors actually know that the evaluators where not entirely blind to group assignment, and for scientific rigor it is important to disclose that.

3) The description of the NMES is lacking details. The electrical stimulation is the main component of the intervention, and further description is critical. Examples of important missing information: What was the duration of the stimulus train? How were the current and frequency chosen by the therapist? What were their mean values, and what did it represent in terms of % of motor threshold? The authors report that the number of NMES per run was slightly, though not statistically significantly, higher for the BCI groups. What about other NMES parameters? If the details of this intervention varied between patients, it is important that these parameters be also included in the statistical tests performed between the BCI and the sham groups, to confirm that the NMES intervention itself was not different between groups and did not explain changes in outcomes.

Minor points:

Figure 2b illustrate the main outcome of the intervention. From the methods' description, the statistical tests seem appropriate and well-designed. However, from the figure, we can see a considerable overlap in the standard deviation error bars between conditions, even though the authors report a statistically significant difference.

spell out SdDTF on first occurrence.

define sLoreta and MNI standard brain on first occurrences.

Please explain why for the connectivity analysis the β band was restricted to 18-24Hz, while 13-30Hz were analyzed for desynchronization analysis. I'm confused as to why they used these specific bands. These values don't match the frequency bands given by sLoreta presented in the methods (low: 12-18, mid: 18-21 high: 24-30).

"Two patients in the BCI group required longer time-outs to be able to deliver BCI commands (15 s)."
-> It's not clear what the "time-outs" refer to. By definition I would assume that it's the time between trials, but from the context it sounds like it might refer to the movement execution phase (active period) of the trial. Please clarify and do not use the term "time out" unless referring to pauses between trials.

Reviewer #1

Comment 1.1: I have some concerns that the study design does not match the stated hypothesis. The hypothesis states that it is the added value of the rich sensory inputs as feedback that gives rise to the boosting of activity-dependent plasticity. If that is the fundamental hypothesis, then it seems that the study should have examined one group that was receiving sensory feedback via FES and one group that is NOT receiving sensory feedback. The sham group in the study still received the same sensory feedback, even if it was not directly tied to the BCI command. As designed, the study really is testing whether or not plasticity and functional recovery arises from sensory feedback via FES alone, or as a result of tying FES to the volitional command derived through BMI. These are two different questions. The authors either need to reformulate their hypothesis, or redesign their experiment to test their stated hypothesis and include a group performing BCI alone (with no passive movements) and another group with a BCI and exoskeleton (where the sensory input received would be different from that of FES).

We thank the reviewer for allowing us to clarify this essential point. The hypothesis states that it is the **contingency** between rich sensory feedback and suitable motor-related cortical activity that drives activity-dependent plasticity. Our approach is designed to deliver associated contingent feedback that is not only functionally meaningful (e.g., via virtual reality or passive movement of the paretic limb by a robot), but also tailored to reorganize the targeted neural circuits by providing rich sensory inputs via the natural afferent pathways.

As stated by the reviewer “tying FES to the volitional command derived through BMI” is the most crucial prerequisite under this hypothesis. It is for this reason that the sham intervention was designed to maintain provision of FES feedback and to “remove” its contingency to the appearance of the desired brain pattern. Of note, the contribution of a BCI in attaining **clinically significant** motor rehabilitation effects is to date debatable, and our study is the first to demonstrate this possibility in a chronic population (please see reply to Comment 1.14). Furthermore, to the best of our knowledge, ours is also the first study to show that patients retain functional improvements in a 9-month follow-up evaluation. As an additional major novelty, we have chosen FES as the source of rich sensory feedback, which we believe should also be largely credited with obtaining clinically significant outcomes. It is true that our study would have greatly benefited from a direct comparison with other (devoid of FES) BCI-based therapies, in order to strictly assess and quantify the contribution of FES-based sensory feedback, as suggested by the reviewer. However, this was judged impossible due to logistical constraints: if we would have added a BCI-robot group, then we should have also included a sham-robot group. Nevertheless, we can rely on previous studies in order to evaluate the efficacy and important role of FES.

Comment 1.2: “We hypothesize...efferent and afferent pathways” is an incomplete sentence. Possibly missing a word?

The syntactic error in this sentence has been corrected. It now reads “*We hypothesize that a BCI-controlled functional electrical stimulation (FES) therapy can drive purposeful cortical reorganization thanks to contingent activation of body natural efferent and afferent pathways.*”.

Comment 1.3: “Synergistic efforts...body function”: Missing a number of prominent references, including Hochberg 2012, Collinger 2013, Bouton 2016, Ajiboye 2017

Thank you for pointing out at this excess of synthesis. Originally, we opted to cite only one prominent example of the recent high-impact publications on assistive BCI scenarios in order to keep our reference list as concise as possible, also considering that our work only concerns the BCI-based rehabilitation scenario. Nevertheless, we agree with the reviewer that more citations would provide a better-defined background for our study, so we have added the references in the revised manuscript.

Comment 1.4: “Nevertheless, the therapeutic role of BCI-based therapies is yet unclear”: Not sure that this sentence fits given that you just spent a paragraph stating all the studies that have shown clinical benefits of BCI based training

In spite of promising results achieved so far by the studies in question, BCI-based stroke rehabilitation is still a young field where different works report variable clinical outcomes. In addition to that, the

number and size of concluded and ongoing clinical trials by no means suffice to safely conclude the existence of clinically significant improvements of those therapies, while the mechanisms of recovery remain unclear. We have slightly modified this sentence to better reflect and specify this reasoning: *“In spite of promising results achieved so far by the studies in question, BCI-based stroke rehabilitation is still a young field where different works report variable clinical outcomes. Furthermore, the efficacy and mechanisms of BCI-based therapies remain largely unclear”*.

Comment 1.5: “We hypothesize that...involved in motor control”: If the hypothesis is essentially that FES is better because it provides sensory feedback during BCI training, than shouldn't the study focus directly on comparing BCI driven FES vs BCI driven exoskeleton control, or some other passive movement control that does not provide sensory feedback? The current setup of the study answers the question of whether or not BCI driven movements provide additional therapy than FES driven rehab alone. Besides, doesn't passive movement of the affected limb also provide sensory feedback via stretch reflexes, Golgi tendon organs, etc... I don't think your hypothesis is really about the providing of sensory feedback, as much as stating that neuroplasticity is greater during active BCI control vs passive movement.

Please refer to our response to Comment 1.1.

Comment 1.6: “All patients...data from previous routine controls”: It is unclear what constitutes this data from previous controls.

These tests aimed at establishing that recruited subjects are in their plateau phase of motor recovery. Therefore, their data correspond to the same clinical outcomes defined as the primary (FMA-UE) and secondary (MRC, ESS, etc) outcomes in our study in order to quantify the effect of the proposed treatment. These data were derived from assessments performed well before (several weeks, months, or even years) the launch of our trial as part of the standard rehabilitation program. This sentence has been slightly modified to clarify this issue: *“All patients were in their plateau phase of recovery, as determined by comparing their clinical scores (primary and secondary outcomes) measured during the eligibility assessment to the equivalent data from previous routine controls.”*

Comment 1.7: “They received conventional arm physical therapy...”: How do you determine the role that the conventional therapy played in affecting your results? Is it that BCI+FES alone was the beneficial combination, or did BCI+FES simply enhance plasticity that was already occurring as a result of this other confounding training?

Please, note that because all patients on both groups underwent conventional arm physical therapy, any eventual effect of this training should on average equally affect both groups. Furthermore, since patients were in their plateau phase of motor recovery, we expected any contribution of conventional arm physical therapy to be minimal.

Comment 1.8: “Each of the EEG channels was spatially filtered...”: It is doubtful that Laplacian spatial filtering alone is enough to remove all the typical artifacts from EEG (specifically ocular, scalp EMG). Scalp EMG is known to dwarf EEG signals, and many advanced algorithms have been used to ensure that EEG is not contaminated by scalp EEG. And yet as far as I can tell, you do not employ any of those algorithms in this study. How are you thus sure that your movement EEG data does not just incorporate scalp EMG activity (for example) during a movement trial, and no scalp EMG during a non-movement trial? This is a classic EEG difficulty that I am surprised the investigators did not naturally discuss.

We apologize for not having addressed this issue in the original manuscript. The Laplacian spatial filtering targeted the isolation and emergence of localized SMR patterns from neighboring ensemble EEG activity rather than coping with artifacts. Our pilot sessions revealed no issues with EEG signal contamination. This should be attributed to the instructions given to the patients and, mainly, to the constant supervision by the therapist. Data collected in each therapeutic session were also immediately inspected by an EEG engineer. Limiting the selected features to the neurophysiologically relevant mu and beta bands and channels (not too frontal, or too occipital) has further diminished any effect of artifacts on BCI performance. Last but not least, despite great progress in online artifact removal methods, these approaches are still imperfect and, most importantly, require subject- and session-

specific customizations necessitating the presence of expert engineers. They are thus barely compliant with the stringent logistic constraints imposed by the clinics (as well as real-world BCI applications in general).

We have now performed an a posteriori quantification of eventual impact of artifacts, showing that any facial EMG contamination was minimal and that the BCI was driven by the intended SMR features and not by artifacts. First, we prove that any influence of facial EMG on the raw EEG signal during motor attempt was minimal. Specifically, for each therapeutic run, the statistics (average, standard deviation) of the artifact-free EEG signal of all channels is estimated after low-pass filtering with cutoff at 20 Hz, given that facial EMG are known to occur above this frequency (Goncharova *et al.*, 2003; Van Boxtel, 2010). Subsequently, we measure the percentage of time that the original, non-filtered (and, potentially, artifact-contaminated) signal is above a certain z-score threshold computed on the artifact-free signal. Values above a z-score of 3.0 are conventionally thought to represent abnormally high amplitudes, indicative of EMG artifacts. Results show that during motor-attempt only 3.03% of EEG samples had an absolute z-score above threshold.

Second, in order to quantify how our BCI was driven by the desired SMR features and not by task-specific artifacts, we show that all 14 patients in the BCI group exhibited mu and/or beta sensorimotor rhythms as control signals, what suggests BCIs did not rely on artifacts. This was derived from the average (across all runs/sessions) feature discriminability between the PSD samples of motor attempt and rest (newly added **Supplementary Figure S2**, also **Fig. 1 below**).

Third, we verify that the BCI accuracy of subjects in the BCI group highly correlates with the discriminancy (average Fisher score) of the selected SMR features (Pearson's correlation, $r=0.5714$, $p=0.0019$). This is not true for the average discriminancy of spectral features of the Fz channel (Pearson's correlation, $r=0.2423$, $p=0.2234$), which is the channel most susceptible to facial and ocular artifacts.

We have added these analyses in the Supplementary Material.

Goncharova II, McFarland DJ, Vaughan TM, Wolpaw JR. EMG contamination of EEG: spectral and topographical characteristics. *Clin Neurophysiol* 2003; **114**: 1580–1593.

Van Boxtel, A. Facial EMG as a tool for inferring affective states. In *Measuring Behaviour Conf* 2010; 104–108.

Figure 1: Sensorimotor rhythm modulation in online therapeutic sessions. The affected hemisphere is on the left.

Comment 1.9: “...machine learning techniques...”: need more info on machine learning approach. Cross-validation approach? How were the optimal features chosen?

In the interest of space, section “Brain-Computer Interface” of the Materials and Methods in the original manuscript provides a brief, but thorough, summary of all the machine learning methods employed, while citations (Galán *et al.*, 2007; Leeb *et al.*, 2013) give full details.

We have purposefully not employed cross-validation. On the one hand, during the training phase of our Gaussian classifier we employed batch, iterative gradient-descent, as discussed in (Leeb *et al.*, 2013). There, we opted for a 50/50 training/testing set split instead of cross-validation to reduce the training time. The estimated parameters of the gradient-descent iteration exhibiting the highest classification accuracy on the testing set were selected to build the final classifier using all the data in the calibration session, which was then employed throughout the therapy. On the other hand, all BCI performance results reported in the manuscript reflect the actual “online” accuracy the subjects experienced during the therapy with the originally trained classification model, so that the use of cross-validation is inapplicable in this case. We have added these details in the Supplementary Materials.

Details and motivations on the feature selection procedure are offered in the discussion of the original manuscript. Reference (Galán *et al.*, 2007) offers the technical details. In our revised manuscript, a brief technical summary of this procedure has been added (text in italics): “...through machine learning techniques (Leeb *et al.*, 2013). Specifically, a Canonical Variate Analysis (CVA)-based method (Galán *et al.*, 2007) is employed to rank all candidate PSD features in terms of discriminant power. Maximum 10 of the candidate features were then manually selected taking into account both discriminancy and prior neurophysiological knowledge (task-relevant frequency bands and channel locations). Selected discriminant EEG features...”

Comment 1.10: “verbal instruction”: Please be more specific. The verbal instructions can have an effect on what strategy the participants used for the BMI.

Thank you for bringing this into our attention. We have extended this paragraph to provide further details on the provided instructions (text in italics): “Cursor movement (output of the BCI) was visible only to the therapist, so that patients could solely focus their attention on the paretic hand. The therapist monitored the cursor and gave verbal instructions to patients on how to perform the task avoiding both compensatory strategies and generation of artifacts that could contaminate the EEG (eye and facial movements). *The instructions dictated a single, sustained and slow attempt of a full (palm and fingers) paretic hand extension that should take place within the 7 seconds of the corresponding trial. In case of residual abilities, overt hand movements were allowed (but not compensatory elbow, shoulder or trunk movements). During the trial duration, participants were explicitly instructed to avoid blinking and shifting their attention off their hand, as well as required to abstain from any head and eye movements, including speaking.*”

Comment 1.11: “...could contaminate the EEG (eye and facial movements).”: Just giving the participants Instruction to minimize these artifacts is not enough. How in your data did you ensure that these artifacts were not present?

Please, refer to our reply to Comment 1.8. Furthermore, we would like to underline that we have not relied only on the instructions given to the patients, but, mainly, on the constant supervision and correction by the therapists.

Comment 1.12: How was the confidence threshold determined?

The relative extract of the original manuscript reads “*The BCI confidence threshold that triggered FES was adjusted at each therapy run for each patient so as to shape the task difficulty in a way that it was hard but feasible (Taub *et al.*, 1994), targeting an online performance of 10-12 FES, out of 15, per run.*”. The important formerly missing detail is that this adjustment was done manually by the therapist, who was constantly monitoring the patient performance and was aware of the history of threshold values used so far for each patient.

Comment 1.13: I would like to see a correlation analysis between the decoded BMI movement and the FES in the sham group. We do not know anything about what BMI signals the sham group provided and whether or not they did/did not fall into the confidence threshold. What would it mean for your results if the sham group was actually not providing BMI commands that exceeded the threshold?

In the original manuscript, we have avoided presenting BCI accuracy metrics for the sham group. This is because, in the sham group, there was no closed-loop BCI driving the FES (no online classification, evidence accumulation and adjustable thresholding to trigger stimulation). Therefore, comparisons with the BCI performances of participants in the BCI group are somewhat ill-posed.

However, we agree with the reviewer that we still need to a) examine the “real” BCI aptitude (i.e., the ability to decode motor attempt) of the sham group as a possible confounding factor and b) explicitly show that our sham design was successful in decoupling any kind of BCI decoding from FES.

Concerning the first point, in order to show that the average BCI aptitude of participants did not differ in the two groups and cannot explain recovery, we rely on the single-sample detection rate, which is the most suitable metric for such comparisons, as it does not depend on the extra, manually adjustable parameters of our evidence accumulation module (mainly the smoothing factor and the confidence thresholding, please see Leeb *et al.*, 2013). This single-sample detection rate was extracted by classifying “offline” the therapeutic-session EEG data of sham group participants using the same type of classifiers as for the BCI group¹. The detection rate metric did not differ between the two groups (BCI: 63.8% ± 17.1%, sham: 63.7% ± 18.1%, unpaired two-sided Wilcoxon non-parametric test, p=0.99). Furthermore, detection rate does not correlate with FMA improvement (Pearson’s correlation, r=-0.33, p=0.1). Therefore, BCI proficiency cannot explain the clinical outcomes.

Regarding the second point, based on these detection rates and using a conservative confidence threshold of 0.55 for all patients, we extract a simulated hit rate (i.e., percentage of trials where motor attempt was finally detected). The correlation between the real FES and the simulated one (as extracted above) in the BCI group was significant for all subjects (p<0.05, average r=0.36, Pearson’s correlation). On the contrary, in the sham group, the average correlation was r=0.035 (not significant for any of the patients, p>0.34, Pearson’s correlation).

In conclusion, the sham stimulation modality successfully decoupled FES stimulation from BCI performance, as intended by our experimental design. As elaborately discussed in our reply to Comment 1.15, the fact that such contingency was enforced only in the BCI group explains the larger clinical improvements in the BCI group. This discussion has been added in the now considerably enriched sub-section of the results entitled “Brain-Computer Interface”.

Comment 1.14: The lack of decline between 6-36 weeks is of note and needs to be further discussed. What were participants receiving during this time? Their standard rehab (passive movements)?

We are particularly grateful to this reviewer as, indeed, the preservation of strong recovery effects revealed by the follow-up evaluation is a unique outcome of our study: The studies in (Ramos-Murguialday *et al.*, 2013; Pichiorri *et al.* 2015) contained no follow-up results, while in Ang *et al.*, 2014 the effects are not preserved follow-up. This point is now highlighted in the introduction and discussion of the revised manuscript.

After concluding the proposed intervention, the participants resumed their normal life where they did not receive any particular therapy. This has been clarified in the manuscript.

Comment 1.15: It is surprising to claim activity-dependent plasticity given that there is on average 4s between initiation of the cortical activity and actuation of the FES. Presumably you would expect that activity dependent plasticity would require a much smaller timescale between cortical input and

¹ The detection rate of participants in the sham group has been extracted using classifiers trained by the same engineers, at the same time (immediately after the calibration session) and with the same type of data (16-channel EEG calibration session) as for the BCI group. Such classifiers were trained even for the sham group despite not being used during the therapy (the reviewer might recall that sham stimulation was based on random generation) as part of the double-blinding procedure.

associated movement. Can you provide evidence of other literature where you see plasticity occurring with such a large latency?

Thank you for this insightful comment (please, see also reply to Comment 4.2). We by no means intend to claim that activity-dependent plasticity is possible with large latencies. Instead, in the revised manuscript, we provide strong evidence that the desired contingency and immediacy are fulfilled only in the BCI group. The new analysis, added in sub-section “Brain-computer interface” of the results and outlined in this reply, supports the idea that the existence of significant clinical benefits only in the BCI group is best explained in the framework of activity-dependent plasticity mechanisms and justified by our experimental design.

The underlying assumption behind the reviewer’s comment seems to be that the relevant EEG patterns appear only at the onset of the attempted movement. Consequently, the reviewer concludes that these occur much earlier than the provision of FES. On the contrary, our protocol is built on the premise that the motor-dependent SMR modulation spreads throughout the trial as patients had to **sustain** the movement (please, see reply to Comment 1.10 above). Therefore, we only view as critical the cortical pattern immediately preceding the FES, where the prerequisite of time contingency between efferent and afferent volleys is satisfied.

The BCI’s evidence accumulation and decision thresholding modules are designed to guarantee that, in the BCI group, FES is only delivered when the patient is currently sustaining such SMR activity, as detected by the BCI decoder. As a consequence, the confidence threshold can only be reached if motor attempt has been sustained for long enough and the latest PSD sample has been classified as “motor attempt”. Furthermore, integrating consecutive PSD samples instead of delivering FES immediately after detecting the first “motor attempt” sample helps to avoid false positives, which can diminish long-term potentiation (or lead to long-term depression).

We can thus compute the contingency table between motor decoding and FES (**new Supplementary Table S1**, also **Table 1 below**), where the diagonal matrix elements represent the different types of contingency, and the non-diagonal elements quantify its absence. New **Table 4** in the revised manuscript (**Table 2 below**) reports standard contingency metrics derived from the contingency table that should lead to long-term potentiation. In particular, Accuracy results (please, see also new **Fig. 6** in the revised manuscript, and also **Fig. 2 below**), which combines all elements of the contingency table, shows that: (i) it was significantly better for BCI patients than for sham patients (two-tailed unpaired t-test, $p < 10^{-4}$) (**Fig. 2a left**), (ii) it correlates with FMA improvement when all patients are taken together (Pearson’s correlation, $r=0.47$, $p=0.013$) (**Fig. 2a right**), and (iii) it correlates with increases in connectivity within the affected sensorimotor cortex in the μ and β bands (Pearson’s correlation, μ : $r=0.49$, $p=0.02$; β : $r=0.55$, $p=0.005$) (**Fig. 2b**). Interestingly, similar results also hold for other metrics in **Table 2** (correlation with connectivity not shown for legibility). Additionally and critically, our intervention successfully decoupled BCI output from FES in the sham group, as can be seen by the much lower number of TPs and much higher number of FPs and FNs in this group, compared to the BCI group.

Table 1. Contingency table between motor attempt detection and FES.

		Motor Attempt Detection	
		Yes	No
FES Stimulation	Yes	True Positives (TP)	False Positives (FP)
	No	False Negatives (FN)	True Negatives (TN)

Table 4. Statistics of metrics of contingency between motor attempt decoding and FES and their correlation to motor recovery across all patients (Δ FMA: post-pre FMA scores). Statistical significance of group differences extracted with two-tailed unpaired t-tests. Statistical significance of correlations to motor recovery extracted with Student's t distribution.

Metric	Definition	Group statistics			Correlation with Δ FMA	
		BCI	Sham	p-val	r	p-val
	TP	72.36%±10.53%	42.30%±11.52%	< 10 ⁻⁶	0.38	0.051
	TN	12.52%±7.31%	12.16%±6.93%	0.89	0.11	0.60
	FP	0.00%±0.00%	21.50%±11.33%	< 10 ⁻⁶	-0.25	0.19
	FN	14.00%±8.43%	24.05%±6.70%	0.002	-0.50	0.009
True Positive Rate (TPR) / Recall / Sensitivity	$TP / (TP+FN)$	83.76%±9.78%	63.79%±1.01%	< 10 ⁻⁶	0.51	0.007
True Negative Rate (TNR) / Specificity	$TN / (TN+FP)$	91.22%±17.05%	34.83%±5.50%	< 10 ⁻¹¹	0.37	0.057
Positive Predictive Value (PPV) / Precision	$TP / (TP+FP)$	100%±0.00%	66.26%±17.88%	< 10 ⁻⁷	0.27	0.175
Negative Predictive Value (NPV)	$TN / (TN+FN)$	49.85%±23.75%	33.46%±18.81%	.059	0.47	0.013
Accuracy	$(TP+TN) / (TP+TN+FP+FN)$	84.88%±7.28%	54.46%±4.85%	< 10 ⁻⁴	0.47	0.013

Figure 2: Accuracy of contingency between last PSD sample classification and FES. (a) *Left.* Values for the BCI and the sham groups ($p < 10^{-4}$). *Right.* Accuracy vs ΔFMA score (post-pre intervention), together with the least-square fit lines for both groups pooled (black line) and for each group separately (color lines). (b) *Left.* Accuracy vs $\Delta Connectivity$ in μ band (post-pre intervention), together with the least-square fit lines for both groups pooled (black line) and for each group separately (color lines). *Right.* Accuracy vs $\Delta Connectivity$ in β band (post-pre intervention), together with the least-square fit lines for both groups pooled (black line) and for each group separately (color lines).

Reviewer #2

Comment 2.1: To my best knowledge, no previous paper has tested the clinical effects of EEG-FES in a controlled experiment with as many stroke patients as in this study. The investigation of neural correlates is also novel, although I am concerned by several aspects of it (details in coming sections). However, one study that came out earlier this year presented a very similar intervention (EEG-FES to improve arm and hand function), and showed comparable clinical improvements after 1 month of treatment:

Ibáñez, J., Monge-Pereira, E., Molina-Rueda, et al. Low Latency Estimation of Motor Intentions to Assist Reaching Movements along Multiple Sessions in Chronic Stroke Patients: A Feasibility Study. *Front Neurosci* 11, 126, 2017

This paper had other similarities, such as the authors also retrained the classifier every day. In contrast, they did not have a control group and the intervention group only had 4 patients, which the authors did not follow up. However, the methods were very similar.

There is at least another conceptually similar study, although it focuses on rehabilitation of lower limb function. Mrachacz-Kersting and colleagues used EEG to trigger a common peroneal nerve stimulus during a specific phase of the movement-related cortical potential. Using this approach, they were able to improve walking function in stroke patients. This study had a control group and was similarly powered to the current paper, although they only performed 3 sessions and did not do a follow-up assessment:

Mrachacz-Kersting, N., Jiang, N., Stevenson, et al. Efficient neuroplasticity induction in chronic stroke patients by an associative brain-computer interface. *J. Neurophysiol.* 2015

Thank you for bringing up the study by Ibáñez et al. to our attention. We have discussed it in the revised version of the manuscript. Despite they also coupled BCI to FES, there are two fundamental differences with respect to our work. Firstly, as the reviewer pointed out, Ibáñez et al.'s study had a small sample (4 patients), did not have a control group, and did not include a follow-up evaluation. Furthermore, only one out of four patients exhibited a clinically significant improvement in the motor function department of the FMA-UE (>5). Secondly, they calibrated the whole BCI decoder (doing feature selection and decoder training) at the beginning of each therapy session in order to optimize BCI performance. But this procedure might hinder brain plasticity since BCI features changed frequently thus modifying the recruited efferent pathways participating in eventual activity-dependent plasticity mechanisms. On the contrary, we only adjusted decisions thresholds to make the task challenging for the patient, not to optimize BCI performance.

Regarding the study by Mrachacz-Kersting et al., they were not actually coupling BCI to FES. Indeed, during the intervention the time to deliver FES was estimated from pre-recorded behavioural and EEG data. Please note that we already cited this work, but without discussing the BCI component (i.e., closed-loop control of FES via brain signals) as there is none.

Comment 2.2: As mentioned above, this study provides clinical evidence that supports quite convincingly emerging ideas in the field. I am more concerned about the impact of the authors' investigation of underlying mechanisms, as the authors did not report a thorough single patient analysis (only a correlation of one of the features with the gross clinical improvement).

Thank you for the suggestion. Regarding the connectivity, we have replaced the original barplots in Figure 4 (now Figure 3) by boxplots that better show the distribution of the values for all patients.

More importantly, we are now reporting the correlation between online BCI metrics, clinical improvement and connectivity which imply the existence of activity-dependent plasticity (please, see reply to Comment 1.15).

Comment 2.3: There is a clear group effect in terms of clinical improvement. However, a single-patient analysis of most aspects of the study is missing. Most critically, the authors only report group neural correlates, namely changes in functional connectivity and synchronization/desynchronization of cortical rhythms. The findings of the later also seem rather vague, and the analysis should be made in a more detailed fashion (see some possible ideas below). I am also concerned about the lack of

comments about 4 (of 14) patients in the intervention group that did not improve clinically. The same about 4 patients in the control group that improved, two of them in a extent similar to the intervention group.

Please, refer to our reply to Comment 2.2. Regarding the 4 BCI patients who did not improve and the 2 sham patient who did, we have not been able to find any factor that explains this issue. The lesion characteristics seem to also be a bad predictor of recovery and unable to explain these exceptions (please, see reply to Comment 3.7). We believe that this can only be addressed by incorporating a larger set of neuroimaging techniques (e.g., fMRI, MEP). We have included this point in the discussion of the revised manuscript.

Comment 2.4: There are a few experiments the authors could do, but that seem rather unrealistic at this point. One would be measuring corticospinal excitability using TMS to generate motor evoked potentials and test more directly the extent of corticospinal changes. The other would be recording functional MRI, rather than EEG and do single patient analysis of hand/digit representations (as in Ejaz et al Nature Neurosci 2015) as well as more precise connectivity measurements.

Thank you for the suggestions. We are considering these additional experiments for future studies.

Comment 2.5: The most important paper that the authors do not discuss and that I am aware of is Ibáñez et al. 2017. There are also at least a couple of reviews that address the use of neurostimulation triggered or controlled by detected movement intent that the authors could benefit from discussing:

Ethier, C., Gallego, J., Miller, L., 2015. Brain-controlled neuromuscular stimulation to drive neural plasticity and functional recovery. *Curr. Opin. Neurobiol.* 33, 95–102

Jackson, A., Zimmermann, J.B., 2012. Neural interfaces for the brain and spinal cord--restoring motor function. *Nat. Rev. Neurol.* 8, 690–9

We have added the references to these papers, and discussed the differences between our work and that of Ibáñez et al. in the manuscript (please, see reply to Comment 2.1).

Comment 2.6: Figure captions should have larger font size.

We have increased the font size for the captions.

Comment 2.7: The paper would benefit from following each main result by a short interpretation. The methods could be moved to the end.

Thank you for your suggestion. For the sake of conciseness, we preferred to put the interpretations only in the Discussion. Regarding the methods, we will follow the format of the journal upon eventual acceptance, as advised by the instructions for authors.

Comment 2.8: HAVE THE AUTHORS DONE THEMSELVES JUSTICE WITHOUT OVERSELLING THEIR CLAIMS? Yes, for the most part. I am only concerned about:

1) some claims that are being made based on two correlations that are barely significant ($p = 0.048$ and $p = 0.046$), and for which the data do not seem strikingly convincing (see “Is the statistical analysis of the data sound?”)

2) the authors also claim that they were the first group to re-adjust the BCI every session to facilitate ease of use. However, at least Ibáñez et al. have done it before

3) I also miss that the authors do not discuss that the intervention patients improved in the gold standard clinical test for stroke (FMA-UE), but not according another scale of upper limb function (ESS, European stroke scale). Neither did they comment on the lack of significant improvements in spasticity (Ashworth scale). They also did not comment on the four BCI patients that did not improve, or the 4 control patients that did improve.

1) Although we agree that the statistics are barely significant, we have performed additional analysis that further support our claims (please, see reply to Comment 2.31). In any case, we have lessened the claims.

2) Please, note that, in fact, our claim is rather the opposite. We have revised the manuscript to clarify this issue in the Discussion. In other works, and in particular Ibáñez et al., they changed the BCI

classifier for each session. As a result, this naturally leads to the use of different features (i.e. different pairs of frequency rhythms-channels) for each session recorded with the patient, possibly adding undesired variability to the therapy. For a more detailed explanation, please, see reply to Comment 2.1.

3) We have now incorporated these elements in the discussion. Briefly, although our intervention only targeted motor upper-limb recovery (better captured by FMA scores), we wanted to check other clinical aspects.

4) As mentioned in the reply to Comment 2.3, we must admit that, despite the different reported correlations, we have not been able to explain why four BCI patients did not improve and four sham patients (although only two of them above the clinical relevance threshold of 5 FMA-UE points) did improve above 5 FMA-UE points. None of the reported analyses (including the new ones) explained these exceptions. No demographic or clinical factor (including lesions, please, see reply to Comment 3.7) could shed light on this matter.

Comment 2.9: I would welcome additional details about how they trained the classifier, and on some of the functional connectivity and synchronization/desynchronization analysis methods. The latter could go as Supplementary information.

We have added more details about the training of the BCI classifier in the new Supplementary Material. Please, see also our reply to Comment 1.9. Regarding functional connectivity and sLoreta, we consider that all technical details are sufficiently covered in the Methods and the references therein.

Comment 2.10: Is the statistical analysis of the data sound? For the most part, yes. I am only concerned about:

1) the analysis in Fig. 4 (bottom): The authors compute the linear correlation (using Pearson) between connectivity changes and clinical improvements. They find that these variables are significantly correlated, which they are, but the P-values are $p = 0.048$, and $p = 0.046$ respectively. Looking at their data and given their small sample size, I think they should be more cautious when interpreting this result. However, they use this result several times to back up other claims.

2) in the sLoreta analysis, the authors pooled all the intervention and control patients together and then computed their post-intervention differences. They then report results for the group differences using a quite generous significance threshold ($P < 0.05$). They should report the exact value. Most importantly, also show single- patient and group pre vs. post differences.

3) Given the large standard deviation in Fig 3b and 4, the authors should report single-patient results in addition to error bars. I presume that examination of within- patient data may also illuminate within group differences

1) Please refer to our replies to Comments 2.8 and 2.31.

2) For the sLoreta analysis, we were limited by the software capabilities. sLoreta did not allow to have a statistical design of 3 factors (group: SHAM vs BCI; time: pre vs post; task: motor vs rest). For this reason, we performed two post-hoc analysis (mixed non-parametric analysis) by splitting the factor time. Our results showed that, while there were no significant differences in the pre-intervention condition, these differences existed in the post-intervention condition as reported in the figure. Regarding the p-values, we now report the minimum p-value obtained, please, see Fig. 4. Regarding the single-subject analysis, please refer to answer to Comment 2.2.

3) Following this and comments from Reviewer 1, we have added additional single-subject analyses. Please refer to reply to Comment 2.2.

Comment 2.11: Nothing is critical, although Supplementary Material providing details on the Classifier and the neurophysiological analyses could be helpful. The authors do provide references, though.

Please, see our reply to Comment 2.9.

Comment 2.12: Use of FES vs. NMES. I think the authors should rather use FES as they do use muscle stimulation to generate a functional movement. This distinction is important, for example, for the Discussion, where some studies they mention used stimulation mostly to activate sensory pathways (e.g., Conforto et al.), which is different to what the authors did.

The reviewer is right. We now use the term FES throughout the manuscript.

Comment 2.13: I do not think the authors should call the BCI “brain control” as they only used detected movement intent to trigger the stimulation; brain-triggered?

Thank you for the suggestion. However, brain control (rather than brain-triggered) is the usual term in the BCI field.

Comment 2.14: There are several acronyms that are not described: K-S, sdDTR, MNI standard brain, SMA, CIMT, etc.

We now define all the acronyms in the revised manuscript.

ABSTRACT:

Comment 2.15: Explain more clearly that association of neural activity and generated movement is key.

In response to similar criticism by all reviewers we have elaborately referred to this issue in the revised manuscript, taking care to add the corresponding new analysis. Please, see our replies to Comment 1.15 and Comment 4.2.

Comment 2.16: Last line: “correlated with the primary outcome” -> “functional improvement”

We have corrected this in the revised manuscript.

INTRODUCTION:

Comment 2.17: A forward step: a step forward

We have corrected this in the revised manuscript.

Comment 2.18: “in particular via massive and timely recruitment of muscle spindle feedback circuits” -> Clarify

We have added some additional details. It now reads: “in particular via massive and timely recruitment of Golgi tendon organs and muscle spindle feedback circuits.”

We have also elaborated this point in the Discussion (see Comment 2.41).

Comment 2.19: “several studies suggest that FES increases cortical excitability” -> the authors only cite one

We have added two more references:

Pitcher, J. B., Ridding, M. C., & Miles, T. S. (2003). Frequency-dependent, bi-directional plasticity in motor cortex of human adults. *Clinical Neurophysiology*, 114(7), 1265-1271.

Ridding, M. C., McKay, D. R., Thompson, P. D., & Miles, T. S. (2001). Changes in corticomotor representations induced by prolonged peripheral nerve stimulation in humans. *Clinical Neurophysiology*, 112(8), 1461-1469.

METHODS:

Comment 2.20: Was the lesion size identified using imaging or clinically? Describe

The lesion size and precise location were identified using magnetic resonance imaging (MRI) or computerized tomography (CT) scans depending on the patient. We have added this information in the revised manuscript and **Supplementary Figure S4** (also Fig. 4 below).

Comment 2.21: Provide brief details about the methods for selecting the classifier features. I don't think canonical variate analysis (the method used by Leeb et al.) is a “state-of-the-art” machine learning technique; it's rather established at this point.

Please, see reply to Comment 1.9. Regarding extracting discriminability of features, there exist several

equivalent methods (Fisher Score, r squared, CVA), none of which is universally superior. In any case, we have removed the qualifier “state-of-the-art” in the revised version of the manuscript.

Comment 2.22: Did the authors leave the classifier constant across sessions and only change the threshold?

Indeed, the same classifier trained with the data of the calibration session for each patient was used throughout the therapy and only the confidence threshold was varied. We are sorry for the misunderstanding in this regard. One of the main constraints we had was to fix the features and classifier, to ensure homogeneity across sessions and maximize any possible plastic effect. On the other hand, we varied the threshold for each session to make the task feasible and challenging for each patient. We have clarified this issue in the Materials and Methods and in the Discussion of the revised manuscript.

Comment 2.23: What would be the maximum number of FES trials per run in the case of perfect performance?

The maximum number was 15, which corresponds to the total number of trials per run throughout the study. We have highlighted this information in the revised manuscript.

Comment 2.24: EEG markers of neuroplasticity: the authors may be measuring changes in functional connectivity that are not mediated by changes in underlying circuitry. Since these are very difficult to differentiate, I'd suggest using another word such as functional connectivity.

We believe that changes in electrophysiological biomarkers are a plausible indicator of neural plasticity. However, we agree with the reviewer that this may not be the case. We have lessened the claims in the manuscript and acknowledged this limitation in the Discussion of the revised manuscript: *“It must be noted that, even though the identified post-intervention changes in electrophysiology are not a direct proof of cortical reorganization, they are highly indicative of those.”*

Comment 2.25: “Indirect cascade influences” -> explain

Indirect cascade influences refers to the case where if region A influences B (denoted as A->B), and region B->C, then A indirectly influences C as well. Our analysis only keeps direct influences. We have clarified this in the manuscript.

Comment 2.26: How was the threshold adjusted? based on a few calibration trials at the beginning of the session?

The threshold was empirically adapted at the beginning of each run by the therapist, based on how the patient performed in the previous run. Please, see also reply to Comment 1.12.

RESULTS:

Comment 2.27: Why do the authors think that online performance seems to overall decrease over time? Functional connectivity changes or changes in cortical rhythms that make the classifier become obsolete? A within-subject cross-session analysis would be an informative Suppl. Fig. to understand if this is a group effect or an spurious result of inter-subject differences. The authors could also compare BCI performance offline with the online threshold as well as with an optimally-selected threshold. This would be another interesting Suppl. Fig.

Thank you for these suggestions. We are raising this point because Fig. 2 might imply such a decrease. The newly added **Supplementary Figure S3** (also **Fig. 3 below**), demonstrates that the grand average of confidence threshold values is increasing over time, while the BCI performance remains stable. Hence, it is the increasing task difficulty (and not changes in the classification performance) that has brought about the slight decline of hit rates that we can see in Fig. 2. Of note, this is a group effect, as the majority of individual patients showcase such an increase in threshold values (not shown in the interest of clarity). We assume that this has probably been beneficial rather than detrimental to the clinical outcome, as maintaining patient motivation throughout the therapy is crucial and the decline of hit rates has only been marginal. We have added a short discussion of this in the revised manuscript.

The reply to Comment 1.13 provides additional “simulated” performance metrics. On the basis of all this information, we believe that extracting another simulated hit rate metric using the subject-specific

optimal confidence thresholds, as suggested here, has no additional value.

Figure 2: Evolution of simulated hit rate and confidence threshold throughout the intervention. Grand average (blue lines), standard deviation (shadows), and corresponding linear fits (red lines) across BCI group participants of a) simulated hit rate over runs with a conservative fixed threshold, and b) confidence threshold value.

Comment 2.28: An interesting Supplementary analysis would be to test BCI performance offline leaving out specific channels, to try and understand the contribution of that specific area. For example, the authors comment on the Discussion that contra-lesional channels may also support recovery. Would their classifier performance decrease drastically without them?

Despite we have not performed any classification analysis on subgroups of channels, we now report the values of feature discriminability, which strongly related to classification accuracy when using linear decoders. For more details please refer to reply to Comment 1.8 and associated **Figure 1** in this

letter (**Supplementary Figure S2**). This new figure shows which areas and frequency bands contributed the most to the BCI performance.

Comment 2.29: How long was the rest period the authors used for inferring cortical connectivity? Did they just append inter-trial periods? (“BCI trial where the patient is asked to rest”). Similarly, did the authors use the movement-related part of the trial for the sLoreta analysis? I could not find this information in the paper.

For the connectivity, we used a baseline period of 2.5 seconds prior to the resting task, which is customary in this type of analysis. In order not to bias the results obtained, we also used a window of 2.5 seconds during the resting task (that lasted 4 seconds in total). We only used the resting task, in particular a total of 15 trials per run. This task was controlled and patients were asked not to perform any movement or blink.

As for the sLoreta analysis, we used the movement-related part (trials) and compared them with the rest trials.

Comment 2.30: Clarify the following statement: “and also between the two groups after the intervention in favor of the BCI group in the μ (two-tailed unpaired t-test, $p = 0.003$) and β bands (two-tailed unpaired t-test, $p = 0.035$)”

We have rephrased the statement, as follows: “Similarly, the BCI group showed a significant increase after the intervention, in μ (two-tailed unpaired t-test, $p=0.003$) and β frequency bands (two-tailed unpaired t-test, $p=0.035$).”

Comment 2.31: “Both changes in connectivity in the μ and β bands showed a significant correlation (Pearson’s correlation, μ : $r = 0.41$, $p = 0.048$; β : $r = 0.41$, $p = 0.04000006$).” As mentioned above, I realize the changes are significant (only very slightly), but by looking at the raw data, it seems to me that the distribution of FMA-UE is quite scattered and the fits quite unreliable (e.g., the positive slope in μ for the BCI group seems to be largely driven by the patient with a connectivity change of 1.5). Please comment on this.

In response to the reviewer’s concern, we have also performed a non-parametric correlation (Spearman’s correlation), which is robust to outliers. New values show a highly significant correlation for μ ($r=0.55$, $p=0.005$) and a significant correlation for β ($r=0.51$, $p=0.01$). We have preferred to report the original Pearson’s correlation, instead of the most favorable Spearman’s, for the sake of consistency with all other correlations reported in the manuscript.

Comment 2.32: I am surprised about the sLoreta results. First, the main change happens in Cz, not around C3* (leg area vs. hand/wrist area). Second, the change in β seems greater in the unaffected as opposed to the affected hemisphere, and again close to the midline. I also have difficulties reconciling these results with the functional connectivity results. The groups were not different in terms of ERD/ERS before the intervention but ERD/ERS power increased post-intervention, for the BCI group only, in both hemispheres. At the same time, functional connectivity only changed for the affected hemisphere and for the BCI group. How do the authors combine these observations?

Although the reviewer is right, changes also expand over the ipsi and contralesional motor regions (e.g., C3* and C4*). Please also note that, following Reviewer 4 suggestions (please, see Comment 4.8), we have now changed the frequencies of analysis for sLoreta to make it consistent with those used in the connectivity. Regarding the link to connectivity results, we believe they are complementary metrics, as one does not imply the other.

DISCUSSION:

Comment 2.33: “Clinically important” -> “significant”

In the manuscript, we wanted to make the distinction between clinically important (or relevant) and statistically significant. Following the literature of stroke rehabilitation, an improvement of 5 FMA points in the chronic stage is considered clinically relevant, whereas any improvement below 5 points may be statistically significant, yet, not clinically relevant. We have rephrased the statement as “significant and clinically important”.

Comment 2.34: “to other sections of the FMA-UE —i.e., sections ‘synergies’ and ‘movement combining synergies’” -> The authors should describe this more clearly in the Results: they do talk about the hand/wrist vs. upper limb items in general, but do not emphasize this important observation.

We have added these findings in the results.

Comment 2.35: The authors state that “BCI turns a weak intervention, such as FES, into an effective one” -> I think the key is the association between motor intent and evoked movement. In my view this is a new intervention, rather than a combination of two.

We thank the reviewer for this observation. From our point of view, the manuscript already conveys the message that our novel combinatorial intervention (with the key prerequisite of contingency) *de facto* is a “new intervention”.

Comment 2.36: “consisting in an increase of neural desynchronization over the affected hemisphere “ -> In the results, the authors say that this increase is bilateral.

We have corrected this in the revised version of the manuscript.

Comment 2.37: While I agree with the following statement: “In particular, the change in EEG connectivity between ipsilesional M1 and premotor areas observed in BCI patients is in line with previous studies that highlighted the strong relationship of this connectivity pattern with motor deficits and their improvement with therapy after stroke (Wu et al., 2015).” it is interesting that Guggenmos and colleagues targeted premotor and sensory cortex and were able to induce clear recovery after a motor cortical stroke. Please discuss.

Although Guggenmos and colleagues’ intervention targeted premotor and sensory cortex in mice, they did not explore changes in connectivity patterns following recovery. This is why we only discussed similar analysis in humans. Please note also that we refer to the work of Guggenmos and colleagues only to justify our experimental design.

Comment 2.38: “brain nodes “ -> brain areas?

We have implemented this correction in the revised manuscript.

Comment 2.39: “purposeful cortical reorganization in relevant cortical areas” -> I agree that there seem to be group changes in ERD/ERS and connectivity but these do not prove cortical reorganization, in the sense of synaptogenesis, neurogenesis or LTP/LTD of existing synapses. It could just be changes in neural activity patterns generated with exactly the same structure.

The reviewer is right. We have lessened the claim, and now report that we believe the electrophysiological findings suggest a cortical reorganization in relevant cortical areas in the Discussion of the revised manuscript. Please, see also our reply to Comment 2.24.

Comment 2.40: The authors should elaborate better on the role of CST projections. Their involvement seems likely, but the reasoning is not entirely clear to me.

Following our results, our hypothesis is that closing the loop between motor intention and peripheral stimulation strengthen CST projections by, plausibly, cortical reorganization. The rationale is that patients exhibit a pattern of motor improvement that follows two stereotypical paths (Prabhakaran *et al.*, 2008; Winters *et al.*, 2015): they will either recover about 70% of their initial motor impairment or show little to no improvement. This seems to be largely independent of the lesion type, patient age, and the amount of therapy that is provided to the patients (Byblow *et al.*, 2015). It turns out that the group with poor improvement is characterized by a greater damage to the cortico-spinal tract (CST) (Byblow *et al.*, 2015; Buch *et al.*, 2016). Hence, we speculate that the BCI-FES intervention strengthened CST projections in chronic patients that, initially, did not follow the proportional path. Unfortunately, we don’t have imaging data to probe this hypothesis. We have clarified it in the discussion.

Buch ER, Rizk S, Nicolo P, Cohen LG, Schnider A, Guggisberg AG. Predicting motor improvement after stroke with clinical assessment and diffusion tensor imaging. *Neurology* 2016; **17**: 1924–1925.

Byblow WD, Stinear CM, Barber PA, Petoe MA, Ackerley SJ. Proportional recovery after stroke

depends on corticomotor integrity. *Ann Neurol* 2015; **78**: 848–859.

Prabhakaran S, Zarah E, Riley C, Speizer A, Chong JY, Lazar RM, et al. Inter-individual variability in the capacity for motor recovery after ischemic stroke. *Neurorehabil Neural Repair* 2008; **22**: 64–71.

Winters C, Van Wegen EE, Daffertshofer A, Kwakkel G. Generalizability of the proportional recovery model for the upper extremity after an ischemic stroke. *Neurorehabil Neural Repair* 2015; **29**: 614–622.

Comment 2.41: “FES recruits more muscle spindles and activates them faster, conveying also somatosensory information. “ -> Please Clarify. FES also activates Golgi tendon organs when the muscle is contracted. Moreover, spindle stimulation experiments normally use higher stimulation frequencies. Critically for the present work, the monosynaptic excitatory projections from spindles onto motoneurons may activate them concurrently with the presumed descending cortical command, thereby causing Hebbian association.

Thank you for helping us with a more detailed description of the mechanism behind our intervention. We have modified this part to say: “FES recruits more muscle spindles and Golgi tendon organs, activates them faster, and conveys also richer somatosensory information. Critically for the present work, the monosynaptic excitatory projections from spindles onto motoneurons may activate them concurrently with the presumed descending cortical command, thereby causing Hebbian association.”

Regarding the stimulation frequency, it is true that normally it is higher than 30 Hz in spindle stimulation experiments. However, lower frequencies also activate spindles.

Comment 2.42: “Remarkably, the distribution of significantly different voxels where neural desynchronization emerged post-intervention and that were correlated with increase of FMA-UE scores (Fig. 5) matched the distribution of selected features for the BCI decoders, both spectrally and spatially (Fig. 3a).” -> These are the affected areas, which are motor, so it’s not surprising that they are included in the decoders. What would happen if they associated activity from non-motor areas? would the intervention still work? This would be interesting to discuss

Our claim here was mostly a sanity check by showing that there is a correlation between the results obtained with the inverse method, and the electrophysiological features that were targeted by the therapy. We have removed this statement from the revised manuscript.

Regarding the second question, and following the results obtained in our work, we believe that any therapy using features not coming from motor areas would have diminished the clinical effect of the intervention. Our hypothesis is that the key component of the therapy is twofold: first, a time contingency between motor intention and feedback, and second, a task coupling between motor-related features and the sensory afferent feedback. We prefer not to include this discussion into the manuscript due to space limitations.

Comment 2.43: “Combination of these three properties may explain why our BCI approach yields relatively larger motor improvements than previous controlled trials with chronic stroke patients (Ramos-Murguialday et al., 2013; Ang et al., 2014). “- > Ibáñez et al got comparable clinical improvements although in a convenience sample of only 4 subjects. Comment

Please, note that, in fact, Ibáñez et al. reported FMA scores on several departments, and not only motor function like we do. In the motor function department, Ibáñez et al showed that only 1 out of 4 patients had a clinically significant improvement. In particular, their subjects improved/deteriorated by 14, 2, -2 and 4 FMA-UE points (4.5±6.8 points on average). Please, see reply to Comment 2.1 for more details.

Comment 2.44: “Nevertheless, the correlation over all patients of 16 FMA-UE improvements with changes in both types of EEG indexes (cortical connectivity and neural desynchronization) makes this

possibility unlikely” -> The correlation coefficient for at least one of those was barely significant.
Rephrase

See reply to Comment 2.31 (and also to Comment 2.8.1).

Reviewer #3

Comment 3.1: Foremost, the study is highly limited in its focus on hand extension. Monotonic improvements such as this have been previously shown with BCIs in both humans and primates. However, the major limitation of focusing on only a single muscle group is that they do not clearly allow for improvement in more than one functional movement direction. For example, it would not be easily possible to train individuals to perform both flexion AND extension movements under the current paradigm which would be essential for meaningful use.

Thank you very much for allowing us to clarify this point. The problem with training chronic stroke patients to perform hand/finger flexion is that, because of their spastic state, this is therapeutically counterproductive. This is why, in this first exploration of the BCI-FES intervention, we only focused on hand/finger extension. Of note, finger extension is a reliable predictor of future motor improvement, at least in acute/subacute stroke patients (Nijland *et al.*, 2010). Additionally, our results show that actually the clinical improvement is not only in the targeted functional movement direction, but also led to a significant improvement in the synergies department of the FMA-UE scale (please, see also Discussion in the revised manuscript and reply to Comment 2.34).

Furthermore, please note that our study is the first one that shows significant and clinically important improvements that remain 6-12 months afterwards (please, see also reply to Comment 1.14). Also, the two previous sham-controlled studies of a BCI-based intervention on chronic patients used a robot, while here we rely on NMES (called now FES in the revised version of the manuscript).

Nijland RHM, van Wegen EEH, Harmeling-van der Wel BC, Kwakkel G. Presence of finger extension and shoulder abduction within 72 hours after stroke predicts functional recovery. *Stroke* 2010; **41**: 745–750.

Comment 3.2: Second, while the authors provided a control group, they did not perform a switch-over control to confirm that improvement in performance was not simply due to a difference between rather than within patients. Even subtle differences in stroke size and impingement on the motor cortex can have dramatic differences in the likelihood of recovery over 6-12 months. While the authors do a commendable job in attempting to match the two groups, this is not sufficient to demonstrate that improvement is causally attributable to the BCI.

This is a good point. One of the main objectives of our experimental design was to have a follow-up evaluation, which was so far lacking in the related literature. While we agree that such subtle differences could have added some variability to the results obtained, we believe that a switch-over (or cross-over) control design would have been counterproductive, as they would have impeded the interpretation of follow-up evaluations. Also, having a different sham control group is the standard design for BCI and robotics rehabilitation clinical studies –at least the major ones we referred to in our manuscript. We have clarified this in the revised version of the manuscript.

Comment 3.3: Third, and in line with the comments above, it is not clear that there was any blinding performed. There is also an unusual procedure where the therapist gave verbal instructions to the patients to avoid eye or other movements that could ‘contaminate’ the EEG signal. Yet, it is not apparent whether the EEG signals were not affected and in whom.

The trial performed was double-blind (please, see also reply to Comment 4.4). We are sorry this was not clear in the original version of the manuscript. We have clarified this issue in the revised manuscript.

As for the artifactual components of the EEG, we have performed a new analysis showing that there was minimal contamination of the signal during the therapy (please, see reply to Comment 1.8).

Comment 3.4: Fourth, the use of an ANOVA is a non-standard way of looking for change in connectivity. While the authors provide a few references to its use, it would be helpful to confirm that similar findings could be obtained by more standard coherence and/or time-series analyses such as Granger causality. On that same note, did the authors examine coherence with the opposite hemisphere? There has been broad evidence that recruitment of contralateral circuitries such as the SMA can significantly influence recovery.

It should be noted that directed transfer function (DTF) is an extension of Granger causality to multiple variables. SdDTF is a modification of DTF using several trials to increase the temporal resolution and adopting partial coherence to avoid indirect cascade influences. Despite we agree that classical Granger causality is more standard in the field, the use of SdDTF is also accepted now.

Regarding the combination of SdDTF + ANOVA, although we agree that ANOVA has not been extensively used to study differences in connectivity, we believe that this does not limit the validity of the analysis. Mixed ANOVA is merely a statistical tool that allows comparing changes of the variable of interest at the population level (a connectivity metric in our case) against changes in the group (BCI vs. sham) and time (pre vs. post intervention).

With respect to the coherence of the affected hemisphere with the healthy one, as already mentioned in the manuscript, we did not find changes in connectivity. Using the same statistical test, the interaction Group*Time had the following p-values:

- left->right hemisphere, sensorimotor network, mu band: p=0.61
- left->right hemisphere, sensorimotor network, beta band: p=0.22
- right->left hemisphere, sensorimotor network, mu band: p=0.16
- right->left hemisphere, sensorimotor network, beta band: p=0.18

Finally, thank you for reminding us about the evidence that recruitment of contralesional circuitries can significantly influence recovery (Rehme *et al.*, 2011; Pundik *et al.*, 2015), although not all studies have found it (Ward *et al.*, 2003; Murase *et al.*, 2004). We discuss this point in the revised version of the manuscript.

Murase N, Duque J, Mazzocchio R, Cohen LG. Influence of interhemispheric interactions on motor function in chronic stroke. *Ann Neurol* 2004; **55**: 400–409.

Pundik S, McCabe JP, Hrovat K, Fredrickson AE, Tatsuoka C, Feng IJ, et al. Recovery of post stroke proximal arm function, driven by complex neuroplastic bilateral brain activation patterns and predicted by baseline motor dysfunction severity. *Front Hum Neurosci* 2015; **9**: 394.

Rehme AK, Eickhoff SB, Wang LE, Fink GR, Grefkes C. Dynamic causal modeling of cortical activity from the acute to the chronic stage after stroke. *NeuroImage* 2011; **55**: 1147–1158.

Ward NS, Brown MM, Thompson AJ, Frackowiak RS. Neural correlates of motor recovery after stroke: a longitudinal fMRI study. *Brain* 2003; **126**: 2476–2496.

Comment 3.5: Lastly, there has been extensive literature on the use of biofeedback and physical therapy for improving motor performance in individuals with stroke. It would be important to demonstrate that the proposed improvement provided through the BCI-FES system was not simply due to simple visual/sensory feedback.

Thank you again for raising another key point. The purpose of the sham-FES control group was to demonstrate that somatosensory feedback alone is not sufficient. At the same time, patients did not receive any other type of feedback (visual or sensory), apart from the instructions from the therapists.

MINOR COMMENTS

Comment 3.6: The references and manuscript structure do not appear to conform to the Nature Communication format.

The reviewer is correct. But, following editorial recommendations, formatting will be done only in the final version of the manuscript after eventual acceptance.

Comment 3.7: For patient comparison, it would be helpful to provide figures that display the precise stroke locations, sizes and distributions across the two patient groups.

Thank you very much for your suggestion. We have calculated the lesion's center of gravity and volume for all subjects based on MRI and CT scans available. No significant differences between the groups have been found neither for lesion volume nor for center of mass. Running statistical tests on single voxels, no statistically significant differences were found a) between groups and b) between high and low responders (independently of the group, criterion of $FMA_{Post} - FMA_{Pre} > 4$). Hence, the lesion does not seem to relate to the clinical outcome. We have added this information to the

Discussion of the revised manuscript together with a **Supplementary Figure S4** (also **Fig. 4 below**) with the lesions' average center of gravity and distribution for BCI and sham groups, as well as for responders and non-responders.

Figure 4: Average center of gravity and distribution of lesions. Average center of gravity marked with crosshair and average distribution of lesions for (a) BCI group, (b) sham group, (c) responders ($\Delta\text{FMA} > 4$), and (d) non-responders ($\Delta\text{FMA} \leq 4$). Right hemisphere lesions are mirrored to the left side. No significant differences between the groups have been found neither for lesion volume nor for center of mass.

Comment 3.8: Given the overlap, it would be important to include references to work such as that of Eberhard Fetz and/or Ali Razai.

We thank the reviewer for these suggested works. We have included some of these references in the manuscript.

Reviewer #4

Comment 4.1: In the field of Brain-Computer Interfaces, classifying between “active” and “rest” states is probably the easiest possible task. In figure 3, the authors report what seems like an average ~70%-80% single trial accuracy. I’m interpreting this to mean that the BCI failed to detect the active state once every 4 or 5 trials, even though patients were actively trying to extend their fingers for 7 seconds.

Please, consider that this is not necessarily so, especially for stroke patients that, because of their lesion, may have weakened sensorimotor rhythms that make this classification hard. In fact, as a recent study shows (López-Larraz et al, 2017), decoding of movement intention in stroke patients (N = 28) is far from trivial. Also, in general, it is easier to decode classes that elicit activity in different brain areas (e.g., imagination of left vs. right hand movements) than “active” vs. “rest” states. Also, please note that in the work of Ramos-Murguialday et al. (2013) (Supplementary Material section 7.3.3) they obtained a performance (trial-based accuracy) of less than 60%, which is barely above random, for the same kind of BCI tasks and population.

In any case, the goal of the BCI employed in this trial was not to maximize recognition performance, but to engage patients and promote brain plasticity. Accordingly, quoting from the original manuscript, “*The BCI confidence threshold that triggered FES was adjusted ...targeting an online performance of 10-12 FES per run.*” In this way we could sustain patient motivation and nearly guarantee the elimination of “false FES delivery”, which could in turn diminish the desired long-term potentiation effects. Inevitably, this had a minor impact on the “hit rate” (please, see Fig. 3 in this reply, new Supplementary Fig. S3 in the revised manuscript and our reply to Comment 2.27). In other words, non-perfect hit rates were the result of a deliberate experimental design rather than a BCI decoder failure.

López-Larraz E, Ray AM, Figueiredo TC, Bibán C, Birbaumer N, Ramos-Murguialday A. Stroke lesion location influences the decoding of movement intention from EEG. In *39th Ann Intl Conf IEEE Eng Med Biol Soc* 2017.

Ramos-Murguialday A, Broetz D, Rea M, Lær L, Yilmaz Ö, Brasil FL, et al. Brain-machine interface in chronic stroke rehabilitation: a controlled study. *Ann Neurol* 2013; **74**: 100–108.

Comment 4.2: When the BCI successfully detected a movement attempt, it actually needed a delay of 3 to 5 seconds from the start of the neural activity related to motor intent. In my opinion this poor temporal resolution is an important confounding factor, which make the distinction between the BCI-FES and sham-FES intervention unclear. The authors’ claim that, in the BCI case, the stimulation is “driven by neural activity”. However, for the sham FES intervention, the delivery of FES also occurred in conjunction with the patients’ attempting hand extension. In both cases, peripheral stimulation is delivered concomitantly with neural activity related to voluntary effort. In the BCI case, the exact timing of the FES onset was related to the algorithm confidence and an arbitrary threshold.

Therefore, the authors’ interpretation that “The synchronous activation of cerebral motor areas and of peripheral effectors might have induced Hebb-like plasticity and strengthened CST projection” in the BCI group but not in the sham group is difficult to accept, as the conjunction of cerebral activity and peripheral stimulation is similar in both groups.

Thank you for this important observation. We have proceeded with additional analysis to address this major point. Please, refer to our reply to Comment 1.15, who has raised a similar point on whether the reported clinical outcome can be explained in the framework of activity-dependent plasticity.

Briefly, although it is true that in both groups all trials should, in principle, contain “motor intent” (assuming that all subjects followed the instructions), we show that only in the BCI group the FES has been time contingent to reliable motor decoding. Given the instruction for sustained motor attempt, we hypothesize that neural correlates of motor attempt spread throughout a trial and are not confined to the attempted movement onset. Thus, limiting the analysis to the decoding of the last PSD sample in the trial (where time contingency with eventual FES is preserved) and defining a contingency table between motor decoding and FES (please, see Table 1 in this reply and Supplementary Table S1 in the revised manuscript), we can show that only BCI participants received FES contingent to reliable motor neural patterns. Various contingency metrics derived from this matrix (please, see Table 2 in this reply

and Table 4 in the revised manuscript) differ significantly in the two groups and correlate significantly with motor recovery. This suggests that the efficacy of the proposed therapy could indeed be explained in the framework of Hebbian plasticity, and that our experimental design has been carefully tailored to comply with its prerequisites only in the BCI group.

This information has been added in the sub-section “Brain-computer interface” of the Results and in the Discussion of the revised manuscript.

Comment 4.3: However, the difference between the groups may lie in the intensity of the voluntary effort. In the discussion, the authors clarify that the BCI task was made difficult by adjusting parameters every session. “Active participation and close attention play an important role in promoting plasticity”. Thus it’s possible that the BCI patients were more actively engaged than the sham group. If the modulation of voluntary effort was the determining factor for the therapeutic effects, it would be important to emphasize that.

-> Is this what Figure 5 suggests? It’s not clear to me. My understanding is that the data for this figure were obtained not during BCI control, but in a separate recording, similar to the calibration task, while patients responded to instructions to rest or attempt movement, with no FES delivered. While this figure is compelling in that it shows improved neuronal activation for the BCI group, it does not show a difference in neuronal modulation during the task. Can the authors produce analyses of EEG activity recorded during the therapy session, in order to understand whether the BCI group produced a greater voluntary effort?

As the reviewer has correctly figured out, Figure 5 (now Figure 4 in the revised manuscript) reports on the pre- and post-intervention high-density EEG sessions and, consequently, conveys no information about the intensity of voluntary effort during the therapy sessions. Although an objective and accurate metric to quantitatively assess motivation and effort is hard to derive, several pieces of evidence point to the direction that the recovery cannot be explained on this premise.

First, it must be underlined that our experimental design has been carefully shaped to eliminate such a bias. Specifically, the study was double-blind (both patient and therapist –as well as the clinical outcome evaluators– were unaware of the patient’s group allocation, please see also our reply to your Comment 4.4 below), so that the inherent or therapist-driven focus and motivation could not systematically differ in favor of one group or the other.

Second, as mentioned in the original manuscript, both groups performed the same number of runs per session on average (BCI = 6.0 ± 0.7 , sham = 5.9 ± 0.7), which indicates that all subjects exhibited a similar high level of engagement and motivation across the intervention. As a reminder, patients could stop the intervention whenever they wanted (minimum 2 sessions, maximum 8 sessions).

Third, to better assess patient’s motivation in the two groups, as recommended by the reviewer, we analyzed the “online” therapy (low-density) EEG data. Assuming that a significant bias in terms of patient’s voluntary effort should be reflected in the “quality” of EEG motor correlates and, consequently, in the BCI aptitude, we show (please, see also our reply to Comment 1.13) that the motor attempt detection rate metric (single-sample detection rate in online trials) does not differ between the two groups (BCI: $63.8\% \pm 17.1\%$, Sham: $63.7\% \pm 18.1\%$; $p=0.99$, unpaired two-sided Wilcoxon non-parametric test). This indicates that both groups were engaged in motor attempt equally over the duration of a trial (please, see also response to Comment 4.2 above).

Concluding, we believe there is enough evidence to eliminate the case of a motivational/engagement bias between the two groups. This information has been added in the sub-section “Brain-computer interface” of the Results in the revised manuscript.

Comment 4.4: The authors point out that : “There is a possibility that evaluators could have known the group to which some patients were allocated. This fact may have inflated the effect size.” This is a very odd observation. If the authors know, because of a comment from evaluators or patients for example, that there are cases when evaluators did know the group to which patients belong, then it should be clearly and plainly admitted. If the authors are just speculating and never had any indication that this could be the case, then this comment should be removed from the manuscript. Now I’m assuming that the reason why this comment is there, is that the authors actually know that the

evaluators were not entirely blind to group assignment, and for scientific rigor it is important to disclose that.

The trial has been explicitly designed to be double-blind (both participants and therapists/evaluators were supposed to be blind). In order to comply with this guideline, group allocation was never revealed to therapists/evaluators and the sham group stimulation properties were designed to replicate the stimulation dynamics of an average participant in the BCI group. Nevertheless, for scientific rigor, we feel obliged to report that there is anecdotal evidence suggesting that very experienced therapists (having participated in previous studies with the same BCI) may well have guessed a participant's allocation by observing the BCI feedback dynamics during the therapeutic sessions. We discuss this point in the revised version of the manuscript.

Comment 4.5: The description of the FES is lacking details. The electrical stimulation is the main component of the intervention, and further description is critical. Examples of important missing information: What was the duration of the stimulus train? How were the current and frequency chosen by the therapist? What were their mean values, and what did it represent in terms of % of motor threshold? The authors report that the number of FES per run was slightly, though not statistically significantly, higher for the BCI groups. What about other FES parameters? If the details of this intervention varied between patients, it is important that these parameters be also included in the statistical tests performed between the BCI and the sham groups, to confirm that the FES intervention itself was not different between groups and did not explain changes in outcomes.

We have extended the corresponding paragraph of the Materials and Methods in the revised manuscript and included a section in the Supplementary Material to give details on the FES train shape and duration. Of note, the reported parameters (stimulation frequency, current amplitude and pulse width, along with the shape/duration of the stimulation train and the electrode placement) encompass the whole range of configurable aspects of the commercial FES device used.

Regarding the selection of the free FES parameters, the original manuscript read *“Electrical stimulation parameters such as current amplitude (ranging between 10 and 25 mA), pulse-width (500 μ s), and stimulation frequency (ranging from 16 to 30 Hz), as well as electrodes placements were configured at each session by the therapist in order to elicit the desired hand extension movement in a comfortable way for the patient.”* This configuration was done in the beginning of each therapeutic session, before the first run, targeting the same movement across all patients and sessions. This strategy was based on the belief that, given the large variability of physiological responses and tolerance to the same FES train even in different sessions of the same participant, eliminating the FES as a confounding factor consists mostly in trying to trigger the same overt behavioral output (elicited movement) rather than balancing each individual parameter. This discussion has been added to the Supplementary Material.

In addition to this, available data demonstrated that the two free parameters (stimulation frequency and amplitude) did not confound our results. The stimulation frequency was in practice fixed to 25 Hz for all patients. This proved to be enough to achieve the desired movement for all patients. Regarding the stimulation amplitude for each single session, we are in possession of the data of 17/27 patients (8 BCI, 9 sham). The stimulation current was between 10 and 29 mA, with a total mean of 18.85 ± 4.78 mA. The maximum increase of stimulation current between sessions 1 and 10 was +3 mA, while the biggest decrease was -5 mA. The mean difference across patients was only -0.53 mA. Most importantly, none of the following metrics exhibited significant differences between the two groups or correlates significantly with FMA improvement: i) average and standard deviation of current amplitude values used, ii) start values, iii) end values, iv) max-min value difference, v) maximum consecutive change, and vi) average value used. Only the maximum value used was significantly different between BCI and sham groups ($p=0.041$, average BCI: 18.4 mA, average sham: 23.0 mA). This effect is attributed to an “outlier” BCI participant, who was the only one to use the lowest possible stimulation current (10 mA). Even in this case the highest maximum amplitude was observed in the sham group, what intuitively should have promoted the sham rather than the BCI therapy. This discussion has been added to the Supplementary Material.

Comment 4.6: Figure 2b illustrate the main outcome of the intervention. From the methods' description, the statistical tests seem appropriate and well-designed. However, from the figure, we can see a considerable overlap in the standard deviation error bars between conditions, even though the authors report a statistically significant difference.

Indeed, the reviewer's intuition is correct: The between-group differences are not statistically significant pre-intervention ($p=0.69$, already reported in the manuscript, since it was a major goal of our allocation strategy to balance the primary outcome at baseline), but also post-intervention ($p=0.24$) and follow-up ($p=0.26$), despite the trend of increasing group difference is evident in the latter cases. For the sake of precision, please find below the exact figures in format mean \pm standard deviation.

Although significance at the level of group differences would be of course desirable, this result is hard to attain given the trial size. Still, the results included in the manuscript show that the BCI intervention is significantly superior to the sham one: ANOVA analysis reveals a significant effect of the TIME \times GROUP interaction ($F_{1,46} = 3.5$, $p=0.04$) and the post-hoc testing shows that a significant increase exists only for the BCI group ($p=0.005$), but not for the sham group ($p=0.21$). Furthermore, the group difference of the FMA-UE Post-Pre change (Δ FMA) is significant ($p=0.014$). All p-values refer to two-tailed paired/unpaired t-tests. Also, the effect size of the BCI-FES intervention was large (Cohen's $d = 1.03$).

The actual numbers illustrated are (in format mean \pm standard deviation):

FMA-UE Pre-Intervention BCI: 21.6 \pm 10.8
FMA-UE Pre-Intervention Sham: 19.9 \pm 11.2
FMA-UE Post-Intervention BCI: 28.3 \pm 14.5
FMA-UE Post-Intervention Sham: 22.0 \pm 12.2
FMA-UE follow-up BCI: 28.5 \pm 12.2
FMA-UE follow-up Sham: 22.7 \pm 13.1

Comment 4.7: spell out SdDTF on first occurrence. Define sLoreta and MNI standard brain on first occurrences.

We now define and spell all the acronyms the first time they appear in the text.

Comment 4.8: Please explain why for the connectivity analysis the β band was restricted to 18-24Hz, while 13-30Hz were analyzed for desynchronization analysis. I'm confused as to why they used these specific bands. These values don't match the frequency bands given by sLoreta presented in the methods (low: 12-18, mid: 18-21 high: 24-30).

The reviewer is right. Regarding the connectivity analysis, we relied on a beta subband that was suggested as the most common one for such analysis. As for sLoreta, we have changed the analysis to match the frequency bands to the one analyzed in connectivity. The new results are reported in the manuscript. Please note that with the new frequency bands the correlation between desynchronization and FMA was not significant anymore, and thus we have removed the findings from the manuscript.

Comment 4.9: "Two patients in the BCI group required longer time-outs to be able to deliver BCI commands (15 s)." -> It's not clear what the "time-outs" refer to. By definition I would assume that it's the time between trials, but from the context it sounds like it might refer to the movement execution phase (active period) of the trial. Please clarify and do not use the term "time out" unless referring to pauses between trials.

By time-out we refer to the event of "expiration"/lapse of the active period of a motor attempt trial in an online run, i.e., the time within which the patient should be able to reach the confidence threshold and receive FES. On the other hand, for the time period in-between two trials, we conventionally use the term "inter-trial interval". Since the "timeout" for such cases is very common concept in engineering and we cannot find a more suitable alternative, we have preserved this term in the revised manuscript. However, we understand that this term might be easily misunderstood in the clinical world, so we have clarified this definition in the Materials and Methods and in all places of the revised manuscript where the corresponding parameter is mentioned. We have also illustrated the concept of "time-out" in the caption of Figure 1.e.

Reviewers' comments:

Reviewer #1 (Remarks to the Author):

Summary

The study investigates the combined use of non-invasive (EEG) brain machine interface with neuromuscular electrical stimulation (NMES) to restore motor function to persons with chronic stroke. The study hypothesizes that BMI-NMES will enhance functional recovery because it is "tailored to reorganize the targeted neural circuits by providing rich sensory inputs via the natural afferent pathways so as to activate all spare components of the central nervous system involved in motor control...[and]...elicits functional movement and conveys proprioceptive and somatosensory information, in particular via massive and timely recruitment of muscle spindle feedback circuits". The study compares the functional recovery and cortical plasticity of two groups: group 1 received BMI-triggered NMES, while another received sham NMES. The investigation shows statistically significant improvement of the BMI group and not the sham group based upon clinical functional metrics (Fugl-Meyer Upper Extremity, Muscle Strength measures, etc...). The BMI group also showed enhanced activity in mu and beta bands that were not present in the sham group. The investigators conclude that this study offers a novel use of BMI and NMES for stroke recovery in addition to elucidating the underlying mechanisms of this recovery.

Comments

No new major concerns. The authors have thoroughly addressed my previous concerns and have better clarified how the experimental paradigm directly addresses the stated hypothesis.

Reviewer #2 (Remarks to the Author):

1. Hypothesis: After reading the authors' responses to all four reviewers, I'm slightly confused about the hypothesis of this study. In particular:

- a) The authors state that the key is the association between the (presumably) ongoing efferent command and the "naturalistic" feedback provided by the FES. While this is reasonable, couldn't the association of the antidromic stimuli caused by FES and the efferent command be the key (see, e.g., Taylor & Martin, J Neurosci 2009)? Or, perhaps more reasonably, the combination of both? I think this aspect at least needs to be discussed.
- b) In the authors' response to Comment 1.15, they state that "FES is only delivered when the patient is currently sustaining such SMR activity" -> But how could they be sure if a 2-s long FES train was delivered when motor intent was detected? More specifically, did the authors check that there was voluntary effort going on during the 2-s stimulation window? Since, as far as I know, there is no reason to think the association between the motor command and FES is only important at the moment the stimulation starts, this analysis could potentially explain why four patients from the BCI-FES group didn't improve functionally.
- c) "in particular via massive and timely recruitment of GTOs and muscle spindles" -> I don't think this statement is totally accurate. What evidence supports that the association between the SMR and FES was timely? And is it even a requirement? Have in mind that according to classic STDP studies in cell culture, animals and humans (e.g, the Taylor paper mentioned above or the work by Monica Perez), a few ms change in the relative timing between the two sets of stimuli changes the effect completely (although this has only been shown when associating a few pre-synaptic and post-synaptic spikes).

2. After reading the paper and the responses to the reviewers, I've realized I had not fully understood how the "sham" condition had been implemented during my first reading of the paper. I find the

analysis the authors present to address Comment 4.2 interesting, however I still have one concern related to item 1.a above. Accumulated evidence from a few studies using stimulation to assist the ongoing voluntary effort suggests that in the case of "continuous effort," the timing may not be extremely critical (compared to classic STDP-like paradigms). I am thinking about the work of the Courtine group, Everaert et al. *Neurorehabil Neural Repair* 2010, Popovic M et al *Neurorehabil Neural Repair* 2011 and a few other studies. Therefore, while the authors show that the likelihood that there's an ongoing voluntary effort at the time the FES is triggered is significantly (and largely) smaller for the sham group than for the BCI-FES group, I think they also need to show that the likelihood is significantly smaller during the whole 2-s stimulation period -as I guess it happened. Moreover, since the BCI threshold was selected manually, it can't be assumed that the FES was always started at the beginning of motor effort; maybe the classifier had to accumulate too much evidence causing a delay?

I think that showing more convincingly that there was "less association" between motor effort and FES in the sham group compared to the BCI-FES group is a critical step to further back up the main claims.

4. The paper and the response letter have claims that are not appropriately supported, or that are overstatements based on the current state of the art. Some examples are:

- Letter, page 7: "contingency metrics (...) that should lead to long term potentiation" -> potentiation of what? It's fair to speculate about potential mechanisms in the Discussion, but I don't think their successful functional results says anything definitive about the underlying mechanism. For example, how do the changes in scalp EEG rhythms relate to changes of the underlying circuitry?

- "More importantly, we are now reporting the correlation between online BCI metrics, clinical improvement and connectivity which imply the existence of activity-dependent plasticity" (from the letter, but similar ideas are in the paper) -> As I stated before, the observed EEG changes do not necessarily demonstrate any underlying structural changes: couldn't the patients be "learning" how to modulate their rhythms while keeping the neural circuitry mostly unaltered (as many animal studies suggest)? This should be discussed as a possibility not mentioned as a finding. Please, modify throughout the paper.

5. In relation with my previous comment, mathematical relationships that capture "flow of information" (as stated in the original DTF paper) do not necessarily reflect the actual connectivity of the brain, only the association between neural activity. Rephrase to "functional connectivity" or "flow of information" or some other term.

6. "Brain-controlled" vs. "Brain-triggered": Given that there are numerous examples of continuous control using both invasive and non-invasive signals, the term of brain control can be easily misleading. A possible alternative term that has been adopted by some in the field is "brain switch," if the authors do not want to use "triggered."

7. After reading the responses in the letter I have more doubts about how was the classifier decision threshold selected? I think I had mistakenly assumed that the patients did a few trials at the beginning of each session that served to guide the therapist's decision. The authors state: "the task difficulty in a way that it was hard but feasible (Taub et al., 1994), targeting an online performance of 10-12 FES, out of 15, per run." How could the therapist foresee this performance? Also, since as the authors acknowledge (comment 4.4) expert therapists could figure out what group patients belonged to, could this not lead to a bias? I've tried to find detailed information in the Paper and the Supplement, but did not. Am I missing something?

8. Regarding Comment 2.31: I appreciate the authors' usage of a different statistical test, but I still find this unconvincing given how the large inter-subject variability in both groups. I would do a multi-

fold cross-validation analysis, leaving out subsets of patients from each group and see how robust their results are. The same for the sLoreta analysis. I think it is fundamental given the small group sizes and the large inter-subject variability in most metrics.

9. There are a couple recent studies that raise important concerns about functional connectivity and coherence measures based on EEG. A very thorough simulation study by the Marinazzo group (Anzolin et al, BioRxiv 2018) shows that all projection algorithms and functionality connectivity measures have drawbacks. However, they also show that Linearly Constrained Minimum Variance (LCMV) beamforming generally outperforms eLORETA. Because of this and given that the authors projected their scalp EEGs in a "standard brain" rather than an accurate model of each patient's brain, I'd strongly suggest them to repeat this analysis with different algorithms: if the main results hold when they do this and multi-fold cross-validation I suggested above, that would make their observations more compelling. Also, the authors need to comment on whether the stroke may impact the performance of source localization algorithms, and provide references.

MINOR COMMENTS

- * Figure 2: are those LS fits significant? Most of them seem to be driven by a few data points.
- * In comment 2.27 the authors mention that increased task difficulty may have had a beneficial. Do they think that purposely increasing task difficulty could be beneficial in future studies?
- * "Additionally and critically, our intervention successfully decoupled BCI output from FES in the sham group." -> I think to show this decoupling, the need to analyze the SMR activity during the entire 2-s FES window.
- * Comment 2.8: 1) The authors rightfully propose that re-adjusting the decoder every day, as Ibáñez et al did, may add undesired variability. However, the opposite may be true: maybe, by doing so, a decoder that is calibrated every day captures the ongoing therapy-drive adaptation. It'll be interesting if the authors also commented on this in the Discussion.
- * In the paper, when talking about the improvements, do not only provide the mean +/- SD but also the ranges.
- * Figure 3 is missing the letters that signal the panels
- * "We speculate that the BCI-FES intervention strengthen CST projections in chronic patients that, initially, did not follow the proportional path." -> The proportional path?
- * "Our results put forward a mechanistic interpretation of our BCI intervention as they show how the necessary time contingency between FES and motor decoding" (...) "Another mechanism that might have played a key role is the recruitment of muscle spindles and Golgi tendon organs via FES." -> I thought the authors' main hypothesis was that afferent feedback was the key for inducing plasticity. Therefore, the activation of GTO and spindles is central to their mechanism; otherwise where does the feedback come from?
- * I think the authors should argue better why GTO activation can lead to better arm function. I can see the benefit for walking after a spinal cord injury, but for reaching after a brain stroke?

Reviewer #3 (Remarks to the Author):

The authors did a good job at addressing all of my main concerns. From my evaluation of their responses to the other reviewers, they also put a lot of effort into their responses and in improving the paper. I am happy with the current manuscript and would recommend publication.

Reviewer #4 (Remarks to the Author):

The initial manuscript was very interesting, but perhaps incomplete. The authors were very thorough in addressing the numerous reviewers' concerns, and as a result, have produced a much more complete and rigorous report.

I am satisfied with the authors' responses to all of my previously noted concerns. I believe that important questions remain about the exact mechanisms and conditions leading to the motor improvements, but this study provides an important indication that matching peripheral stimulation with motor effort is a determining factor in driving a useful plastic reorganization. I believe that the identification of this principle will have an important impact in guiding future research, as well as for the development of clinical applications. I would thus like to recommend this manuscript for publication.

My only minor concern relates to table 4: Authors list no false positive (FP = 0%) for the BCI group. They define the true negative rate as $TNR = TN / (TN+FP)$. Using this formula, TNR should be 1, for any nonzero TN value. Why is this not the case? Is there a mistake?

Reviewer #1

Comment 1.1: The study investigates the combined use of non-invasive (EEG) brain machine interface with neuromuscular electrical stimulation (NMES) to restore motor function to persons with chronic stroke. The study hypothesizes that BMI-NMES will enhance functional recovery because it is “tailored to reorganize the targeted neural circuits by providing rich sensory inputs via the natural afferent pathways so as to activate all spare components of the central nervous system involved in motor control...[and]...elicits functional movement and conveys proprioceptive and somatosensory information, in particular via massive and timely recruitment of muscle spindle feedback circuits”. The study compares the functional recovery and cortical plasticity of two groups: group 1 received BMI-triggered NMES, while another received sham NMES. The investigation shows statistically significant improvement of the BMI group and not the sham group based upon clinical functional metrics (Fugl-Meyer Upper Extremity, Muscle Strength measures, etc...). The BMI group also showed enhanced activity in mu and beta bands that were not present in the sham group. The investigators conclude that this study offers a novel use of BMI and NMES for stroke recovery in addition to elucidating the underlying mechanisms of this recovery.

Comments

No new major concerns. The authors have thoroughly addressed my previous concerns and have better clarified how the experimental paradigm directly addresses the stated hypothesis.

We are grateful to Reviewer 1 for the constructive feedback he provided to us, and for his contribution in improving the clarity of our hypothesis and the overall quality of our manuscript.

Reviewer #2

Comment 2.1: Hypothesis: After reading the authors' responses to all four reviewers, I'm slightly confused about the hypothesis of this study. In particular:

a) The authors state that the key is the association between the (presumably) ongoing efferent command and the "naturalistic" feedback provided by the FES. While this is reasonable, couldn't the association of the antidromic stimuli caused by FES and the efferent command be the key (see, e.g., Taylor & Martin, J Neurosci 2009)? Or, perhaps more reasonably, the combination of both? I think this aspect at least needs to be discussed.

We thank Reviewer 2 for pointing out this very important discussion point. We agree on the fact that both phenomena (i.e. naturalistic feedback via afferent pathways, and antidromic activations via efferent pathways) could contribute to the recovery of motor function. Non-invasive functional stimulation, as the one used in our protocol, provides a limited selectivity (spatial resolution), and can hardly dissociate both pathways, thus such differentiation was not attempted.

As stated in the introduction, the core of our hypothesis is that, irrespectively of the exact nature and composition of FES-induced effects, contingency between SMR detection and FES seems to be the "key" to recovery. We have modified the corresponding discussion to read *"While we believe all these mechanisms are likely to co-occur and contribute to recovery, our data are not sufficient to disentangle the exact role of each individual mechanism or their combination."*

Comment 2.2: In the authors' response to Comment 1.15, they state that "FES is only delivered when the patient is currently sustaining such SMR activity" -> But how could they be sure if a 2-s long FES train was delivered when motor intent was detected? More specifically, did the authors check that there was voluntary effort going on during the 2-s stimulation window? Since, as far as I know, there is no reason to think the association between the motor command and FES is only important at the moment the stimulation starts, this analysis could potentially explain why four patients from the BCI-FES group didn't improve functionally.

We apologize that the phrasing used in our reply to Comment 1.15 by Reviewer 1 ("FES is only delivered when the patient is currently sustaining such SMR activity"), has falsely conveyed the impression that FES delivery is, throughout its duration, coupled to sustained SMR modulation. In fact, our protocol has been designed to only guarantee that **there exists (sustained) SMR modulation at the onset of FES stimulation**. To be more precise, it is guaranteed that, at the end of a "hit" trial (i.e. a trial followed by FES delivery), the last PSD sample was classified as "motor attempt detected". Nevertheless, we are very grateful to the reviewer for the suggestion to analyse the FES period. Following this suggestion, we have extended our analysis of contingency metrics between SMR detection and FES from 1 sec before to 2 sec after the end of trial (i.e. during the whole FES stimulation period for "hit" trials). Please note that SMR detection requires one-second backwards window for the PSD computations, and that time $t=0$ corresponds to the analysis and values reported in the manuscript and coincides with FES onset for "hit" trials. Since subjects were instructed to proceed with sustained motor attempt, including during the FES and the corresponding functional hand extension movement, we anticipated that motor attempt detection and FES stimulation should co-occur to some extent. In order to make the last point clear, we have clarified and extended the description of the instructions in the material and methods section to add the sentence *"Moreover, participants were briefed to sustain the motor attempt during the FES stimulation until the full hand extension was achieved."*

Following our online SMR classification framework, we study the evolution of these contingency metrics (see Table 4 of the manuscript) over sliding windows shifted by 62.5 msec. Figure R1 in this reply (new Supplementary Figure S4 in the revised manuscript) presents the evolution of all metrics in question. It is evident that our protocol successfully induced differences in motor attempt-FES contingency between the two groups (BCI vs Sham) in favor of the BCI group **even well within the 2-sec-long FES interval**. However, it is also evident that the differences for most metrics **peaks at $t=0$** ,

i.e. the FES onset already reported in Table 4 of the manuscript (also in the revised manuscript).

Furthermore, the correlation of contingency metrics to motor recovery (difference of FMA Post-Pre, Δ FMA) is significant inside the 2-sec FES interval as well. Nonetheless, the correlation was maximum at the FES onset (Figure R2 in this reply, new Supplementary Figure S5 in the revised manuscript), as already reported in the manuscript.

Please note that it is very well documented that FES, especially stimulation above the motor threshold as employed in our study, induces SMRs related to the sensory FES and movement feedback (Reynolds et al., 2015). Thus, these induced SMRs may interfere with the SMRs associated to motor attempt, what might explain the drop in contingency-related metrics during FES. Second, in the case of those trials where FES was not delivered (timeout trials), which are also considered for this analysis, subjects may have induced movement or muscular artifacts, as the trial as such had ended. Thirdly, we cannot ignore the possibility that subjects may have gradually reduced the motor attempt effort after FES onset, despite the explicit instruction to avoid this. In sum, these new results have to be considered with caution.

We have included this analysis in the revised manuscript, with the figures appearing in the supplementary material (Supplementary Figures S4 and S5). We have modified the respective part of the Results to read *“In addition, analysis of all these aspects of contingency in 2 s long intervals around the FES onset shows that our experimental design has successfully coupled SMR activity to FES in the BCI group, also well into the FES period (Supplementary Fig. S4). Sustained contingency remains a good predictor of recovery in this interval (Supplementary Fig. S5). These results suggest that contingency (i.e., co-occurrence) of SMR and FES is critical, although precise timing between FES and SMR onsets is probably not a requirement.”*

Reynolds C, Osuagwu BA, Vuckovic A (2015). Influence of motor imagination on cortical activation during functional electrical stimulation. *Clin Neurophysiol* **126**, 1360–1369.

Figure R1: Contingency metrics between SMR detection and FES over sliding PSD windows in the interval [-1, 2] s surrounding the end of motor attempt trials (FES onset for “hit trials”). Mean and standard deviation are plotted for BCI and Sham groups for metrics: (a) TP, (b) TN, (c) FP, (d) FN, (e) TPR, (f) TNR, (g) PPV, (h) NPV, (i) Accuracy. Circular dots denote significant ($p < 0.05$) group (BCI vs Sham) differences for the respective metric at that time point. The time axis refers to the position of the right-side edge of the 1-sec long sliding PSD windows, so that $t=0$ is the PSD window at the FES onset.

Figure R2: Pearson correlations of motor recovery (ΔFMA) with contingency metrics over sliding PSD windows in the interval $[-1, 2]$ s surrounding the end of motor attempt trials (FES onset for “hit trials”). Metric type and statistical significance as denoted in the legend. The time axis refers to the position of the right-side edges of the 1-sec long sliding PSD windows, so that $t=0$ is the PSD window at the FES onset.

Comment 2.3: c) "in particular via massive and timely recruitment of GTOs and muscle spindles" -> I don't think this statement is totally accurate. What evidence supports that the association between the SMR and FES was timely? And is it even a requirement? Have in mind that according to classic STDP studies in cell culture, animals and humans (e.g. the Taylor paper mentioned above or the work by Monica Perez), a few ms change in the relative timing between the two sets of stimuli changes the effect completely (although this has only been shown when associating a few pre-synaptic and post-synaptic spikes).

Thank you for allowing us to clarify this point. By “timely”, we have referred in the manuscript (Table 4 and relevant discussions) to the demonstrated fact that, only in the BCI group, **the FES onset of hit trials always arrives while there is an ongoing SMR pattern** (reliable, as ensured by the application of our evidence accumulation approach). On the contrary, in the Sham group, FES arrived at random points, and thus mostly in the absence of concurrent SMR activity. Speculating that the underlying mechanisms promoted by our intervention are akin to associative learning, and prompted by previous reviewers’ comments, we have assumed that such timing might be crucial. Indeed, in the light of the new analysis requested by this reviewer (see our reply to Comment 2.2), FES onset is where the group differences of most of the metrics we have employed to evaluate contingency is greatest (new Supplementary Figure S6 and Figure R1 in this reply) and their correlation with motor recovery peaks (new Supplementary Figure S7 and Figure R2 in this reply). Hence, certain time contingency seems to be important.

However, as both counterparts of contingency in this case (SMR and FES) are events of certain duration, we agree with the reviewer’s opinion (as stated in Comments 2.3 and 2.4) that extremely precise timing (i.e. like that shown to be critical for STDP) is probably not a requirement. Indeed, the same figures show that contingency is larger in favor of the BCI group and many of the respective metrics still explain FMA Post-Pre difference **also well into the FES interval**. In the light of these new analyses, we have removed the word “timely” from this sentence.

Comment 2.4: After reading the paper and the responses to the reviewers, I've realized I had not fully understood how the "sham" condition had been implemented during my first reading of the paper. I find the analysis the authors present to address Comment 4.2 interesting, however I still have one concern related to item 1.a above. Accumulated evidence from a few studies using stimulation to assist the ongoing voluntary effort suggests that in the case of "continuous effort," the timing may not be extremely critical (compared to classic STDP-like paradigms). I am thinking about the work of the

Courtine group, Everaert et al. Neurorehabil Neural Repair 2010, Popovic M et al Neurorehabil Neural Repair 2011 and a few other studies. Therefore, while the authors show that the likelihood that there's an ongoing voluntary effort at the time the FES is triggered is significantly (and largely) smaller for the sham group than for the BCI-FES group, I think they also need to show that the likelihood is significantly smaller during the whole 2-s stimulation period -as I guess it happened. Moreover, since the BCI threshold was selected manually, it can't be assumed that the FES was always started at the beginning of motor effort; maybe the classifier had to accumulate too much evidence causing a delay? I think that showing more convincingly that there was "less association" between motor effort and FES in the sham group compared to the BCI-FES group is a critical step to further back up the main claims.

As our analysis reported in Comment 2.2 shows, and as also commented in our reply to Comment 2.3, the reviewer's intuition is indeed verified: our paradigm's design has effectively ensured contingency between SMR activity and FES in the BCI group also well into the FES period (Figure R1).

In our paradigm, we did not want to reward the beginning of motor effort (motor attempt in our terminology), but rather the elicitation of SMR patterns during motor attempt. Furthermore, we apply evidence accumulation in order to minimize false positives, what inevitably causes some delays between emergence of SMR and robust detection of such SMR that triggers FES. As the reviewer suggests, precise timing is probably not critical, which is also suggested by the new results.

Comment 2.5: The paper and the response letter have claims that are not appropriately supported, or that are overstatements based on the current state of the art. Some examples are:

- Letter, page 7: "contingency metrics (...) that should lead to long term potentiation" -> potentiation of what? It's fair to speculate about potential mechanisms in the Discussion, but I don't think their successful functional results says anything definitive about the underlying mechanism. For example, how do the changes in scalp EEG rhythms relate to changes of the underlying circuitry?

We agree with the reviewer that although our results provide evidence that the proposed intervention delivers clinical outcomes by inducing functional activity-dependent plasticity, it is by no means definite and conclusive, due to the intrinsic limitations of EEG imaging. Most importantly, the available data cannot substantiate the existence and nature of any effect down to the level of cells. In the revised manuscript, we have further toned down of any claims made regarding the underlying mechanisms and underlined the speculative nature of the respective discussions. On this particular point, we have replaced "long-term potentiation" with "functional activity-dependent plasticity".

Comment 2.6: "More importantly, we are now reporting the correlation between online BCI metrics, clinical improvement and connectivity which imply the existence of activity-dependent plasticity" (from the letter, but similar ideas are in the paper) -> As I stated before, the observed EEG changes do not necessarily demonstrate any underlying structural changes: couldn't the patients be "learning" how to modulate their rhythms while keeping the neural circuitry mostly unaltered (as many animal studies suggest)? This should be discussed as a possibility not mentioned as a finding. Please, modify throughout the paper.

We agree that the presented data only provide evidence of functional plasticity undergoing the proposed intervention (which, furthermore, suffers all limitations related to EEG imaging). No evidence of structural plasticity is offered whatsoever. It has never been our intention to claim the demonstration of structural plasticity and we apologize that our previous manuscripts might have conveyed this misconception.

In the revised version of the manuscript we refer to "activity-dependent plasticity", which does not necessarily imply structural plasticity, as it might also well be a reference to functional plasticity (as in our case). In this respect, functional plasticity has been shown to modify behaviour, hence, assuming that it is a cause and not an epiphenomenon of modified behavior, it is not unlikely that it can also promote motor recovery (i.e. by exploiting spared neural circuits). Also, although we by no means want to raise a claim for structural plasticity effects in our revised manuscript, it has been early shown that functional plasticity usually accompanies structural reorganization (even in the same population studied in our work, see for instance Schaechter et al., 2006). Therefore, as suggested by the reviewer,

although we cannot substantiate undergoing structural plasticity, we can also not exclude it.

We have modified the corresponding paragraph in the discussion of the revised manuscript to read: *“Even though the identified post-intervention changes in electrophysiology are not a proof of an underlying cortical reorganization, they are highly indicative of functional plasticity mechanisms accompanying the proposed intervention and potentially promoting motor recovery. In turn, functional plasticity has been shown to be associated with cortical reorganization (Schaechter et al., 2006). Hence, while we can offer no direct evidence of structural plasticity effects undergoing the proposed therapy, we speculate that this would be a reasonable expectation in the light of our findings.”*

Schaechter JD, Moore CI, Connell BD, Rosen BR, Dijkhuizen RM (2006). Structural and functional plasticity in the somatosensory cortex of chronic stroke patients. *Brain* **129**, 2722–2733.

Comment 2.7: In relation with my previous comment, mathematical relationships that capture "flow of information" (as stated in the original DTF paper) do not necessarily reflect the actual connectivity of the brain, only the association between neural activity. Rephrase to "functional connectivity" or "flow of information" or some other term.

We agree with this point as, indeed, we always meant “functional connectivity”. We have corrected it everywhere.

Comment 2.8: "Brain-controlled" vs. "Brain-triggered": Given that there are numerous examples of continuous control using both invasive and non-invasive signals, the term of brain control can be easily misleading. A possible alternative term that has been adopted by some in the field is "brain switch," if the authors do not want to use "triggered."

We understand the reviewer’s point. We have replaced “brain-controlled” by “brain-actuated” (we apologize that our corresponding reply was not updated with this change and has caused confusion). We find that “brain-actuated” is a good compromise, as it is more generic and all-inclusive. Please, note that we find “brain-triggered” to be rather unrepresentative of the functional role the brain has in the proposed intervention: It is important that the brain is not only providing the trigger for the FES, but it is also the organ receiving the feedback and adapting in order to induce recovery. Hence, it is not simply one among other equivalent options for triggering FES (e.g. unaffected hand, the therapist, EMG-trigger, etc.), what might be implied by the term “brain-triggered”.

Comment 2.9: After reading the responses in the letter I have more doubts about how was the classifier decision threshold selected? I think I had mistakenly assumed that the patients did a few trials at the beginning of each session that served to guide the therapist's decision. The authors state: "the task difficulty in a way that it was hard but feasible (Taub et al., 1994), targeting an online performance of 10-12 FES, out of 15, per run." How could the therapist foresee this performance? Also, since as the authors acknowledge (comment 4.4) expert therapists could figure out what group patients belonged to, could this not lead to a bias? I've tried to find detailed information in the Paper and the Supplement, but did not. Am I missing something?

We are sorry to the reviewer that our last answer has been confusing. The applied principle was that the therapist started in the first run of each session (day) with the same threshold as used in the last run of the previous session. The first run of the first session started with a fixed threshold (0.75). The threshold was increased/decreased by the therapist at the end of each run if the patient’s achieved hit rate was too good/bad, respectively. By adjusting this threshold after each run, the therapist was trying to bring about a targeted “performance” of 10-12 hits out of 15 trials at the next run. Although it has been clearly untenable to reach the desired hit rate on each individual run, especially given the subjects’ anticipated BCI performance fluctuations that obviously affect the hit rate and cannot be predicted by the therapist, we did manage to have on average performances in the range of 10-12 out of 15, to keep the motivation and challenge for each participant as high as possible. The exact same principle was applied in the sham group, and we ensured that there was no bias between groups.

The new text replaces “at each run” with “after each run” and reads now *“The confidence threshold that triggered FES was adjusted after each run for each patient by the therapist, so as to shape the task difficulty in a way that it was hard but feasible (Taub et al., 1994), targeting an average online*

performance of 10-12 FES, out of 15, per run.”

Comment 2.10: Regarding Comment 2.31: I appreciate the authors' usage of a different statistical test, but I still find this unconvincing given how the large inter-subject variability in both groups. I would do a multi-fold cross-validation analysis, leaving out subsets of patients from each group and see how robust their results are. The same for the sLoreta analysis. I think it is fundamental given the small group sizes and the large inter-subject variability in most metrics.

We understand the reviewer's reservations, and would like to thank him for further pointing this problem and suggesting a new analysis. Regarding the sLoreta results, it is unfortunately not possible to perform a cross-validation analysis due to software limitations. We have thus lessened the claims of such findings and moved the sLoreta analysis to the supplementary materials.

Regarding the connectivity results, we have performed additional analyses to strengthen our hypothesis that connectivity changes are correlated with improvement in the FMA scores. Particularly, and following the reviewer's suggestion, we have analyzed the results from three perspectives: first, a model robustness analysis, where we perform a cross-validation on the goodness of the model; second, a confidence analysis, where we evaluate the confidence of the results obtained; and third, a predictive analysis, where we evaluate the predictive capabilities of our models again using cross-validation.

Original analysis in the manuscript: for comparison reasons, we detail here the Pearson correlation values between Δ Connectivity and Δ FMA scores reported in the manuscript: $r_{\mu} = 0.41$, $p=0.045$; $r_{\beta} = 0.48$, $p=0.02$. Please note that we have noticed a typo on the original results reported for the beta band in the manuscript. The values reported here are the correct ones. We apologize for this issue.

Model robustness evaluation: We evaluated the correlations obtained using cross-validation. Due to the low number of examples, we chose leave-one-out cross validation, where each fold is composed of all the samples but one, thus leading to as many folds as samples. The correlations obtained with this evaluation where (MEAN \pm SEM): $r_{\mu} = 0.42 \pm 0.006$ (12 out of 24 correlations are significant ($p<0.05$); minimum $p = 0.01$, maximum $p = 0.08$); $r_{\beta} = 0.48 \pm 0.007$ (23 out of 24 correlations are significant ($p<0.05$); minimum $p = 0.005$, maximum $p = 0.06$). This result, together with the Spearman correlation values reported in our previous response document, substantiates that the correlation was not driven by single outliers in the data.

Confidence analysis: We evaluated whether our obtained correlation values were actually significantly different from 0, and with which confidence. To this end, we performed a bootstrapping with replacement (5000 iterations) to build a distribution of correlation values, and extract the confidence values from such distributions (see Figure R3 below and Supplementary Figure S7 in the revised manuscript). The confidence intervals at 95% for both correlations were of $r_{\mu} \in [0.13, 0.64]$ and $r_{\beta} \in [0.07, 0.70]$ and thus significantly different from 0 at $\alpha=0.05$.

Predictive analysis: to strengthen the validity of the results obtained, we built linear models to assess the predictive capabilities of FMA scores based on connectivity. Similarly to the FMA-accuracy model already reported in the manuscript, we built regressors using leave-one-out cross validation that were tested on the remaining testing example, and evaluated using R^2 . With this approach, the results obtained were of $R^2(\mu) = 0.24$; $R^2(\beta) = 0.26$, and thus connectivity of μ and β bands alone explained 25% of the total variance of the FMA scores. Interestingly, a linear model considering both μ and β connectivity frequency bands boosted even more the variance explained, $R^2(\mu, \beta) = 0.36$.

Figure R3: Correlation bootstrapping distributions (histogram, based on 5000 permutations with replacement) for the mu (left) and beta (right) frequency bands. Black thick lines represent the lower and upper confidence bounds at 95%.

In sum, we believe that these additional analysis further support the validity of our results and conclusions drawn from them. We have included these analyses in the Supplementary Material, and mention them in the revised manuscript.

Comment 2.11: There are a couple recent studies that raise important concerns about functional connectivity and coherence measures based on EEG. A very thorough simulation study by the Marinazzo group (Anzolin et al, BioRxiv 2018) shows that all projection algorithms and functionality connectivity measures have drawbacks. However, they also show that Linearly Constrained Minimum Variance (LCMV) beamforming generally outperforms eLORETA. Because of this and given that the authors projected their scalp EEGs in a "standard brain" rather than an accurate model of each patient's brain, I'd strongly suggest them to repeat this analysis with different algorithms: if the main results hold when they do this and multi-fold cross-validation I suggested above, that would make their observations more compelling. Also, the authors need to comment on whether the stroke may impact the performance of source localization algorithms, and provide references.

All effective connectivity analysis presented in our original and revised manuscripts is done directly on scalp channel locations, thereafter averaged to monitor effective connectivity among different scalp regions of interest. We have explicitly specified this in the Materials and Methods of the revised manuscript. There is no connectivity analysis in our work done on the projected space of sLoreta or other inverse method. We agree that inverse methods have important limitations, especially once the projections are used for further analysis. However, the particular limitations mentioned here by the reviewer do not apply in the case of our analysis. We have only employed sLoreta to study SMR activity in deeper sources. As mentioned in our reply to Comment 2.10, we have lessened the claims related to the latter analysis and moved it to Supplementary Material.

Comment 2.12: Figure 2: are those LS fits significant? Most of them seem to be driven by a few data points.

We assume the reviewer is referring to the correlations between ΔFMA and $\Delta Connectivity$ illustrated in the manuscript (Figure 3, there are no linear fits presented in Figure 2). Please, refer to our reply to Comment 2.10 in this document.

Comment 2.13: In comment 2.27 the authors mention that increased task difficulty may have had a beneficial. Do they think that purposely increasing task difficulty could be beneficial in future studies? This methodology is well established with respect to robotic-assisted rehabilitation under the terminology "assistance-as-needed" (where task difficulty is modulated through the level of assistance provided by an assistive device) and has been already applied to novel treatment regimes (Marchal-Crespo and Reinkensmeyer, 2009; Louie and Eng, 2016). We definitely think that this notion needs to be further explored and exploited in future BCI-based rehabilitation studies and we believe we have

provided a first implementation here through the confidence threshold adjustment. As this issue is marginally inside the scope of our manuscript and it is already briefly mentioned in the Discussion, we opted to not proceed with further amendments regarding this comment.

Marchal-Crespo L, Reinkensmeyer DJ (2009). Review of control strategies for robotic movement training after neurologic injury. *J Neuroeng Rehabil* **6**, 20.

Louie DR, Eng JJ (2016). Powered robotic exoskeletons in post-stroke rehabilitation of gait: A scoping review. *J Neuroeng Rehabil* **13**, 1:53.

Comment 2.14: "Additionally and critically, our intervention successfully decoupled BCI output from FES in the sham group." -> I think to show this decoupling, the need to analyze the SMR activity during the entire 2-s FES window.

Please, see our reply to Comment 2.2.

Comment 2.15: Comment 2.8: 1) The authors rightfully propose that re-adjusting the decoder every day, as Ibáñez et al did, may add undesired variability. However, the opposite may be true: maybe, by doing so, a decoder that is calibrated every day captures the ongoing therapy-drive adaptation. It'll be interesting if the authors also commented on this in the Discussion.

Indeed, the (beneficial or detrimental) effects of timing and intensity of adaptation have not been well studied neither in the framework of BCI skill acquisition nor, even more, in the context of rehabilitation. We still think that even if some adaptation proves to be beneficial, there exists no definite knowledge on the subject as of now and we have essentially opted for the "safest" choice in our own study.

We have modified the respective part in the revised manuscript to more elaborately discuss this issue: *"This might be related to the fact that authors calibrated the whole BCI decoder (including feature selection) at each session in order to optimize BCI performance. Parsimonious adaptation of the BCI decoder might be beneficial by capturing and exploiting the evolution of functional plasticity during recovery. However, very frequent recalibration might overall hinder plasticity since continuously changing BCI features substantially modify the recruited efferent pathways participating in eventual activity-dependent plasticity mechanisms. The effects of timing and intensity of adaptation in BCI-based rehabilitation paradigms is a little studied topic that warrants further investigation."*

Comment 2.16: In the paper, when talking about the improvements, do not only provide the mean +/- SD but also the ranges.

We have added the ranges in all such occurrences of the clinical outcomes in the result section and in Table 3 (minimum and maximum improvement in square brackets after the mean/STD).

Comment 2.17: Figure 3 is missing the letters that signal the panels.

Thank you for spotting this inconsistency. We have added all panel labels to the figure.

Comment 2.18: We speculate that the BCI-FES intervention strengthen CST projections in chronic patients that, initially, did not follow the proportional path." -> The proportional path?

We have replaced "proportional path" by "proportional recovery rule".

Comment 2.19: "Our results put forward a mechanistic interpretation of our BCI intervention as they show how the necessary time contingency between FES and motor decoding" (...) "Another mechanism that might have played a key role is the recruitment of muscle spindles and Golgi tendon organs via FES." -> I thought the authors' main hypothesis was that afferent feedback was the key for inducing plasticity. Therefore, the activation of GTO and spindles is central to their mechanism; otherwise where does the feedback come from?

We apologize that our phrasing in the manuscript has been confusing. Indeed, activation of spindles and GTOs are the major component of "rich afferent feedback" and are by no means "another mechanism". We have modified this sentence to read *"As rich afferent feedback is central in our hypothesis, the recruitment of muscle spindles and Golgi tendon organs via FES might have played a key role."*

Comment 2.20: I think the authors should argue better why GTO activation can lead to better arm function. I can see the benefit for walking after a spinal cord injury, but for reaching after a brain stroke?

The reviewer is right: while during locomotion GTO input leads to a di-synaptic excitation of extensor motoneurons, for upper limb movements the GTO input is inhibitory –leading to the inhibition of the homonymous muscle. However, our argument only highlights the role of GTO as a proprioceptive sensory receptor organ that senses changes in muscle tension.

We have kept GTO in the revised manuscript (in particular in the 3rd paragraph of the Discussion, page 18), but we can remove it if the reviewer thinks is more appropriate.

Reviewer #3

Comment 3.1: The authors did a good job at addressing all of my main concerns. From my evaluation of their responses to the other reviewers, they also put a lot of effort into their responses and in improving the paper. I am happy with the current manuscript and would recommend publication.

We are grateful to Reviewer 3 for the constructive feedback he provided to us, and for acknowledging our effort to comprehensively and transparently address the responses of all reviewers. We believe that this review process allowed us to improve the clarity and overall quality of our manuscript.

Reviewer #4

Comment 4.1: The initial manuscript was very interesting, but perhaps incomplete. The authors were very thorough in addressing the numerous reviewers' concerns, and as a result, have produced a much more complete and rigorous report. I am satisfied with the authors' responses to all of my previously noted concerns. I believe that important questions remain about the exact mechanisms and conditions leading to the motor improvements, but this study provides an important indication that matching peripheral stimulation with motor effort is a determining factor in driving a useful plastic reorganization. I believe that the identification of this principle will have an important impact in guiding future research, as well as for the development of clinical applications. I would thus like to recommend this manuscript for publication.

We are grateful to Reviewer 4 for acknowledging the improvement of the completeness and rigorousness of our manuscript. We agree with the Reviewer about the fact that further research is needed to clarify the mechanisms and conditions leading to motor improvements in similar scenarios. We share the hope that our study will contribute with additional evidence and facilitate further developments in this direction.

Comment 4.2: My only minor concern relates to table 4: Authors list no false positive (FP = 0%) for the BCI group. They define the true negative rate as $TNR = TN / (TN+FP)$. Using this formula, TNR should be 1, for any nonzero TN value. Why is this not the case? Is there a mistake?

We apologize for this inconsistency and sincerely thank the reviewer for spotting this error. An erroneous transfer of some of the FP results (non-zero values mistakenly used for a few subjects in the computation of some of these results) in our analysis software has affected the results we have shown in Table 4 of the manuscript for this metric and those based on it: TNR, PPV, Accuracy. We have updated Table 4 in the revised manuscript with the correct figures. It is apparent that any changes are extremely minor (and more in favor of the points made) and have absolutely no impact on the claims made. Please, also note that, as a result of the analysis requested by Reviewer 2, we now report these contingency metrics and their correlation to FMA improvement not only at the FES onset (Table 4, which we have maintained), but also in a period around it (new Supplementary Figures S4 and S5 and Figures R1 and R2 in this reply).

REVIEWERS' COMMENTS:

Reviewer #2 (Remarks to the Author):

Millán and colleagues have carefully addressed the majority of my previous concerns about their paper. I think that the manuscript in its current form highlights better their interesting clinical results, without overstating the mechanistic insights; I also think the Discussion has improved. I too appreciate the additional information provided by the new analyses. I only have a few minor comments:

- Figure R2: is the significant level at the bottom (colored squares) the most conservative threshold across all metrics for that time sample?
- I still don't understand why the authors can't cross-validate their sLoreta analysis, although I couldn't find detailed information on their toolbox. Couldn't the authors just input a subset of BCI and sham patients to their toolbox, identify the differences, and then take other two different subsets of patients, and so on? This should let them assess how stable their results are.
- I had some problems following their bootstrap analysis. Perhaps I missed some details, but I think the authors should explain their analysis more clearly.
- The "proportional recovery rule" needs a reference or some background for people outside the stroke rehab field
- Methods: define SdDTF
- I should have brought this up before, but it'd be great if the authors could show data similar to Figure 4b for a couple example patients. I presume it'll be noisy, but it would be informative. It could very well go into the Supplement
- "FES recruits muscle spindles and GTOs, activates them faster, and conveys also richer somatosensory information" -> Add a reference.

Reply to Reviewers

Reviewer #2

Comment 2.1: Millán and colleagues have carefully addressed the majority of my previous concerns about their paper. I think that the manuscript in its current form highlights better their interesting clinical results, without overstating the mechanistic insights; I also think the Discussion has improved. I too appreciate the additional information provided by the new analyses. I only have a few minor comments:

We thank Reviewer 2 for acknowledging the merits of our revision and helping us to further improve our manuscript. Please, find below our responses to these comments.

Comment 2.2: Figure R2: is the significant level at the bottom (colored squares) the most conservative threshold across all metrics for that time sample?

Indeed, symbols in red denote statistical significant correlation of the corresponding variables with recovery at the most conservative confidence interval tested ($p < 0.01$). In the revised Supplementary Information, we have slightly modified the legend of Supplementary Figure 5 in order to make this point clearer. Specifically, we have replaced squares with lines (of the corresponding color) in the legend, so as to show that the p-values refer to all kinds of variables (which is reflected in the symbols' shape) and not only to squares (which illustrate False Negatives, FN).

Comment 2.3: I still don't understand why the authors can't cross-validate their sLoreta analysis, although I couldn't find detailed information on their toolbox. Couldn't the authors just input a subset of BCI and sham patients to their toolbox, identify the differences, and then take other two different subsets of patients, and so on? This should let them assess how stable their results are.

We have proceeded with the suggested test in the revised Supplementary Information (page 6) document using leave-one-subject-out cross-validation. For all the folds except for one, the same MNI coordinates were found, and the results were significant for 9/12 (75%) of the folds. In summary, neural desynchronization at μ and β frequency bands in the regions of interest were larger (significant and robust across subjects in the case of μ) for the BCI group compared to the sham group after intervention.

Comment 2.4: I had some problems following their bootstrap analysis. Perhaps I missed some details, but I think the authors should explain their analysis more clearly.

We have revised the corresponding section of the Supplementary Information (page 4) providing further details and we are confident that the final version is considerably clarified.

Comment 2.5: The "proportional recovery rule" needs a reference or some background for people outside the stroke rehab field

Done. Please, see page 11.

Comment 2.6: Methods: define SdDTF

Done. Please, see page 20.

Comment 2.7: I should have brought this up before, but it'd be great if the authors could should data similar to Figure 4b for a couple example patients. I presume it'll be noisy, but it would be informative. It could very well go into the Supplement

Done. We have added the new Supplementary Figure 2 with the data of 4 exemplary patients.

Comment 2.8: “FES recruits muscle spindles and GTOs, activates them faster, and conveys also richer somatosensory information” -> Add a reference.

Since we have not found any direct reference to support this statement, we have slightly changed it and added a reference:

“However, FES depolarizes more motor and sensory axons, sending larger sensory volleys from muscle spindles and Golgi tendon organs into the CNS^[51].” (page 11)